# *Tnfaip2/exoc3*-driven lipid metabolism is essential for stem cell differentiation and organ homeostasis

Sarmistha Deb[1] , Daniel A Felix[1], Philipp Koch[1] , Maharshi Krishna Deb[1] , Karol Szafranski[1] , Katrin Buder[1], Mara Sannai[1], Marco Groth[1] , Joanna Kirkpatrick[1], Stefan Pietsch[1], André Gollowitzer[2,3] , Alexander Groß[4] , Philip Riemenschneider[1], Andreas Koeberle[2,3] , Cristina González-Estévez[1,*] & Karl Lenhard Rudolph[1,5,**]

## Abstract

Lipid metabolism influences stem cell maintenance and differentiation but genetic factors that control these processes remain to be delineated. Here, we identify *Tnfaip2* as an inhibitor of reprogramming of mouse fibroblasts into induced pluripotent stem cells. *Tnfaip2* knockout impairs differentiation of embryonic stem cells (ESCs), and knockdown of the planarian para-ortholog, *Smed-exoc3*, abrogates *in vivo* tissue homeostasis and regeneration—processes that are driven by somatic stem cells. When stimulated to differentiate, *Tnfaip2*-deficient ESCs fail to induce synthesis of cellular triacylglycerol (TAG) and lipid droplets (LD) coinciding with reduced expression of vimentin (*Vim*)—a known inducer of LD formation. *Smed-exoc3* depletion also causes a strong reduction of TAGs in planarians. The study shows that *Tnfaip2* acts epistatically with and upstream of *Vim* in impairing cellular reprogramming. Supplementing palmitic acid (PA) and palmitoyl-L-carnitine (the mobilized form of PA) restores the differentiation capacity of *Tnfaip2*-deficient ESCs and organ maintenance in *Smed-exoc3*-depleted planarians. Together, these results identify a novel role of *Tnfaip2* and *exoc3* in controlling lipid metabolism, which is essential for ESC differentiation and planarian organ maintenance.

**Keywords** *Exoc3*; lipid metabolism; organ homeostasis; stem cell differentiation; *Tnfaip2*

**Subject Categories** Metabolism; Stem Cells & Regenerative Medicine

## Introduction

Lipid metabolism is increasingly recognized as a key process required for the maintenance and the differentiation capacity of stem cells. In *Drosophila*, genetic impairments in mitochondrial lipid metabolism result in loss of germline stem cells, which is rescued by enhancing mitochondrial lipid metabolism (Sênos Demarco *et al*, 2019). In addition, the generation of enteroendocrine cells from intestinal stem cells in the *Drosophila* gut was shown to be modulated by dietary lipids impinging on sterol levels and Notch signaling (Obniski *et al*, 2018). Similarly, in vertebrates, mitochondrial lipid metabolism is important for the maintenance of stem cells, such as hematopoietic stem cells (Gurumurthy *et al*, 2010; Ito *et al*, 2012). In addition, lipid metabolites can also contribute to the instruction of differentiation. For example, epoxyeicosatrienoic acids (EETs) enhance bone marrow-derived hematopoiesis (Lahvic *et al*, 2018) and phosphatidylinositol (PtdIns) transfer proteins (PITPs) stimulate a golgi lipid signaling pathway controlling cell polarity during neurogenesis (Xie *et al*, 2018).

The important role for lipid metabolites in differentiation and self-renewal of stem cells was also supported by the finding on cultured pluripotent cells. Autotaxin-dependent autocrine stimulation of lipid signaling along with LIF/BMP4 signaling induces the entry of iPSCs into a naïve stage (Kime *et al*, 2016). In contrast, fatty acids deprivation from culture medium by itself suffices to impair differentiation of naïve ESCs into a primed stage thus enabling the prolonged culture of naïve ESCs (Cornacchia *et al*, 2019). Together, these data imply that lipid metabolism influences the differentiation and self-renewal of pluripotent stem cells as well as somatic stem cells in tissues of the organism. Genetic factors that control lipid metabolism and the function of stem cells in organ maintenance and regeneration remain yet to be delineated in greater detail. Such studies could improve our understanding of alterations in stem cell function that affect tissue homeostasis and disease development in response to changes in lipid metabolism that occur during aging, metabolic diseases or in response to changes in diets.

Here, we reasoned that stable iRNA screening may enable the identification of genes that influence the transition of somatic cells into iPSCs. Assuming that some of these genes would also be

1   Leibniz Institute on Aging – Fritz Lipmann Institute e.V., Jena, Germany
2   Institute of Pharmacy, Friedrich-Schiller-University, Jena, Germany
3   Michael Popp Institute and Center for Molecular Biosciences Innsbruck (CMBI), University of Innsbruck, Innsbruck, Austria
4   Institute of Medical Systems Biology, Ulm University, Ulm, Germany
5   University Hospital Jena, Friedrich Schiller University, Jena, Germany
    *Corresponding author. Tel: +49 3641 656628; Fax: +49 3641 656351; E-mail: Cristina.Gonzalez.Estevez@gmail.com
    **Corresponding author. Tel: +49 3641 6818; Fax: +49 3641 6595818; E-mail: Lenhard.Rudolph@Leibniz-Fli.de

important for the function of somatic stem cells *in vivo*, planarians were employed as a model to identify genes that influence the maintenance and regeneration of tissues *in vivo*. Our study revealed that the knockdown of *Tnfaip2*—a downstream target of TNFα/NFκB signaling (Chen *et al*, 2014)—enhances reprogramming of mouse fibroblasts into iPSCs. The study shows that *Tnfaip2* and its planarian para-ortholog, *Smed-exoc3*, are essential for the differentiation of murine embryonic stem cells (ESCs) in culture and for the *in vivo* maintenance of tissue homeostasis and regeneration in planarian, respectively. Mechanistically, the study reveals that *Tnfaip2* induces *Vim*-dependent lipid droplet (LD) formation and increases cellular triacylglycerol (TAG) content in early stages of ESC differentiation. In line with the conserved functionality between *Tnfaip2* and *Smed-exoc3*, significant reductions in TAG levels were also observed in *Smed-exoc3*-depleted planarians. The administration of fatty acids, palmitic acid (PA), and its mitochondrial carrier, palmitoyl-L-carnitine (PC), restores differentiation capacity of *Tnfaip2*-deficient ESCs as well as organ maintenance in *Smed-exoc3*-deficient planarians *in vivo*. Together, these results provide experimental evidence for an essential role of *Tnfaip2/Smed-exoc3* in promoting ESC differentiation as well as organ homeostasis and regeneration in planarians by regulating lipid metabolism.

## Results

### shRNA screening of cancer-related genes identifies knockdown of *Tnfaip2* as an enhancer of reprogramming of murine embryonic fibroblasts (MEFs) into induced pluripotent stem cells (iPSCs)

To identify genes that may regulate cell plasticity and differentiation, an shRNA-mediated gene knockdown screen was conducted during reprogramming of MEFs into iPSCs. We employed a previously described shRNA library consisting of 1,772 sequence-verified miR30-based shRNAs targeting around 1,000 putative cancer-related genes (Wang *et al*, 2012). MEFs were prepared from three independent mouse embryos of C57BL/6J mice at day 13.5 post coitum (E13.5). MEFs were co-transduced at passage-3 with (i) a polycistronic vector expressing an mCherry reporter along with the four Yamanaka factors (4-Factors = 4F: *Oct4*, *Sox2*, *Klf4*, and *c-Myc* = OSKM, Sommer *et al*, 2009), and (ii) a vector expressing a GFP reporter along with the above-mentioned shRNA library. Four days after infection, double positive (GFP$^+$/mCherry$^+$) MEFs were FACS-sorted and were allowed to reprogram for another 14 days toward the generation of iPSCs.

Reprogrammed iPSCs that have silenced the transgene expression (mCherry-negative) and expressed the pluripotency marker, SSEA1, were used for the study. To identify shRNAs that enhanced the reprogramming process, we sorted reprogrammed cells (SSEA1$^+$, mCherry$^-$) and non-reprogrammed cells (SSEA1$^-$, mCherry$^+$). Deep sequencing analysis identified shRNAs that exhibited a differential prevalence in iPSCs compared to non-iPSCs (Fig 1A, Dataset EV1). Aiming to include planarian as an experimental *in vivo* system to study whether candidate genes may also influence the function of somatic stem cells, follow-up experiments focused on *Tnfaip2* shRNA since it was one of the top candidates in the reprogramming

screen (Fig 1A, Dataset EV1) and, moreover, had an orthologous gene in planarian (see below).

To test the reproducibility of the knockdown effect of *Tnfaip2* on reprogramming, a second shRNAs against *Tnfaip2* (AGAGATTTCTTTTTTATAT) was designed using the SplashRNA software (Pelossof *et al*, 2017). Both shRNAs showed a knockdown efficiency of *Tnfaip2* mRNA expression of >70% (Fig EV1A). MEFs were isolated from *Oct4*-eGFP transgenic mice and were cotransfected with (i) a lentivirus expressing either one of the shRNAs against *Tnfaip2* along with a BFP reporter, and (ii) a lentivirus harboring a polycistronic OSKM cassette co-expressing an mCherry reporter. Alkaline phosphatase (AP) staining of reprogrammed colonies or FACS analysis of *Oct4*-GFP reporter expression (Lengner *et al*, 2007) revealed that depletion of *Tnfaip2* led to a 2- to 3-fold elevation in the reprogramming efficiency (Fig 1B–E). The knockdown of *Tnfaip2* also led to enhanced reprogramming when only three Yamanaka factors (= 3F: *Oct4*, *Sox2*, and *Klf4* = OSK) were used in the absence of *c-Myc* (Fig 1B and C). Of note, induction of pluripotency with three factors generated a similar number of iPSC colonies in *Tnfaip2* knockdown MEFs as reprogramming induced by four factors (OSKM) in control MEFs without the knockdown of *Tnfaip2* (Fig 1B and C). FACS analysis of MEFs from *Oct4*-eGFP reporter mice (Lengner *et al*, 2007) showed that the percentage of iPSCs obtained 18 days after transfection of four factors (OSKM) was on average 11.5% ± 3% for control shRNA (shScr)-infected MEFs (Fig 1D and E). This result was in the range of previous reports of numbers of iPSCs that can be generated under the described conditions from a similar polycistronic lentiviral vector (Cheloufi *et al*, 2015). The knockdown of *Tnfaip2* by two independent shRNAs (shTnfaip2#1 and shTnfaip2#2, see above) elevated the reprogramming efficiency (measured by reactivation of the *Oct4*-eGFP reporter) to 29.2% ± 4% and 25.8% ± 3% for shTnfaip2#1 and shTnfaip2#2, respectively (Fig 1D and E, *n* = 5 repeats per shRNA).

To analyze at what time point during reprogramming, inhibitory effects of *Tnfaip2* on pluripotency induction may get induced, the mRNA expression of *Tnfaip2* was monitored at different days after transfection of the reprogramming factors. This experiment detected a gradual increase in the expression of *Tnfaip2* in non-iPSCs on days 9 to 14 of reprogramming, whereas FACS-purified iPSCs that emerged during the corresponding time points showed very low expression levels of *Tnfaip2* (Fig 1F). Together, these results revealed that the knockdown of *Tnfaip2* increases the reprogramming of mouse somatic cells into pluripotent stem cells. Based on the rise in mRNA expression, the inhibitory effects of *Tnfaip2* may occur late during the time course of reprogramming.

### Knockdown of *Smed-exoc3*—the para-ortholog of *Tnfaip2* in planarian—impairs organ homeostasis and regeneration in *Schmidtea mediterranea*

Although planarians lack the *Tnfaip2* gene (alias *mSec*), phylogenetic analysis indicated that they express a para-ortholog of *Tnfaip2* called *Exoc3* (alias *Sec6*), a component of the exocyst complex (Fig 2A). BLASTP search identified TNFAIP2 and EXOC3 as the mammalian homologs of SMED-EXOC3. Gene occurrence suggests a split of *Exoc3* and *Tnfaip2* by duplication in a vertebrate ancestor prior to the split of cartilaginous and bony fishes (Fig 2A). All analyzed

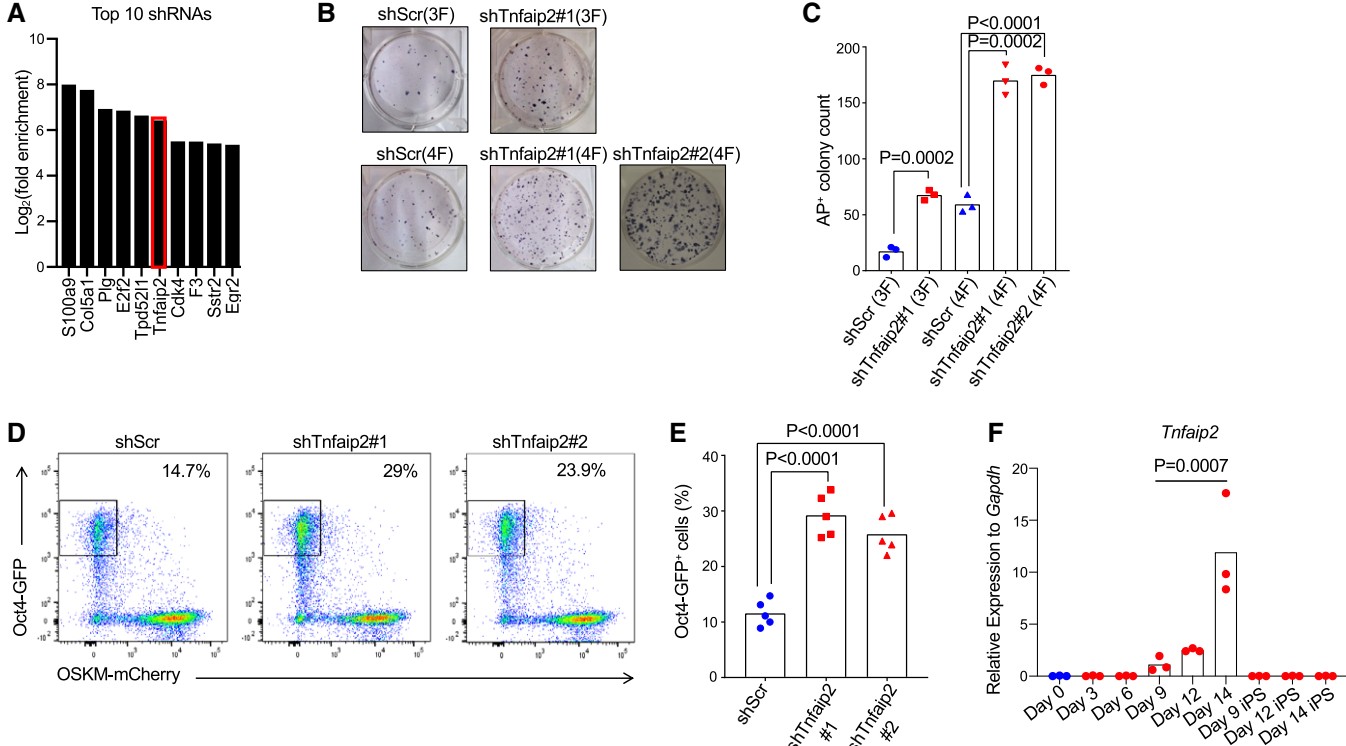

**Figure 1. Knockdown of *Tnfaip2* enhances reprogramming of MEFs into iPSCs.**

A  Stable RNAi screening of cancer-related genes was conducted to identify gene knockdowns that enhance reprogramming of MEFs into iPSCs. The graph shows the 10 shRNAs that were most strongly, positively selected in iPSCs compared to non-reprogrammed cells on day 14 of reprogramming (n = 3 biological replicates).

B–E  Mouse embryonic fibroblasts (MEFs) were co-infected with shRNAs against *Tnfaip2* or a scramble shRNA and reprogramming factors. The cotransduced cells were sorted and were reprogrammed for 14 days: (B) Representative images and (C) quantification of alkaline phosphatase-positive (AP$^+$) iPSC colonies on day 14 of reprogramming by transfection of 4 reprogramming factors (=4F: *Oct4, Sox2, Klf4, c-Myc* = OSKM) or three factors (=3F: OSK). [n = 3 biological replicates per group; data were normally distributed (P > 0.05 as per Shapiro–Wilk test) and analyzed by one-sided t-test analyzed by one-sided t-test with Holm–Sidak correction for multiple testing]. (D) Representative FACS profiles and (E) percentage of *Oct4*-GFP$^+$ iPSCs obtained after 14 days of reprogramming of MEFs from *Oct4*-eGFP mice using 4F-transduction on day zero that were infected with shRNAs targeting *Tnfaip2* or a scrambled shRNA control [n = 5 biological replicates; data were normally distributed (P > 0.05 as per Shapiro–Wilk test) and analyzed by one-sided t-test with Holm–Sidak correction for multiple testing].

F  mRNA of *Tnfaip2* was measured by RT–qPCR at the indicated days of reprogramming by 4F-transduction on day zero [n = 3 biological replicates; log-transformed data were not normally distributed (P > 0.05 as per Shapiro–Wilk test), P-value for the upregulation of *Tnfaip2* on consecutive test days 9, 12, and 14 was calculated starting with the probability of maximum rank-based difference, i.e. having two groups of three data points perfectly separating, $P_{1,b} = 0.1$ (bidirectional) and $P_{1,u} = 0.05$ (unidirectional); the probability of finding a triple series of max. difference starting at some time point is $P_{triple,tp} = P_{1,b} \cdot P_{1,u} \cdot P_{1,u}$, and finding such a triple somewhere across the 5-step time series is $P_{triple} = P_{triple,tp} + 2 \cdot (1 - P_{1,u}) \cdot P_{triple,tp} = 0.0007$.

vertebrates (in particular, both, cartilaginous and bony fishes) carry *Exoc3* and *Tnfaip2*. A higher rate of sequence diversification (as indicated by branch length) in the *Tnfaip2* gene suggests that this gene has undergone neo- or sub-functionalization (Fig 2A). Notably, also in mammals *Tnfaip2* shares considerable homology to *Exoc3* (Hase *et al*, 2009). *Tnfaip2* has been shown to interact with components of the exocyst complex, and it has an essential function for the formation of tunneling nanotubes (for review see Jia *et al*, 2018). The knockdown of *Exoc3* phenocopied the knockdown of *Tnfaip2* in increasing the reprogramming of MEFs into iPSCs suggesting that there was a functional overlap in these two genes in inhibiting reprogramming of somatic cells into iPSCs (Fig EV1B–E).

To further investigate the possible role of *Smed-exoc3* in an *in vivo* model, the planarian species, *Schmidtea mediterranea*, was employed. *S. mediterranea* maintains a life-long reservoir of somatic stem cells (commonly referred to as neoblasts) including a small subpopulation of cells with clonogenic capacity harboring

pluripotent potential to reconstitute the entire population of somatic stem cells in lethally irradiated planarians when transplanted as single cells (Wagner *et al*, 2011). Planarians depend on somatic stem cells to maintain tissue homeostasis and also to regenerate tissues in response to injury. They can regenerate all parts of their body following amputation. Together, these characteristics make planarians a good model to study the biology of somatic stem cells in tissue maintenance and regeneration *in vivo*.

To analyze whether *Smed-exoc3* can influence the function of somatic stem cells, planarians were subjected to starvation stress. Starvation leads to shrinkage of the worms while the body plan is maintained (Felix *et al*, 2019; Pellettieri, 2019). Seven days after the last feeding, planarians were subjected to a first round of injection with double-stranded (ds) RNA against *Smed-exoc3* (exoc3(RNAi)-treated). One single dsRNA is acceptable for experimentation since off-target effects have not been observed with this technique in planarians employing inhibitory double-stranded RNAs (dsRNA)

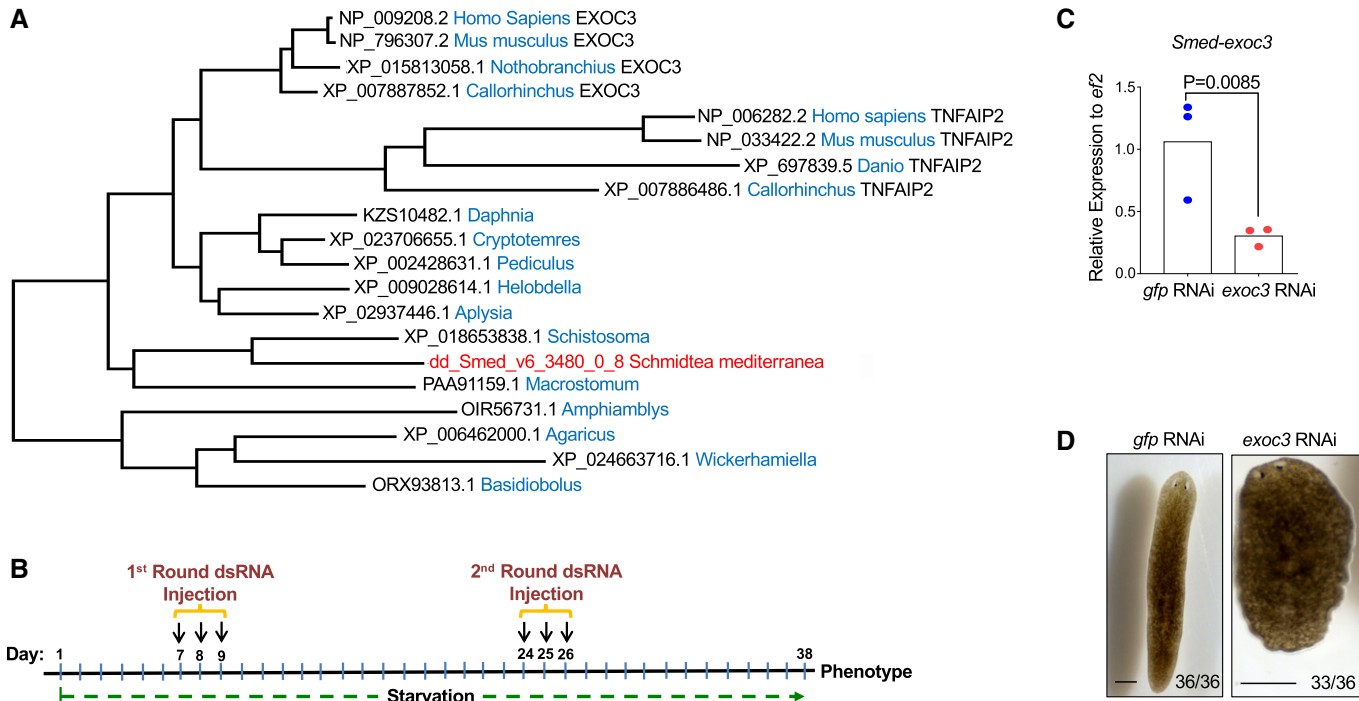

**Figure 2. Smed-exoc3 is the planarian para-ortholog of Tnfaip2.**

A  Maximum Likelihood phylogenetic tree analysis of mouse TNFAIP2 and planarian SMED-EXOC3. Note that TNFAIP2 emerged in vertebrates before the split of cartilaginous and bone fish since both lineages carry the TNFAIP2. *Callorhinchus* is a representative of cartilaginous fishes, while *Danio* and *Nothobranchius* are representatives of bony fishes. Note that branch lengths proportionally reflect the minimal number of substitutions along the branch. The longer branch length of TNFAIP2 versus EXOC3 in vertebrate species suggests that TNFAIP2 has undergone sub-functionalization. The species with both TNFAIP2 and EXOC3 are indicated, other species in the lower part only have EXOC3, which is not indicated.

B  Experimental scheme for injections of planarians (*S. mediterranea*) with dsRNA against *Smed-exoc3* or control injections (*gfp(RNAi)*).

C  RT–qPCR analysis of *Smed-exoc3* mRNA expression of total RNA (whole body-derived) from *exoc3(RNAi)*-treated planarians compared to control-injected planarians on day 38 of the injection scheme [$n = 3$ biological replicates; log-transformed data were normally distributed ($P > 0.05$ as per Shapiro–Wilk test); data were analyzed by one-sided *t*-test].

D  Representative photographs of *exoc3(RNAi)*-treated planarians versus control planarians at day 38 of the injection protocol as shown in panel B ($n = 4$ experimental replicates with 6–10 biological replicates per experiment, scale bar: 0.5 mm). Number ratios in the photographs depict the total number of animals showing the indicated phenotype.

that are 0.5–2 kb long; the *exoc3* dsRNA in our experiments was 650bp (see Table EV1 for the sequence of oligonucleotides employed for dsRNA amplification). After another 2 weeks of starvation, planarians underwent a 2nd round of *exoc3(RNAi)* treatment and were then left for another 12 days on starvation (day 38 of the RNAi injection protocol, Fig 2B). RNA-seq analysis of freshly isolated X1 cells (constituting proliferative somatic stem cells, Hayashi *et al*, 2006, Dataset EV2, look for "Exocyst complex component 3" in column K) and RT–qPCR analysis of total (whole body derived) RNA from *exoc3(RNAi)*-treated planarians (Fig 2C) revealed a reduction in mRNA levels by 50–70% in comparison with controls. Phenotypic analysis of the animals on day 38 of the RNAi injection protocol (Fig 2B) revealed hallmark features of a loss of tissue homeostasis such as head regression and overall shrinkage of *exoc3(RNAi)*-treated planarians compared to *gfp(RNAi)*-treated controls (Fig 2D).

The phosphorylation at the 10th serine residue of the Histone 3 (H3S10p) is a reliable indicator for the number of cells in G2/M phases of the cell cycle. This marker is generally used to analyze changes in the neoblast population—the only proliferating cell

compartment in planarians (Hendzel *et al*, 1997; Newmark & Sánchez Alvarado, 2000). Immunostaining against anti-H3S10p showed an increase in the mitotic population in *exoc3(RNAi)*-treated planarians compared to controls already on day 34 of the RNAi injection protocol (Figs 2B and 3A and B). Thus, this phenotype occurred already before overt defects in tissue homeostasis (such as head regression) became apparent, suggesting that the accumulation of H3S10p-positive cells precedes the loss of tissue homeostasis in *exoc3(RNAi)*-treated planarians. FACS analysis confirmed an increase in the percentage of X1-neoblasts in *Smed-exoc3*-depleted planarians compared to controls (Fig 3C and D). The data showed that the knockdown of *Smed-exoc3* leads to an increase in the relative number of X1 cells suggesting that *Smed-exoc3*-depleted neoblasts could not proceed in generating of differentiated cells. In line with this interpretation, RT–qPCR analysis verified reduced expression of marker genes of differentiating progenitor cells from the epidermal lineage, *prog-1* (*Smed-NB.21.11e*), *prog-2* (*Smed-NB.32.1g*), and *Smed-odc-1* (Eisenhoffer *et al*, 2008), in total (whole body derived) RNA from *exoc3(RNAi)*-treated planarians compared to controls (Fig 3E and F).

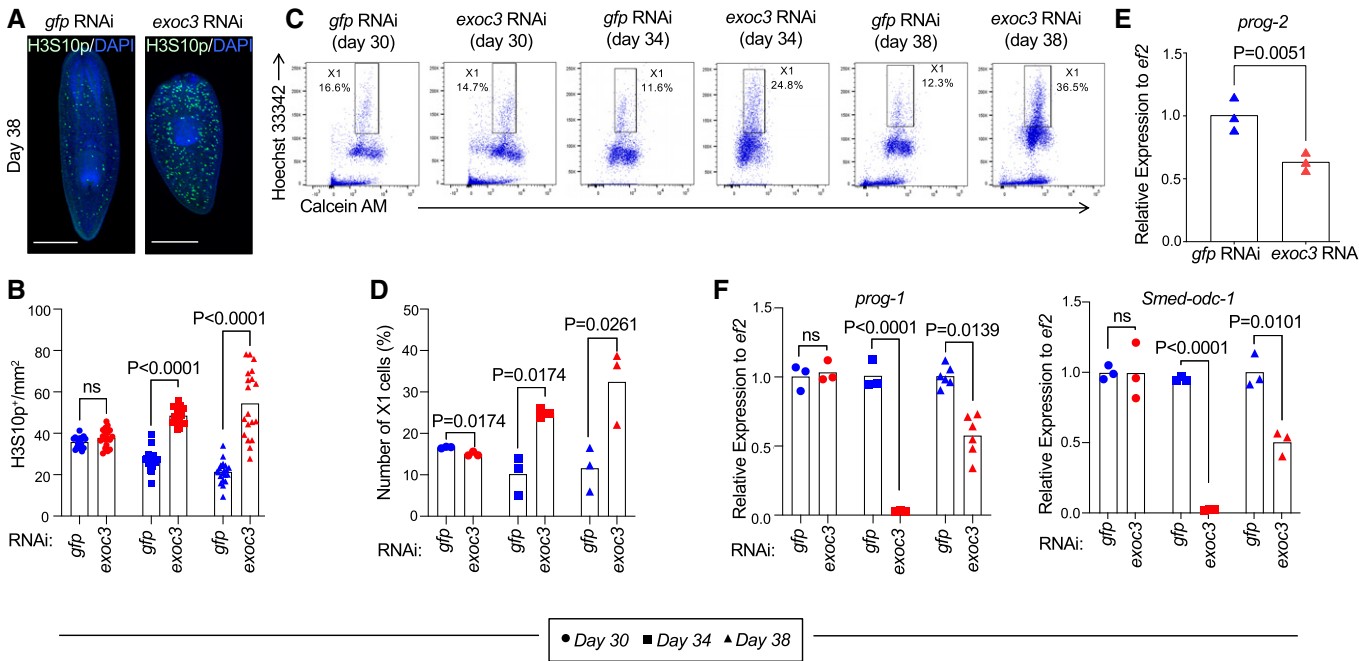

**Figure 3. *Smed-exoc3* repression increases the relative number of stem cells in planarians and reduces the expression of progenitor cell markers.**

A–F    Planarians were injected with *Smed-exoc3* dsRNA = *exoc3(RNAi)* or with control *gfp(RNAi)*. (A) The representative photographs show staining for phosphorylation of 10[th] serine residue of Histone 3 (H3S10p, a stem cell marker in planarians) at days 38 of the injection protocol ($n = 3$ experimental replicates with 8–10 biological replicates per experiment, scale bar: 0.5 mm). (B–F) *exoc3(RNAi)*-treated planarians versus *gfp(RNAi)*-treated controls were analyzed at the indicated time points of the dsRNA injection protocol: (B) Quantification of H3S10p$^+$ cells per mm$^2$ in planarians at day 30, 34, and 38 of the injection protocol [$n = 3$ experimental replicates with 5–7 biological replicates per experiment; data were normally distributed ($P > 0.05$ as per Shapiro–Wilk test) and analyzed by multiple *t*-tests with Holm–Sidak correction for multiple comparisons]. (C,D) FACS-based time course analysis on changes in the fraction of somatic stem cells (X1-population) at indicated time points. Cells were obtained from whole body-trypsinized planarians of *gfp(RNAi)*-injected controls and *exoc3(RNAi)*-treated planarians and stained with the cytoplasmic dye Calcein-AM and the nuclear (DNA) dye Hoechst 33342: (C) Representative FACS profiles and (D) quantification of the relative number of X1 cells [$n = 3$ biological replicates; data were normally distributed ($P > 0.05$ as per Shapiro–Wilk test) and analyzed by multiple *t*-tests with Holm–Sidak correction for multiple comparisons]. (E, F) RT–qPCR analysis for makers of differentiating progenitor cells from the epidermal lineage on total (whole body derived) RNA from *exoc3(RNAi)*-treated planarians (E) Expression analysis of differentiation marker *prog-2* on day 38 of the injection scheme [$n = 3$ biological replicates; log-transformed data were normally distributed ($P > 0.05$ as per Shapiro–Wilk test and analyzed by one-sided *t*-test], (F) Expression analysis of marker genes of differentiated cells (*prog-1* and *Smed-odc-1*) on day 30, 34, and 38 of the injection scheme [$n = 3$ biological replicates per group and time point except for *prog-1* expression at day 38, which includes six biological replicates; log-transformed data were normally distributed ($P > 0.05$ as per Shapiro–Wilk test) and analyzed by multiple *t*-tests with Holm–Sidak correction for multiple comparisons]. For *Smed-odc-1*, log-transformed mean centered group data were normally distributed and ($P > 0.05$ as per Shapiro–Wilk test) and analyzed by multiple *t*-tests with Holm–Sidak correction for multiple comparisons].

To analyze consequences of *Smed-exoc3* knockdown on organ maintenance in planarians, immunofluorescence (IF) staining was carried out against the following differentiation markers:

(i) 3C11 (alias anti-synapsin), a planarian pan-neural marker (Cebrià, 2008). This analysis displayed a reduction in neuronal layer thickness of bilobed cephalic ganglia (cg) and diminished sensory neurons spiking out of the ganglia in the central nervous system (CNS) in *Smed-exoc3* knockdown planarians compared to control animals (Fig 4A).

(ii) VC1—a marker of photosensitive cells in the eye (Sakai *et al*, 2000). This analysis revealed a distorted eye phenotype in the head region of *exoc3(RNAi)*-treated planarians compared to controls, characterized by a severe reduction and deformations of photosensitive as well as a strong atrophy of the optic chiasm (Fig 4B).

To further analyze these defects in organ homeostasis, a transcriptome analysis was conducted on freshly isolated X1 cells from *exoc3(RNAi)*-treated planarians compared to *gfp(RNAi)*-treated planarians on day 38 of the injection scheme (Fig 2B). Differentially expressed genes (DEGs) in X1 cells from *exoc3(RNAi)*-treated planarians versus X1 cells of *gfp(RNAi)*-treated controls (Dataset EV2) were compared with the transcriptome profile of previously published planarian somatic stem cell gene sets, including the "Neoblast dataset" from Solana *et al*, 2012, the "Neoblast 1 dataset" from Plass *et al*, 2018, and the "*smedwi-1* high dataset" from Fincher *et al*, 2018 (Fig 5A; Dataset EV3). We observed a significant enrichment of genes from the 3 published datasets with *Smed-exoc3*-regulated DEGs, indicating that *Smed-exoc3* gene status alters neoblast-associated transcriptome signatures in planarians. Binomial test analysis for each of these overlapping fractions of genes within the list of DEGs of *Smed-exoc3*-depleted planarians versus controls, revealed a significantly bigger proportion of downregulated genes than expected (ground probability = 0.5534) for 2 of the overlaps ("*smedwi-1* high dataset" from Fincher *et al*, 2018: $P = 0.0299$ and "Neoblast dataset" from Solana *et al*, 2012: $P < 0.0001$),

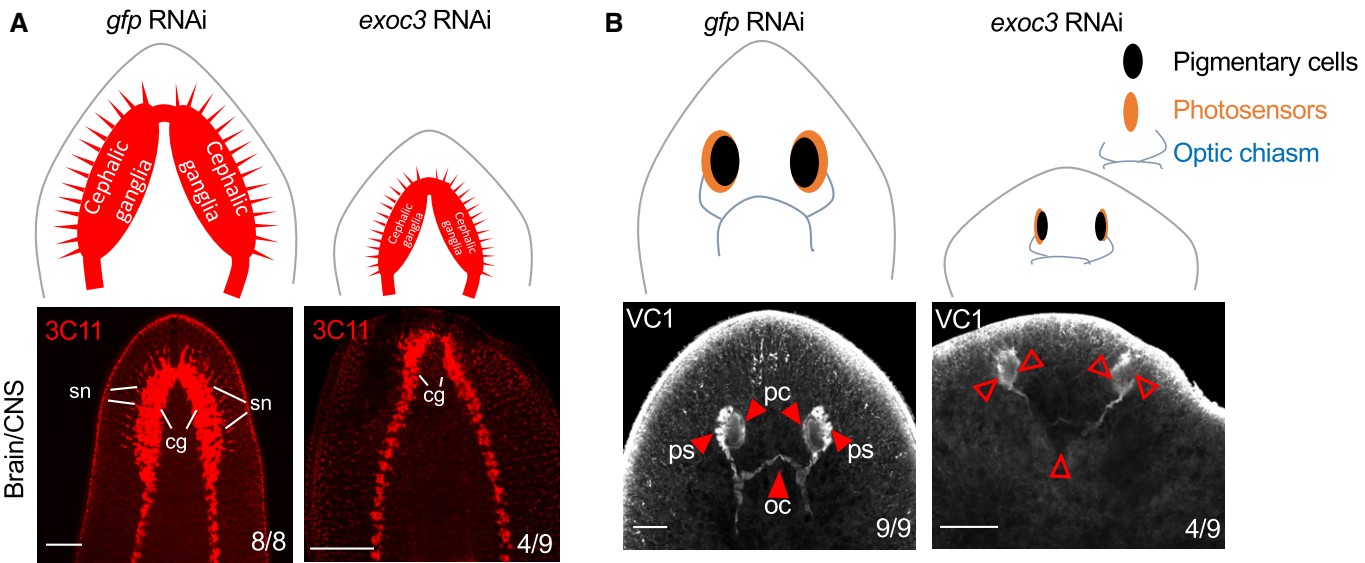

**Figure 4. Suppression of *Smed-exoc3* abrogates organ homeostasis in planarians.**

A, B Planarians were injected with *exoc3(RNAi)* or with control *gfp(RNAi)*. Organ phenotypes were analyzed on day 38 of the injection protocol. Organ homeostasis was analyzed by staining against specific markers: (A) The structure of the brain/central nervous system (CNS) was analyzed by staining against synapsin (3C11). The cartoon represents the phenotypic change (upper panel). Representative photographs are shown in the lower panel (3 experimental replicates with 2–3 biological replicates per experiment, scale bar: 0.5 mm). Number ratios in the photographs indicate the total number of animals exhibiting the represented phenotype. There was an increase in the occurrence of atrophic cephalic ganglia and sensory neuron loss in *exoc3(RNAi)*-treated planarians versus controls (cg: cephalic ganglia; sn: sensory neurons). (B) Planarian visual system was analyzed by staining against VC1. Schematic cartoon (upper panel) and representative images (lower panel) are shown (3 experimental replicates with three biological replicates per experiment, scale bar: 0.5 mm). Number ratios in the photographs indicate the total number of animals exhibiting the represented phenotype. Arrows point to photosensitive cells (ps); pigmentary cells (pc) and to the optic chiasm (oc).

but not for the 3rd overlap ("Neoblast 1 dataset" from Plass *et al*, 2018: *P* = 0.4628, Fig 5B–D). A literature survey revealed that a large number of neoblast-related genes that were dysregulated by *Smed-exoc3* depletion have a known role in the regulation of stemness and differentiation (Fig 5B–D, Dataset EV3). Together, these data indicated that *Smed-exoc3* depletion results in disturbed gene expression of neoblast genes that control stemness and differentiation. While *Smed-exoc3* depletion resulted in downregulation but also in upregulation of these genes, the data suggested that *Smed-exoc3* depletion may arrest neoblast cells at certain transitional states between stemness and differentiation.

To determine whether *Smed-exoc3* downregulation would affect the capacity of neoblast cells to generate differentiated progenitor cells, DEGs of neoblasts from *exoc3(RNAi)*-treated planarians versus *gfp(RNAi)*-treated controls (Dataset EV2) were compared to the expression of genes characterizing clusters of progenitor cells that exhibit expression of the neoblast marker *smedwi-1* (referred to as *smedwi-1+*) but a distinct gene expression signature from the neoblast population indicating that these cells represent a population of progenitor cells differentiating into different lineages (Fincher *et al*, 2018). This comparison revealed a significant overlap of differentiation-associated genes in 19 out of 22 *smedwi-1+* progenitor cell clusters with gene expression changes in X1-neoblasts in response to *Smed-exoc3(RNAi)* treatment (Fig EV2; Dataset EV4). These results supported the conclusion that *exoc3(RNAi)* may interfere with the capacity of neoblast cells to generate differentiated progenitor cells. We also compared DEGs of neoblasts from *exoc3*

*(RNAi)*-treated planarians versus controls with differentiation markers that are highly expressed in post-mitotic cells ("X2 epithelial progenitor dataset" in Zhu *et al*, 2015, Dataset EV5) but also expressed in neoblasts (Eisenhoffer *et al*, 2008; Pearson & Sánchez Alvarado, 2010; Zhu *et al*, 2015). There was a significant overlap between these differentiation-associated genes and gene expression changes in X1-neoblasts of *Smed-exoc3*-depleted planarians versus controls (Fig 5E and F). Remarkably, all of these overlapping genes were significantly downregulated in the X1-neoblasts of *Smed-exoc3*-depleted planarians versus the X1 fraction of control planarians and represent a bigger proportion of downregulated genes than expected (Fig 5E; Dataset EV5, Binomial test, ground probability = 0.5534, *P* < 0.0001). Together, this transcriptome analysis supports the conclusion that *exoc3(RNAi)* disturbs the expression of neoblast-associated genes that control stemness and differentiation as well as the capacity of neoblasts to generate differentiated cells.

To test whether *Smed-exoc3* knockdown would also impair regeneration, immunofluorescence (IF) staining was carried out on amputated planarians. *Smed-exoc3* was depleted by double round of dsRNA injection with 7-day inter-treatment intervals followed by amputation of head and tails of the planarians. Regeneration was assessed 12-day post-amputation (Fig 6A) in the trunks. *Smed-exoc3*-depleted trunks exhibited a strong regenerative defect characterized by a failure of blastema formation and shrinkage of the animals (Fig 6B). IF staining revealed a strongly reduced expression of the differentiation markers VC1 (Fig 6C) and 3C11 (Fig 6D). These results stood in line with the data on impairments in organ

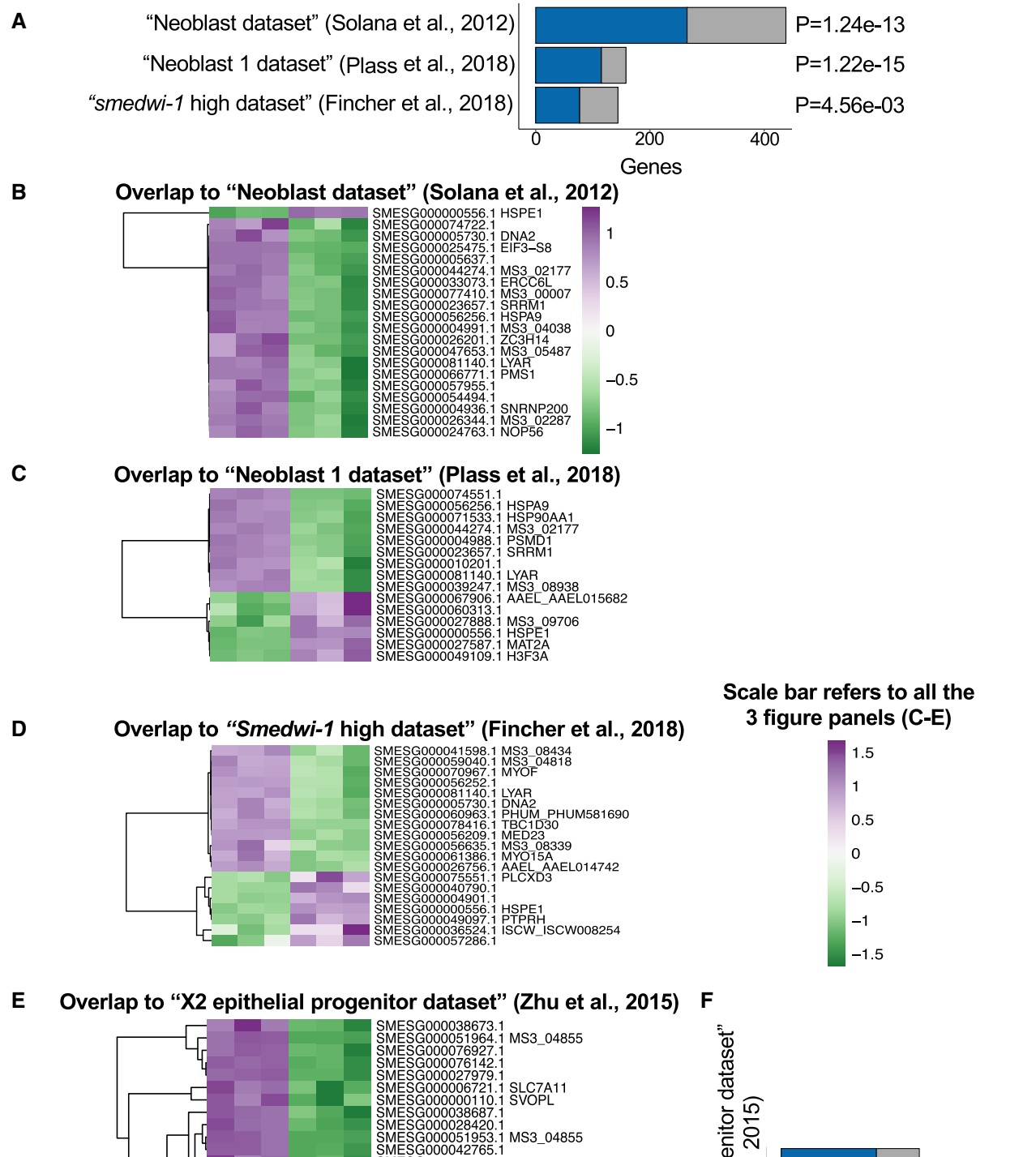

**Figure 5.**

**Figure 5.  Depletion of *Smed-exoc3* influences expression of genes related to stem cell maintenance and differentiation.**

A–F   RNA-seq was conducted on freshly isolated X1 cells from *exoc3(RNAi)*-treated compared to X1 cells from *gfp(RNAi)*-treated planarians on day 38 of the injection protocol. The transcriptome analysis results were compared with previously published gene expression profiles: (A, F) The bar-graphs depict the overlap (blue) of DEGs in X1 cells of *Smed-exoc3*-depleted planarians versus controls (*n* = 3 biological replicates; adjusted *P* < 0.05) with genes expressed in the indicated sample sets of (A) neoblast stem cells or (F) differentiating progenitor cells of epidermal lineage from the indicated publications while the sum of overlapping (blue) and set-exclusive (gray) genes represents the total number of genes of a cluster that were analyzed in this study. *P*-values right of the bars represent significance of overlap (hypergeometric test). (B–E) The heat maps depict the intersection of DEGs with strong expression signals (log$_2$ fold change >0.5 or < −0.5) discovered in X1 cells of *exoc3*-depleted planarians versus controls (*n* = 3 biological replicates) with the indicated, previously published datasets on gene expression profiles in (B–D) neoblast cells or (E) differentiating progenitor cells of epidermal lineage. The color scale represents the gene-wise z-score calculated from normalized gene expression levels while purple and green indicate upregulated and downregulated genes in X1 cells of *exoc3*-depleted planarians versus controls, respectively.

homeostasis (Fig 4A and B) but showed an even more pronounced phenotype supporting the hypothesis that *Smed-exoc3* is required for the function of neoblast cells in maintaining tissue homeostasis and regeneration in planarians.

## *Tnfaip2* is required for differentiation of pluripotent ES cells in embryoid body cultures

To test whether *Tnfaip2* would also lead to an impairment in the capacity of ESCs to generate differentiated cells, a CRISPR-Cas9 approach was employed to generate *Tnfaip2* knockout (KO) ESCs. One of the selected ESC clones harboring a premature stop codon in 2$^{nd}$ exon (Fig EV3A) was used for further studies in comparison with the parental ES cell line (E-14; Neri *et al*, 2013). To determine the possible influence of the *Tnfaip2* genotype on ES cell differentiation, hanging drop cultures of ESC-derived embryoid bodies (EBs) were analyzed (Behringer *et al*, 2016). Both WT and *Tnfaip2*$^{-/-}$ EBs developed dense spheres, which are characteristics of undifferentiated embryoid bodies (Fig 7A). Transferring EBs (3 days after hanging drop culture initiation) into differentiation-inducing medium for 3 days promotes differentiation into ectoderm, mesoderm, and endoderm lineages (Takahashi & Yamanaka, 2006; Behringer *et al*, 2016). Under these conditions, WT EBs started to display a rosette structure around the dense sphere—a characteristic sign of differentiating EBs in culture (Fig 7A). In contrast, *Tnfaip2*$^{-/-}$ EBs failed to differentiate and did not develop the rosette structure of differentiated cells (Fig 7A).

DEGs detected in the transcriptome analysis of *Tnfaip2*$^{-/-}$ EBs on day 3 of differentiation versus WT EBs were compared to gene sets of the AmiGO 2 data base (Carbon *et al*, 2009) related to the term "stem cell maintenance: positive and negative regulators" as well as to the differentiation-associated terms "mesoderm development" and "ectoderm development". Five additional genes were added to the "stem cell maintenance" gene sets based on our own literature searches. Binomial analysis revealed that the proportion of upregulated genes in the category of "positive regulators of stem cell maintenance" in *Tnfaip2*$^{-/-}$ EBs versus WT EBs was significantly bigger than expected (Fig 7B—upper part; Dataset EV6; *P* = 0.0297, ground probability = 0.4965) including hallmark pluripotency genes, such as *Lif, Klf2*, and *Nanog*. Moreover, the proportion of downregulated genes in the category "mesoderm development" in *Tnfaip*$^{-/-}$ EBs compared to WT EBs was significantly larger than expected (Fig 7C; Dataset EV6; binomial test, *P* = 0.0004, ground probability = 0.5035). The observed overlaps to negative regulators of stem cell maintenance (Fig 7B—lower part) or markers of ectoderm development (Fig 7D) were not significant but the number of genes in these sets was rather low for statistical testing.

Immunofluorescence staining against selected ectodermal markers (SOX1) and mesodermal markers (Brachyury = T) confirmed elevated expression of these two targets in WT EBs versus *Tnfaip2*$^{-/-}$ EBs after 3 days of *in vitro* differentiation (Fig 7E).

To further investigate the role of *Tnfaip2* during generation of differentiated cells, the expression profile of *Tnfaip2* mRNA was analyzed by RT–qPCR at different time points after differentiation induction of WT EBs. This experiment revealed an increase of *Tnfaip2* expression at early time points after differentiation induction of EBs compared to ESCs (Fig 7F). Together, these results support the conclusion that the induction of *Tnfaip2* contributes to differentiation of ESCs.

## Deletion of *Tnfaip2* suppresses induction of vimentin, lipid droplet (LD) formation, and triacylglycerol (TAG) accumulation in ES cells in response to differentiation induction

To identify *Tnfaip2*-dependent effector mechanisms that contribute to impairments in differentiation of ES cells, a proteomic time course analysis was conducted on *Tnfaip2*$^{-/-}$ versus WT ES cells at different time points after induction of differentiation in embryoid body (EB) cultures (see Fig 7A). EBs were analyzed 3 days after initiation of hanging drop cultures (referred to as day 0). At this point, medium was changed to differentiation-inducing medium and EBs were analyzed on days 0, 1, 2, and 3 after the medium change. This analysis identified a total number of around 4,900 proteins on each indicated days of differentiation (Dataset EV7). A principal component analysis of differentially expressed proteins of *Tnfaip2*$^{-/-}$ EBs versus WT EBs was conducted in comparison with the proteome of *Tnfaip2*$^{-/-}$ ESCs versus WT ESCs. *Tnfaip2*$^{-/-}$ ESCs and WT ESCs clustered relatively close together. In contrast, differentiation-induced *Tnfaip2*$^{-/-}$ EBs diverged from WT ESCs, most strongly on day 3 of differentiation (Fig 8A).

The intermediate filament protein vimentin (VIM) was identified as the only detectable protein among the top 100 differentially expressed proteins (cut-off criteria: average log$_2$ratio> ±1.2 and -log$_{10}$ Qvalue < 2) that was downregulated at all the four time points (days 0, 1, 2, and 3) of differentiation induction of *Tnfaip2*$^{-/-}$ EBs versus WT EBs (Fig 8B; Dataset EV8). RT–qPCR analysis revealed that *Vim* expression was also downregulated on mRNA level at day 0 and day 1 of differentiation induction of *Tnfaip2*$^{-/-}$ EBs versus WT EBs (Fig 8C). TNFα signaling has previously been reported to activate VIM expression (Mor-Vaknin *et al*, 2003), and VIM is known to instruct cell differentiation in the hematopoietic system (Benes *et al*, 2006) as well as cell fate changes during EMT (Mendez *et al*, 2010).

VIM is an architectural protein that is known to be required for the formation of LDs that store fatty acids (FAs) in form of neutral

**A**

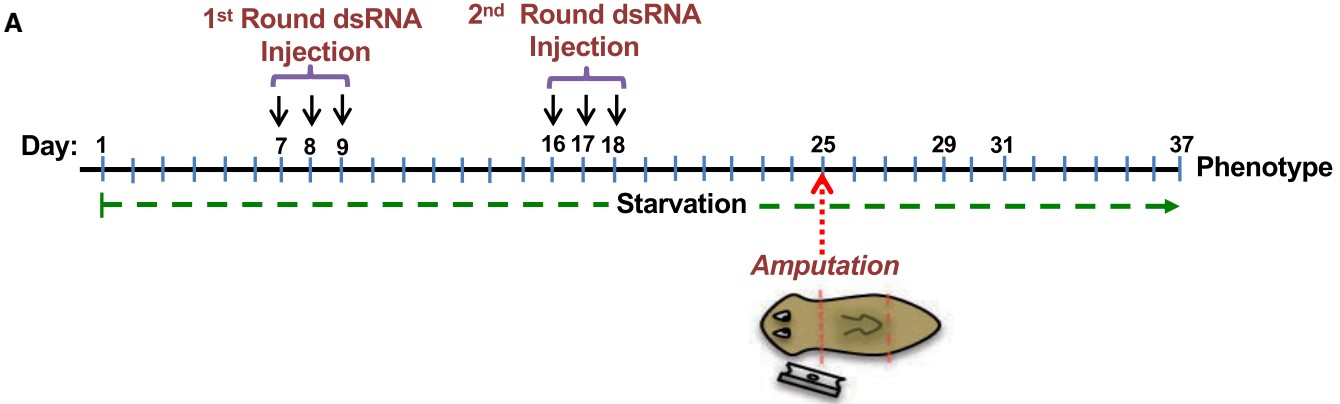

**B**

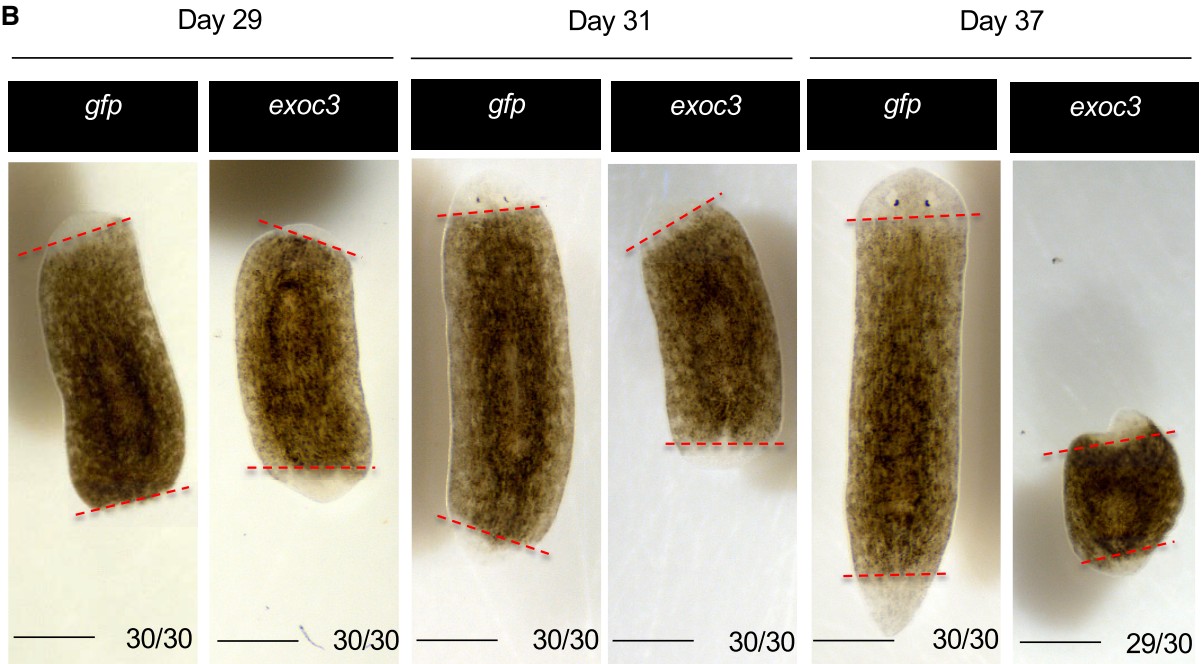

**C**

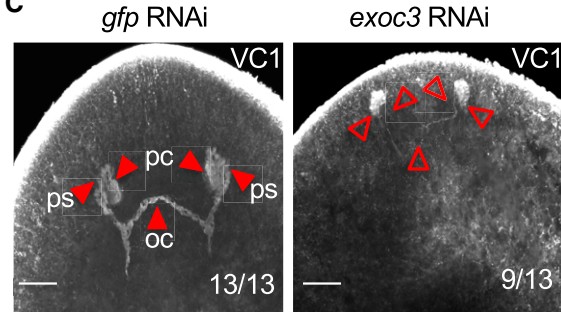

**D**

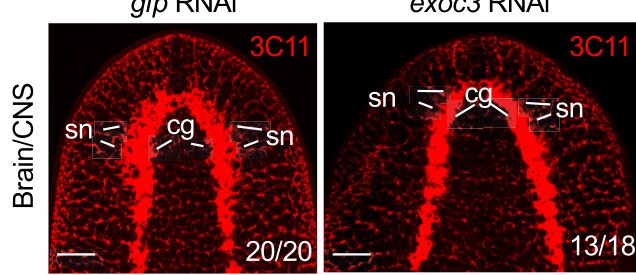

**Figure 6.**

Figure 6.  Downregulation of *Smed-exoc3* disrupts regenerative potential of planarians.

A    Experimental scheme to analyze the effect of *Smed-exoc3* depletion on regeneration of head and tail amputated planarians.
B–D  Planarians (*S. mediterranea*) were injected with *exoc3(RNAi)* or with control *gfp(RNAi)*. (B) Representative photographs of morphology of *exoc3* knockdown planarian versus control planarian on days 29, 31, and −37 of the protocol shown in (A). (3 experimental replicates with 10 biological replicates per experiment; red dotted lines demarcate sites of amputation, scale bar: 0.5mm). Number ratios in the photographs indicate the photographically represented phenotypes. (C,D) Organ regeneration was analyzed by immunostaining of different organ compartments in regenerating *exoc3(RNAi)-treated* planarians versus *gfp(RNAi)*-treated control on day 37 of the protocol shown in (A): (C) Eyes were analyzed by staining against VC1. Representative images are shown (3 experimental replicates with 4–5 biological replicates per experiment). Arrows point to photosensitive cells (ps); pigmentary cells (pc) and to the optic chiasm (oc). Number ratios in the photographs refer to planarians with regenerative defects for the indicated phenotype (scale bar: 0.5 mm). (D) Representative photographs of staining with planarian pan-neural marker 3C11 (alias Synapsin) marking cephalic ganglia and sensory neurons (3 experimental replicates with 6–7 biological replicates per experiment). Number ratios in the photographs refers to planarians exhibiting the photographically represented phenotype, characterized by underdeveloped bilobed cephalic ganglia and atrophy of sensory neurons in *exoc3(RNAi)*-treated planarians versus controls (scale bar: 0.5 mm, cg: cephalic ganglia; sn: sensory neurons).

lipids, which in turn increases the formation and size of LDs (Franke *et al*, 1987; Evans, 1994; Lieber & Evans, 1996). Interestingly, FA metabolism has been shown to influence differentiation of mouse ESCs (Yanes *et al*, 2010). To analyze whether the regulation of LD formation could be involved in differentiation defects of $Tnfaip2^{-/-}$ EBs, LD formation was analyzed by BODIPY 493/503, a widely used dye to stain and visualize neutral lipids including triacylglycerols (TAGs). FACS analysis on days 0–3 after differentiation induction revealed that WT EBs were highly positive for BODIPY 493/503 staining, whereas $Tnfaip2^{-/-}$ EBs exhibited significantly reduced signals (Fig 8D and E). Transmission electron microscopy (TEM) validated this result showing that WT EBs exhibited significantly higher numbers of LDs compared to $Tnfaip2^{-/-}$ EBs on day 1 after differentiation induction (Fig 8F and G).

LDs are important compartments for lipid metabolism. LDs grow by fusion, the incorporation of neutral lipids (i.e., TAGs), and local TAG biosynthesis (Walther & Farese, 2012; Wilfling *et al*, 2013; Olzmann & Carvalho, 2019). LD-localized TAGs are a reservoir of FAs, which promote LD biogenesis and growth (Olzmann & Carvalho, 2019). In addition, FAs have been implicated to induce differentiation in various stem cell systems, such as intestinal and hematopoietic stem cells (Ito *et al*, 2012; Bailey *et al*, 2015). To analyze whether deficiencies in the dynamic regulation of LDs in differentiation-induced $Tnfaip2^{-/-}$ EBs versus WT EBs are associated with changes in the lipid profile, we analyzed the abundance and composition of major TAG species by ultraperformance liquid chromatography ESI tandem mass spectrometry. This analysis revealed that cellular TAG content strongly increases in WT EBs during the first 3 days after differentiation induction. This response was significantly impaired in $Tnfaip2^{-/-}$ EBs on day 1 and day 3 after differentiation induction (Fig 8H).

## Application of palmitic acid and palmitoyl-L-carnitine rescues differentiation defects of *Tnfaip2*-deficient ESCs

Together, the above data suggested that loss of *Tnfaip2*-mediated *Vim* expression associates with defects in LD formation and decreases in cellular TAG accumulation as well as with impaired differentiation capacity of EB cultures. Of note, it was shown that FA depletion in culture medium is sufficient to keep human ESCs in a transition stage from naïve to primed ES cells (Cornacchia *et al*, 2019) suggesting that failure in LD formation and reductions in cellular TAG levels could indeed functionally contribute to impairments in *Tnfaip2*-deficient EBs and possibly also to defects in the capacity of neoblast cells in planarians to generate differentiated cells thus leading to defects in organ maintenance and regeneration.

To test this hypothesis, a rescue experiment with palmitic acid (PA) treatment was employed. PA was chosen as it is one of the most abundant FA in animals. In addition, our proteome analysis had shown that levels of carnitine palmitoyltransferase (CPT1A), an essential enzyme that converts fatty acyl-CoA into fatty acyl-carnitine (palmitic acid into palmitoyl carnitine) for transportation into mitochondria (Leji *et al*, 2000), were reduced in $Tnfaip2^{-/-}$ ES cells versus WT ESCs and also in $Tnfaip2^{-/-}$ EBs versus WT EBs during *in vitro* differentiation, specifically on days 0, 1, and 2 (Fig 9A). Additionally, a significant decrease in *Cpt1a* mRNA expression was observed on day 0 and day 1 of differentiation induction of $Tnfaip2^{-/-}$ EBs as compared to WT EBs (Fig 9B).

It has been reported that PA and its mobilized form, palmitoyl-L-carnitine (PC, which can penetrate through mitochondrial membranes), are essential drivers of ectoderm and mesoderm specification in mouse ESCs (Yanes *et al*, 2010). Of note, autophagy-mediated lipid degradation and FA oxidation were shown to be essential for normal neutrophil differentiation (Riffelmacher *et al*, 2017). Interestingly, PA/PC treatments partially restored the differentiation potential of $Tnfaip2^{-/-}$ EBs but had no significant effect on differentiation of WT EBs (Fig 9C and D). To check whether treatment with PA/PC restores differentiation-associated gene expression profiles in $Tnfaip2^{-/-}$ EBs, we conducted transcriptome analysis of $Tnfaip2^{-/-}$ EBs that have undergone *in vitro* differentiation for 3 days either in the absence or presence of PA/PC (Dataset EV9). Of note, PA/PC treatment of $Tnfaip2^{-/-}$ EBs reverted the prolonged expression of positive regulators of stem cell maintenance (Fig 9E—upper part, binomial test, $P = 0.0056$, ground probability = 0.5366, asterisks mark reverted changes in gene expression that were seen in $Tnfaip2^{-/-}$ EBs versus WT EBs depicted in Fig 7B—upper part). PA/PC treatment of $Tnfaip2^{-/-}$ EBs also rescued the expression levels of mesoderm development markers (Fig 9F; binomial test, $P < 0.0001$, ground probability = 0.4634, asterisks mark the rescue in the attenuated gene expression seen in $Tnfaip2^{-/-}$ EBs versus WT EBs depicted in Fig 7C). The observed rescue in expression levels of negative regulators of stem cell maintenance (Fig 9E—lower part) or markers of ectoderm development (Fig 9G) were not significant but the number of genes in these sets was rather low for statistical testing. Immunofluorescence analysis confirmed an upregulation of early ectoderm and mesoderm markers, SOX1 and T in PA/PC-treated $Tnfaip2^{-/-}$ EBs versus control-treated $Tnfaip2^{-/-}$ EBs (Fig 9H, compare to Fig 7E). Together, these data provided experimental evidence that decreases in lipid metabolism contribute to defects in

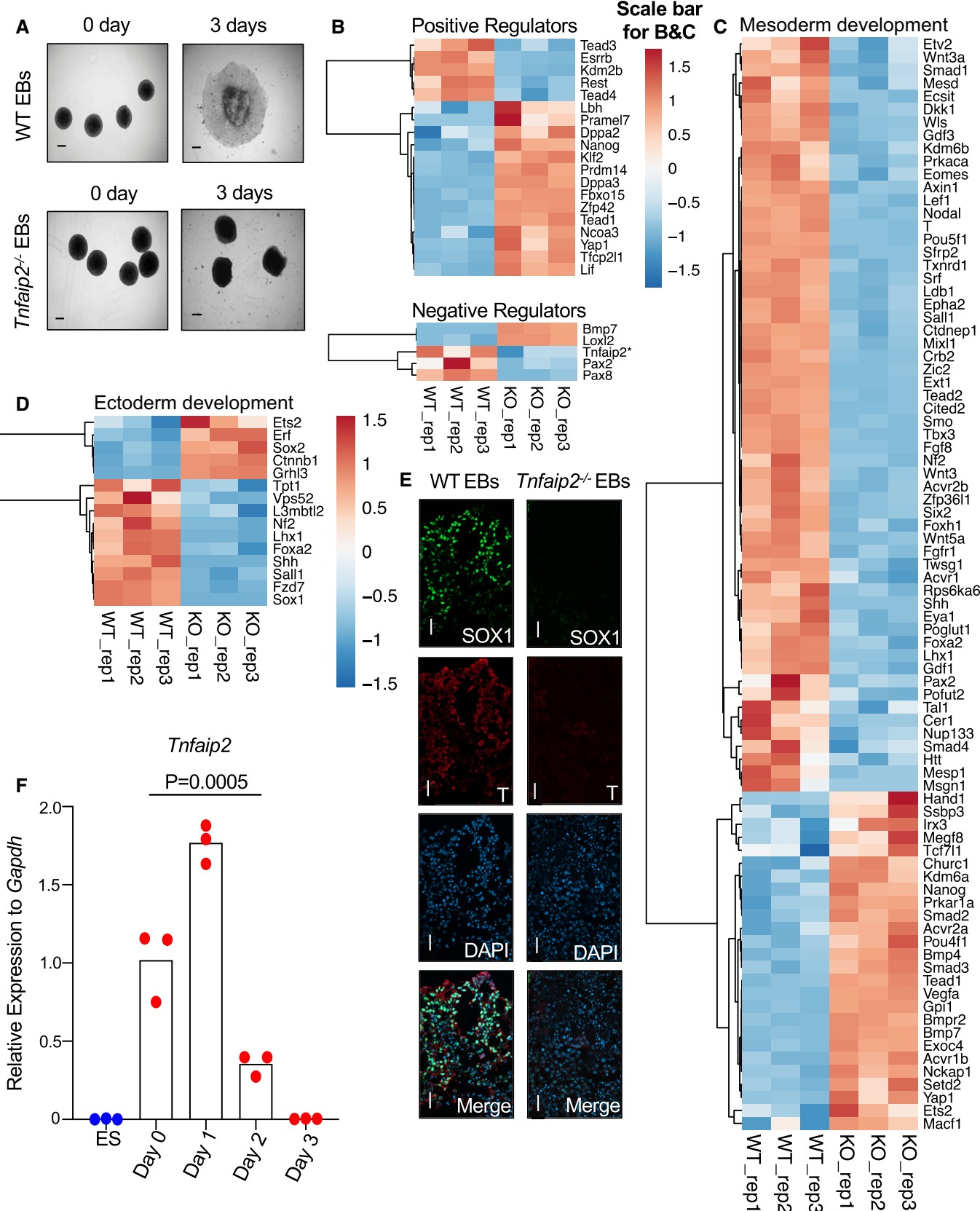

**Figure 7.**

**Figure 7. *Tnfaip2* deletion abrogates differentiation potential of ESCs.**

A–F One thousand ESCs of the indicated genotypes were placed into hanging drop cultures to form embryoid bodies (EBs). After 3 days (referred to as day 0), differentiation was induced by exposure to differentiation medium (days 1 to days 3 after medium change). (A) Representative photographs of EBs of the indicated genotypes at day 0 and day 3 after differentiation induction (experiments were conducted in three biological replicates; *n* = 5–42 EBs per biological replicate, scale bar: 0.02 mm). (B–D) RNA-seq analysis was conducted on *Tnfaip2⁻/⁻* EBs versus WT EBs on day 3 of differentiation induction of EBs (*n* = 3 biological replicates). The heat maps depict the intersection of DEGs with the following gene sets retrieved from the AmiGO 2 data base (Carbon *et al*, 2009): (B) "stem cell maintenance: positive and negative regulators", (C) "mesoderm development" and (D) "ectoderm development" from the AmiGO 2 data base (Carbon *et al*, 2009). *Tnfaip2* and 5 other stem cell-related genes (*Tfcp2l1, Dppa3, Fbxo15, Zfp42,* and *Klf2*) were additionally incorporated based on literature searches. The color scale represents the gene-wise z-score calculated from normalized gene expression levels. The asterisk refers to the gene identified in this study. (E) Representative image of immunofluorescence staining against the ectodermal marker (SOX1) and the mesodermal marker (T alias Brachyury) in WT and *Tnfaip2⁻/⁻* EBs on day 3 of differentiation induction (3 repeat experiments were conducted on a total number of 9–10 EBs per genotype, scale bar: 20µm). (F) mRNA expression of *Tnfaip2* measured by RT–qPCR in ESCs on the indicated days of differentiation induction of WT EBs [*n* = 3 biological replicates; log-transformed data were normally distributed (*P* > 0.05 as per Shapiro–Wilk test), the *P*-value for the upregulation of *Tnfaip2* on consecutive time points (days 0, 2, and 3) was calculated starting with the probability of maximum rank-based difference, i.e. having two groups of three data points perfectly separating, $P_{1,b}$ = 0.1 (bidirectional) and $P_{1,u}$ = 0.05 (unidirectional); the probability of finding a triple series of max. difference starting at some time point is $P_{triple,tp} = P_{1,b} \cdot P_{1,u} \cdot P_{triple,tp}$, and finding such a triple somewhere across the 4-step time series is $P_{triple} = P_{triple,tp} + (1 - P_{1,u}) \cdot P_{triple,tp} = 0.0005$.

ESC differentiation in *Tnfaip2⁻/⁻* EBs versus WT EBs, which can be rescued by PA/PC treatment.

### *Tnfaip2* and *Vim* act epistatically to impair reprogramming of MEFs into iPSCs

To further analyze the pathway of *Tnfaip2 and Vim* in impairing the reprogramming of somatic cells into iPSCs, we first conducted experiments to analyze whether knockdown of *Vim* would contribute to the inhibition of de-differentiation of somatic cells during reprogramming. To this end, MEFs from *Oct4*-eGFP reporter mice (Lengner *et al*, 2007, see above) were lentivirally infected with two verified shRNAs inducing *Vim* knockdown (Fig EV3B) or with a scrambled control shRNA. Induction of reprogramming was conducted with a 4-factor polycistronic lentivirus construct (OSKM,

see above). Of note, *Vim* knockdown led to a similar increase in reprogramming efficiency (Fig 10A–C) as the knockdown of *Tnfaip2* (Fig 1B–E). Moreover, there was only a small increase in the reprogramming efficiency upon codepletion of *Tnfaip2* and *Vim* compared to the single knockdown of either *Tnfaip2* or *Vim* (Fig 10D and E). Albeit this increase was significant for the latter comparison, the low size of the codepletion effect relative to the single knockdown effects of either *Tnfaip2* or *Vim* suggested that *Tnfaip2* and *Vim* act epistatically in inhibiting reprogramming. RT–qPCR analysis of shRNA-infected MEFs revealed that *Tnfaip2* knockdown also led to a reduction in *Vim* mRNA expression, whereas *Tnfaip2* mRNA expression levels did not change in response to *Vim* knockdown (Fig 10F and G). These data suggested that *Tnfaip2* acts upstream in regulating *Vim* expression but not the other way around. Next, we combined *Vim* overexpression and

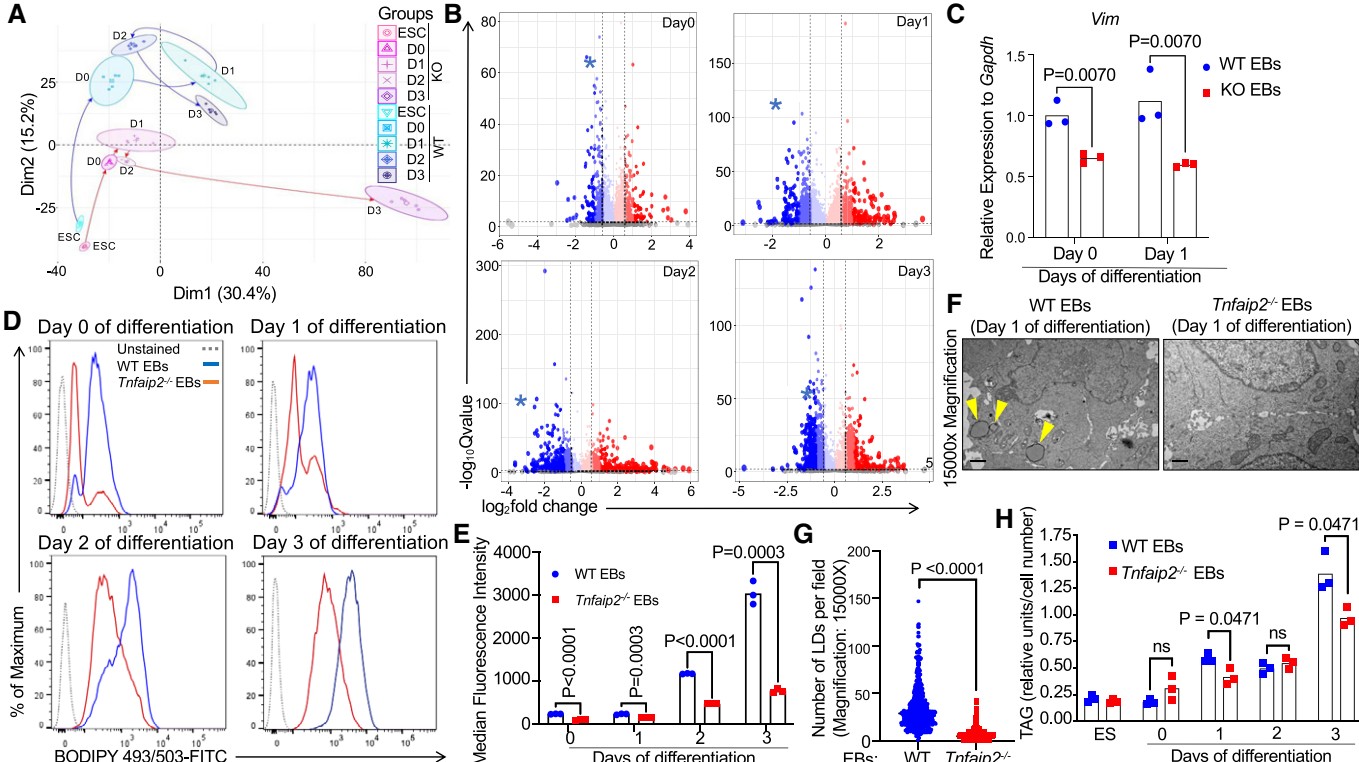

**Figure 8.**

**Figure 8.** *Tnfaip2* depletion impairs *Vim* expression, lipid droplet (LD) formation (LD), and triacylglycerols (TAGs) of differentiation-induced embryoid bodies.

A–F   One thousand ESCs of the indicated genotypes were placed into hanging drop cultures to form EBs. After 3 days (referred to as day 0), differentiation was induced by plating EBs onto gelatin-coated tissue culture dishes in differentiation-inducing medium. (A) Principal component analysis (PCA) of the proteome profiles of *Tnfaip2*$^{-/-}$ (KO) versus WT ESCs and EBs on day 0 to day 3 after differentiation induction (*n* = 5 biological replicates per group, enlarged symbol denotes the mean). (B) The volcano plots show the differentially expressed proteins identified from the proteome analysis of *Tnfaip2*$^{-/-}$ EBs versus WT EBs at the indicated days after induction of differentiation. *Vim* (marked in blue asterisk) was the only protein that featured among top 100 differentially expressed proteins at all time points by implementing selection criteria of an average log$_2$ratio> ±1.2 and a -log$_{10}$Qvalue < 2. *Vim* expression was reduced at all time points of *in vitro* differentiation in *Tnfaip2*$^{-/-}$ EBs versus WT EBs (*n* = 5 biological replicates, see also Dataset EV8 for top 100 regulated proteins). Sphere size illustrates the magnitude of log$_2$fold changes as does color coding: blue-downregulated proteins, red-upregulated proteins; log2fold marking by color intensity: strong intensity> 2, medium intensity> 1.5, faded color> 1. (C) mRNA expression of *Vim* on indicated days of WT EB and *Tnfaip2*$^{-/-}$ EB differentiation (*n* = 3 biological replicates; log-transformed data were normally distributed (*P* > 0.05 as per Shapiro–Wilk test) and analyzed by multiple *t*-tests with Holm–Sidak correction for multiple testing). (D,E) FACS analysis of lipid droplet (LD) content of WT (blue) and *Tnfaip2*$^{-/-}$ (red) EBs by BODIPY493/503 staining at the indicated days after differentiation induction: (D) Representative FACS blot of BODIPY staining. Unstained WT EBs served as a negative control (gray). (E) Quantification of staining intensity [*n* = 3 independent cultures per genotype; data were normally distributed (*P* > 0.05 as per Shapiro–Wilk test) and analyzed by multiple *t*-tests with Holm–Sidak correction for multiple comparisons]. (F, G) Transmission electron microscopy (TEM) was used to determine the number of LDs in EB cultures derived from WT and *Tnfaip2*$^{-/-}$ ES cells on day 1 after differentiation induction. (*n* = 5 independent cultures per genotype, *n* = 100 images per replicate). (F) Representative micrographs of TEM analysis of EB cells of the indicated genotype. Yellow arrows point to LDs, scale bars: 1 μm. (G) Quantification of LDs per TEM field. Data were not normally distributed (*P* < 0.05 as per Shapiro–Wilk test) and analyzed by Mann–Whitney test. (H) UPLC-MS/MS analysis revealed a reduction in triacylglycerol (TAG) content in *Tnfaip2*$^{-/-}$ EBs compared to WT EBs at the indicated time points after differentiation induction [*n* = 3 independent cultures of EBs per time point; data of EBs on day 0 to day 3 of differentiation were statistically analyzed; mean centered group data were normally distributed and (*P* > 0.05 as per Shapiro–Wilk test) and analyzed by one-sided *t*-test with Holm–Sidak correction for multiple testing].

*Tnfaip2* knockdown as well as *Tnfaip2* overexpression and *Vim* knockdown to functionally test the hierarchy of these two genes on reprogramming efficiency of 4-factor (OSKM) transduced MEFs (Fig 11A and B). While *Tnfaip2*-depleted MEFs with ectopic expression of *Vim* led to a significant reduction in the formation of iPSCs as compared to control vector-targeted MEFs, *Vim*-depleted MEFs overexpressing *Tnfaip2* yielded a significantly higher reprogramming efficiency compared to the controls (Fig 11C and D). Together, these data revealed genetic evidence that *Tnfaip2*-dependent induction of *Vim* acts epistatically in inhibiting the reprogramming of MEFs into iPSCs.

To analyze the influence of FA on *Tnfaip2*-dependent effects on reprogramming, MEF were reprogrammed with four factors (4F, see above) with or without co-treatment of the cultures with Etomoxir (O'Connor *et al*, 2018)—an irreversible inhibitor of the FA-transporter CPT1A (see above). Etomoxir treatment significantly increased the reprogramming rate of MEFs into iPSCs (Fig 11E and F). Of note, supplementation of Etomoxir to *Tnfaip2*-depleted MEFs undergoing cellular reprogramming did not lead to an additional increase in generation of iPSCs as compared to either of the two interventions by itself (Fig 11E and F). Collectively, these data suggested that *Tnfaip2*, *Vim*, and *Cpt1a* are part of the same mechanism that impairs reprogramming of MEFs into iPSCs.

### Application of palmitic acid rescues organ homeostasis of *Smed-exoc3*-depleted planarians

Our above data indicated that defects in organ homeostasis in planarians in response to *Smed-exoc3*-depletion are associated with an accumulation of neoblast stem cells disturbances in the expression of stemness and differentiation regulating genes in neoblast cells (Fig 5A–F and Dataset EV3–EV5). Interestingly, the expression levels of various components of the exocyst complex including *Exoc3* increase during LD formation and colocalize with LDs (Inoue *et al*, 2015). Since our experiments on ESCs and iPSC had revealed a critical role of TAGs in mediating EB differentiation (Fig 9C–H), we explored the TAG profile of *exoc3(RNAi)*-treated planarians

compared to *gfp(RNAi)*-treated controls by analyzing the abundance and composition of major triglyceride species by ultraperformance liquid chromatography ESI tandem mass spectrometry. Intriguingly, like *Tnfaip2*$^{-/-}$ EBs, targeted quantification of TAGs (at day 38 of the RNAi injection protocol, Fig 2B) revealed that *Smed-exoc3*-depleted planarians, like *Tnfaip2*$^{-/-}$ EBs, contain significantly lower levels of TAGs compared to controls (Fig 12A). To test whether PA/PC treatment could also rescue the *in vivo* defects in differentiation of somatic stem cells and the impairments in organ maintenance in *Smed-exoc3*-depleted planarians, a rescue experiment was conducted by cotreating *Smed-exoc3*-depleted planarian with or without PA/PC (Fig 12B). Control experiments revealed that the PA/PC cotreatment did not affect the knockdown efficiency of *exoc3 (RNAi)* treatment in planarians (Fig 12C). It also did not rescue the double-head phenotype of Smed-β-*catenin-1*-depleted, amputated planarian (Fig 12D) a well-documented, impressive RNAi phenotype in planarians (Iglesias *et al*, 2008; Petersen & Reddien, 2008; Gurley *et al*, 2008). Of note, PA/PC cotreatment of *exoc3(RNAi)*-treated planarians rescued organ maintenance in 20 out of 23 animals (Fig 12E) and the expression of marker genes of differentiating progenitor cells from the epidermal lineage (Fig 12F). Together, these results indicate that PA/PC delivery rescues impairments in organ maintenance of *Smed-exoc3*-depleted planarians without affecting the knockdown efficiency of dsRNA injection.

## Discussion

The current study provides experimental evidence that *Tnfaip2/Exoc3* have a critical role in instructing stem cell differentiation by inducing *Vim*-dependent LD formation. LDs represent major hubs for lipid metabolism involving the biosynthesis, degradation, mobilization, and distribution of TAGs as major components of the LD inner core (Wilfling *et al*, 2013; Olzmann & Carvalho, 2019). TAGs are not only important for the storage of lipids but are also needed for lipid metabolism and the supply of cells with FAs, which have a crucial role in differentiation induction in various stem cell systems

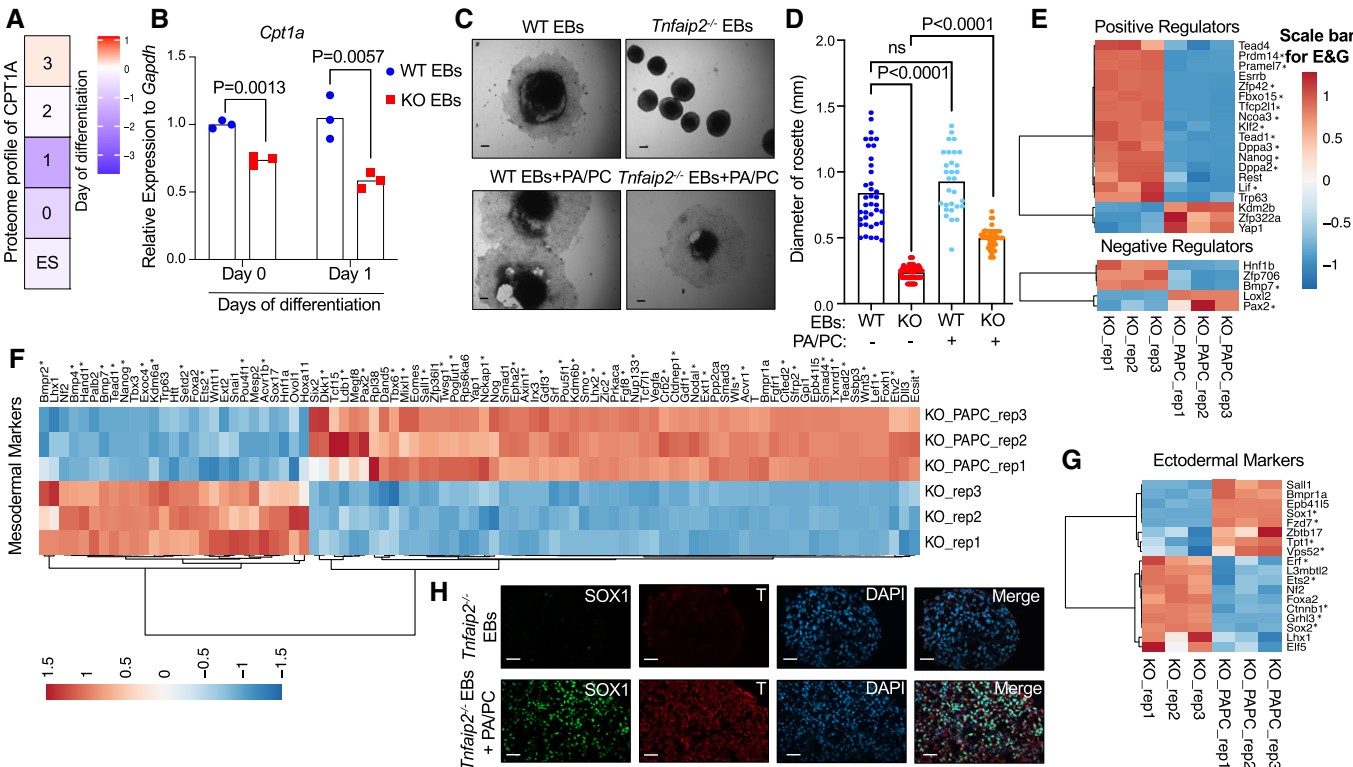

**Figure 9. Restoration of differentiation capacity of *Tnfaip2⁻/⁻* ESCs by TAG supplementation.**

A Heat map of CPT1A protein expression as measured in proteome analysis of *Tnfaip2⁻/⁻* and WT ESCs and during differentiation induction of EB cultures (*n* = 5 biological replicates). The color scale represents average log₂ratios.

B Expression of *Cpt1a* mRNA as determined by qRT–PCR in WT and *Tnfaip2⁻/⁻* EBs at the indicated days of differentiation induction [*n* = 3 biological replicates; log-transformed data were normally distributed (*P* > 0.05 as per Shapiro–Wilk test) and analyzed by multiple *t*-tests with Holm–Sidak correction for multiple comparisons].

C–H Differentiation-induced WT EBs and *Tnfaip2⁻/⁻* EBs were treated with palmitic acid (PA) and palmitoyl-L-carnitine (PC; 8 μM each) or a vehicle control (C, D) The diameters of the rosette structure of differentiated WT EBs and *Tnfaip2⁻/⁻* EBs were measured on day 3 after differentiation induction (*n* = 3 repeat experiments with 7–62 EBs per experiment per group): (C) Representative photographs (scale bar: 0.02 mm) and (D) quantification of the diameter of the rosette structure of the indicated groups [data were not normally distributed (*P* < 0.05 as per Shapiro–Wilk test) and thus analyzed by Mann–Whitney *U*-test with Holm–Sidak correction for multiple comparisons]. (E-G) RNA-seq was conducted on day 3 of differentiation induction of *Tnfaip2⁻/⁻* EBs that were either treated with PA/PC or with a vehicle control (*n* = 3 independent pools of 7–62 EBs per group). The heat maps depict the intersection of DEGs with the following gene sets retrieved from the AmiGO 2 data base (Carbon *et al*, 2009): (E) "stem cell maintenance: positive and negative regulators", (F) "mesoderm development", and (G) "ectoderm development". Five other stem cell-related genes (*Tfcp2l1, Dppa3, Fbxo15, Zfp42,* and *Klf2*) were additionally incorporated based on literature searches. The asterisks indicate the reverted changes in gene expression that were seen in *Tnfaip2⁻/⁻* EBs versus WT EBs depicted in Fig 7B–D. The color scale represents the gene-wise *z*-score calculated from normalized gene expression levels. (H) Representative image of immunofluorescence staining against ectodermal marker (SOX1) and the mesodermal marker (T alias Brachyury) on day 3 of differentiation induction of *Tnfaip2⁻/⁻* EBs ± cotreatment with PA/PC. Three repeat experiments were conducted with a total number of nine EBs per group, same experiment as in Fig 7E (scale bar: 20 μm).

(Gurumurthy *et al*, 2010; Yanes *et al*, 2010; Ito *et al*, 2012; Bailey *et al*, 2015; Lahvic *et al*, 2018; Xie *et al*, 2018; Obniski *et al*, 2018; Sênos Demarco *et al*, 2019). Interestingly, FA deprivation in culture medium is sufficient to stall human pluripotent stem cells (hPSCs) in a pluripotent state at the transition of naïve to primed state hPSCs (Cornacchia *et al*, 2019) implying that lipid metabolism and FA supply is an essential step for the exit from pluripotency. This could also be important for *in vivo* differentiation of pluripotent cells as lipid metabolism is employed during early embryogenesis by the use of fat storages in LDs of oocytes (Johnson *et al*, 2003).

The current study indicates that *Tnfaip2* has a critical role in TAG biosynthesis and LD formation during early stages of ESC differentiation in culture. The failure to increase intracellular TAG levels and LD formation in *Tnfaip2*-deficient EB cultures is associated with a complete inhibition of differentiation induction, which, however, can be fully rescued by FA treatment of the cultures. These findings support the conclusion that *Tnfaip2*-knockout ESCs are in principle differentiation competent if the failure in lipid biosynthesis and FA production can be overcome.

The exact role of *Tnfaip2* in inducing lipid biosynthesis remains to be delineated. It is conceivable that it involves known functions of *Tnfaip2* in protein and organelle trafficking (for review see Jia *et al*, 2018). Our proteomic studies point to a potential role of *Tnfaip2*-dependent *Vim* induction in this process. VIM protein expression is strongly upregulated in response to differentiation induction of ESCs but the deletion of *Tnfaip2* completely abrogates this response during the whole time course of early differentiation of EB cultures on day 0–3 coinciding with a failure of

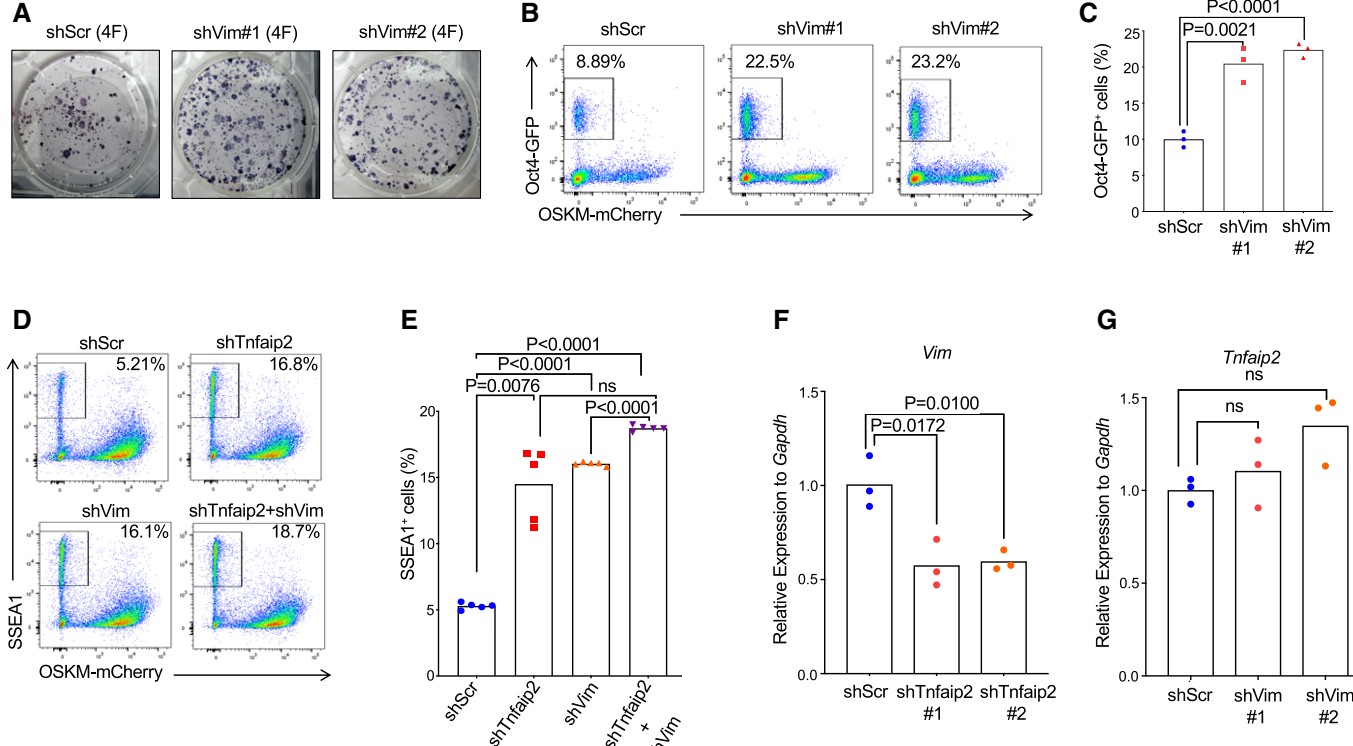

**Figure 10. *Tnfaip2* and *Vim* act epistatically to impair cellular reprogramming.**

A–G  *Oct4*-eGFP reporter MEFs or WT MEFs were infected with (A–C) shRNAs or (D, E) combinations of shRNAs. Cells were co-infected with four reprogramming factors (OSKM) and double-infected cells were FACS-sorted and grown for 14 days, when reprogramming efficiencies were determined by FACS. (A) Representative images of AP staining of iPSC colonies and (B) representative FACS profiles of *Oct4*-positive iPSCs on day 14, (C) quantification of the percentage of *Oct4*-GFP⁺ iPSCs in MEF cultures that were infected with two different shRNAs targeting *Vim* or a scrambled shRNA control [n = 3 biological replicates per group; data were normally distributed (P > 0.05 as per Shapiro–Wilk test) and analyzed by t-test with Holm–Sidak correction multiple comparisons]. (D) Representative FACS profiles of mouse-specific pluripotency cell surface marker SSEA1⁺ iPSCs on day 14 of reprogramming and (E) quantification of the percentage of SSEA1⁺ iPSCs in MEF cultures that were infected with the indicated shRNAs targeting *Vim* and/or *Tnfaip2* [n = 5 biological replicates; data were normally distributed (P > 0.05 as per Shapiro–Wilk test) but showed unequal variance; data were analyzed by Brown–Forsythe and Welch ANOVA and Dunnett's multiple comparisons test]. (F, G) RT–qPCR was performed to determine (F) *Vim* mRNA expression in scrambled shRNA-infected MEFs versus shRNA-*Tnfaip2*-infected MEFs or (G) *Tnfaip2* mRNA expression in scrambled shRNA-infected MEFs versus shRNA-*Vim*-infected MEFs. (F,G) Infected cells were purified by BFP-sorting on day 4 after infection [n = 3 biological replicates; log-transformed data were normally distributed (P > 0.05 as per Shapiro–Wilk test) and analyzed by t-test with Holm–Sidak for multiple comparisons].

differentiation. TNFα signaling has previously been reported to activate VIM (Mor-Vaknin *et al*, 2003), and VIM is known to instruct cell differentiation in the hematopoietic system (Benes *et al*, 2006) as well as cell fate changes during epithelial–mesenchymal transition (Mendez *et al*, 2010). How *Tnfaip2*-dependent processes—for example in protein trafficking—contribute to the induction of VIM expression appears as an important area of future research. Interestingly, the knockdown of *Vim* phenocopied the knockdown of *Tnfaip2* in enhancing reprogramming of MEFs into iPSCs but codeletion of both genes did not lead to additive effects on iPSC formation. In addition, *Vim* overexpression abrogates enhancements in reprogramming efficiency induced by *Tnfaip2* knockdown, whereas the overexpression of *Tnfaip2* does not affect the increase in reprogramming efficiency induced by *Vim* knockdown. These results support the conclusion that *Tnfaip2/Vim* act epistatically in controlling the same mechanistic process, which suppresses the reprogramming of MEFs into iPSCs. The study also shows that inhibition of the FA carrier, CPT1A,

increases the reprogramming efficiency of MEFs into iPSCs to a similar level as *Tnfaip2* knockdown, and there is no additive effect of both approaches. Together, these results reveal a new role of *Tnfaip2*-mediated *Vim* induction, *Vim*-mediated LD formation, and FA metabolism/transport in impairing the reprogramming of MEFs into iPSCs.

Unlike in mouse or in other mammals, wherein pluripotent stem cells are restricted to early phases of embryonic development, pluripotent somatic stem cells are present in planarians throughout their life to ensure homeostasis and regeneration of all tissues of the worms. Thus, planarians provide a unique *in vivo* model system to study the function of genes and pathways in regulating pluripotent somatic stem cells *in vivo*. In vertebrates, organ regeneration and homeostasis are driven by organ-specific, adult stem cells with restricted differentiation capacity, including multipotent, oligopotent, or unipotent stem cells. The current study shows that it is possible to use planarians to delineate conserved gene functions as well as gene

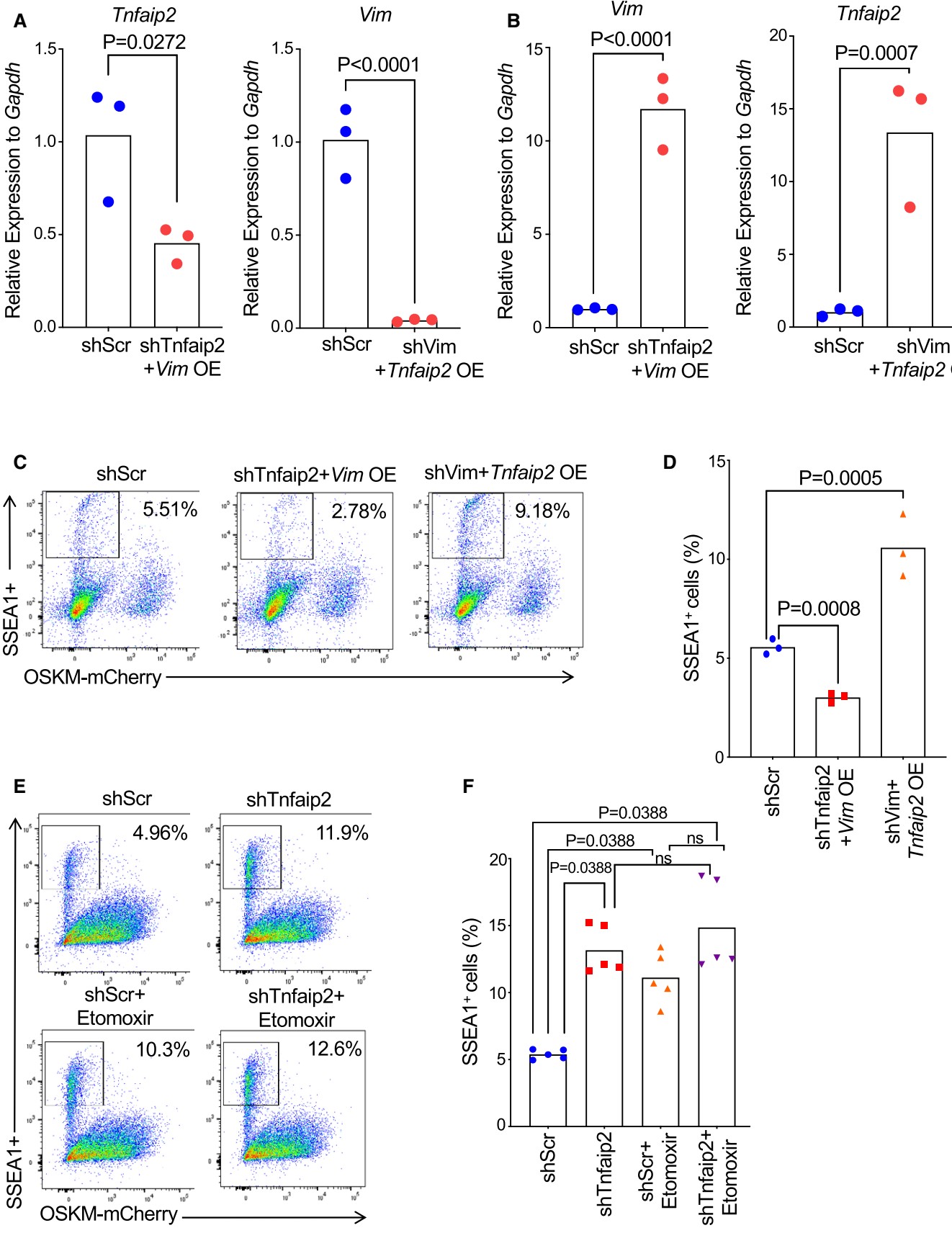

**Figure 11.**

**Figure 11.  *Vim* and *Cpt1a* act downstream of *Tnfaip2* in impairing cellular reprogramming.**

*Oct4*-eGFP reporter MEFs or WT MEFs were (A-D) infected with a vector expressing the indicated shRNA in combination with the indicated cDNA (OE indicates overexpression), or (E,F) infected with the indicated shRNA with or without combined treatment with Etomoxir—a chemical inhibitor of CPT1A.

A–F   Cells were co-infected with four reprogramming factors (4F = *Oct4*, *Sox2*, *Klf4*, *c-myc* = OSKM), and double-infected cells were FACS-sorted for analysis. (A) RT–qPCR analysis on day 4 after infection to determine RNA expression of *Tnfaip2* or *Vim* in *Tnfaip2*-depleted cells with ectopic expression of *Vim* (left) or in *Vim*-depleted cells with ectopic expression of *Tnfaip2* (right). Scramble shRNA (shScr)-infected MEFs served as a control [$n = 3$ biological replicates; log-transformed data were normally distributed ($P > 0.05$ as per Shapiro–Wilk test) and analyzed by *t*-test]. (B) RT–qPCR analysis on day 4 after infection to determine RNA expression of *Vim* or *Tnfaip2* in *Tnfaip2*-depleted cells with ectopic expression of *Vim* (left) or in *Vim*-depleted cells with ectopic expression of *Tnfaip2* (right). Scramble shRNA (shScr)-infected MEFs served as a control [$n = 3$ biological replicates; log-transformed data were normally distributed ($P > 0.05$ as per Shapiro–Wilk test) and analyzed by *t*-test]. (C) Representative images of FACS profiles and (D) quantification of SSEA1$^+$ iPSCs on day 14 after infection of MEF cultures with the indicated combination of shRNAs plus cDNAs (OE) or scrambled shRNA control [$n = 3$ independent cultures per group; data were normally distributed ($P > 0.05$ as per Shapiro–Wilk test) and analyzed by one-way ANOVA and Dunnett's multiple comparison test] (E) Representative FACS profiles and (F) quantification of the percentage of SSEA1$^+$ iPSCs in MEF cultures on day 18 after infection with shRNAs targeting *Tnfaip2* or a scrambled shRNA control with or without continuous treatment with Etomoxir [$n = 5$ biological replicates; data were not normally distributed ($P < 0.05$ as per Shapiro–Wilk test) and analyzed by Mann–Whitney *U*-test with Holm–Sidak correction for multiple comparisons].

specification that control the function of pluripotent, embryonic stem cells in culture as well as pluripotent, somatic stem cells at the organism level (in planarians). It seems promising to go on to use this approach to identify novel mechanisms that control the function of restricted, somatic stem cells in more complex organisms such as vertebrates and mammals.

This study reveals that the planarian para-ortholog of *Tnfaip2*, *Smed-exoc3*, has an essential role in organ homeostasis and regeneration in planarians. The knockdown of *Smed-exoc3* does not lead to a failure in the maintenance of somatic stem cells. However, the study shows that it leads to defects in organ maintenance and regeneration in planarian. Of note, the administration of FAs (PA and its mobilized form PC) was sufficient to completely prevent these defects in organ homeostasis suggesting that failures in FA synthesis may also represent a major reason for the abrogation of differentiation capacity of somatic stem cells in *Smed-exoc3*-depleted planarians. These findings suggest that in vertebrates, both homologs, *Tnfaip2* and *Exoc3*, have a conserved role in maintaining stem cell function by promoting LD formation and the induction of lipid metabolism. This conclusion is also supported by the finding that the knockdown of both genes enhances the reprogramming efficiency of somatic cells into iPSCs suggesting that the role of *Tnfaip2/Exoc3* in

differentiation induction impairs the transition of somatic cells into pluripotency. Interestingly, a crucial role of autophagy-dependent FA generation has been demonstrated for differentiation of neutrophils (Riffelmacher *et al*, 2017) suggesting that the here identified axis of *Tnfaip2/Exoc3*-dependent LD formation and TAG synthesis may also be important for differentiation of more mature cell types. It is possible that additional mechanisms that are mediated by *Tnfaip2* contribute to the control of differentiation and somatic cell function. It has been reported that expression of *Tnfaip2* in iPSC-derived mesenchymal cells is critical for the formation of tunneling nanotubes (TNTs) that mediate mitochondrial transfer into differentiated cardiomyocytes (Zhang *et al*, 2016). It is tempting to speculate that mitochondrial transfer via TNTs could also contribute to the control of differentiation of stem cells during asymmetric cell division of stem cells into self-renewing stem cells and differentiating progenitor cells.

Altogether, the current study identified a critical role of *Tnfaip2/Exoc3* in differentiation of ESCs as well as in maintaining tissue homeostasis and regeneration *in vivo*. Mechanistically, the study shows that *Tnfaip2/Exoc3*-dependent induction of lipid biosynthesis and FA production is to control stem cell differentiation and organ maintenance. These findings could have broader

**Figure 12.   Rescue in organ homeostasis in *Smed-exoc3* depleted planarians by FA supplementation.**

A   TAG levels in *exoc3(RNAi)*-treated planarians versus *gfp(RNAi)*-treated controls as determined by UPLC-MS/MS, [$n = 4$ experimental replicates on pools of 25 planarians per group per experiment; data were normally distributed ($P > 0.05$ as per Shapiro–Wilk test), variances were different between the groups ($P < 0.05$); thus, data were analyzed by Welch's *t*-test].

B   Schematic representation of the regime of RNAi treatment in combination with PA/PC (750 μM each, 160 nl per worm). PA/PC was administered 3 h before injecting dsRNA against *Smed-exoc3* or against *gfp* as a control.

C   RT–qPCR analysis of *exoc3* mRNA expression in planarians of the indicated genotypes and treatment schedule [$n = 3$ biological replicates; log-transformed data were normally distributed ($P > 0.05$ as per Shapiro–Wilk test) and analyzed by one-way ANOVA and Dunnett's multiple comparison test].

D   Representative images of regenerating planarians that were injected with *gfp*-RNAi, Smed-β-catenin-1-RNAi, or Smed-β-catenin-1-RNAi in combination with PA/PC injection (as described in panel B) on day 7, 8 and 9. Heads and tails were cut on day 10, images were taken on day 17 after amputation. (3 repeat experiments with 3–4 biological replicates per experiment; red dotted lines demarcate sites of amputation, scale bar: 0.5 mm). Number ratios in the photographs indicate the total number of animals exhibiting the represented phenotype.

E   Images of body plan maintenance in non-injured planarians that were injected with *gfp*-RNAi, *exoc3*-RNAi, or *exoc3*-RNAi in combination with PA/PC injection (as described in panel B, $n = 3$ experimental replicates with 6–8 biological replicates per experiment, scale bar: 0.5 mm). Number ratios in the photographs indicate the total number of animals exhibiting the represented phenotype.

F   RT–qPCR analysis of marker genes of early stem cell differentiation (*prog-1* and *prog-2*) and late stem cell differentiation (*Smed-odc1*) in planarians exposed to the indicated treatment regiments on day 38 of the experiment as indicated in panel B [$n = 3$ biological replicates; log-transformed data were normally distributed ($P > 0.05$ as per Shapiro–Wilk test) and analyzed by one-way ANOVA and Dunnett's multiple comparison test].

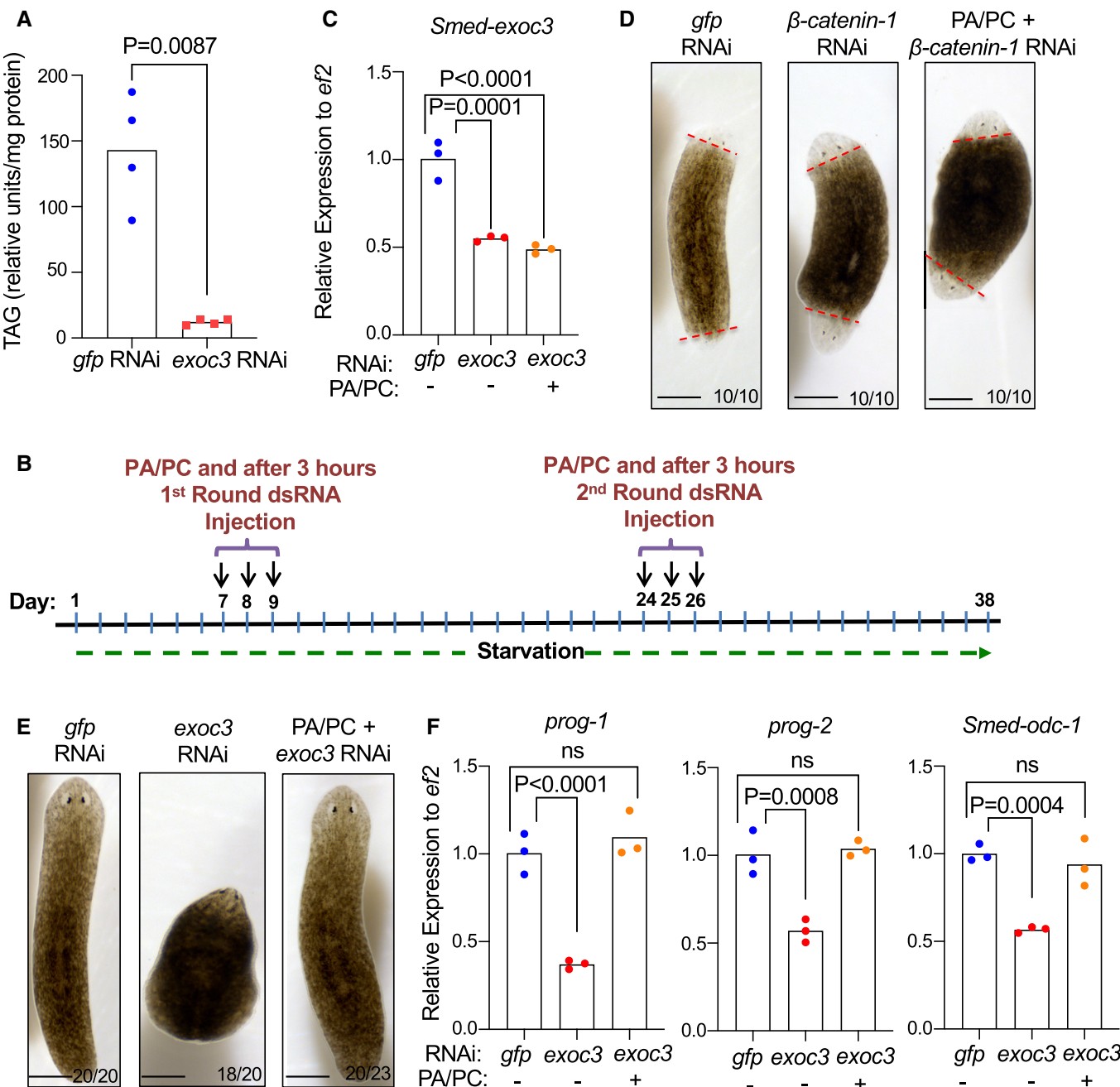

**Figure 12.**

implications for aging and carcinogenesis by linking TNFα/NFκB inflammatory signaling with the control of lipid metabolism, stem cell differentiation, and organ maintenance. Impairments in stem cell function and organ maintenance are associated with increases in aging-associated diseases including cancer. It remains to be investigated whether the chronic activation of inflammatory signaling during aging and carcinogenesis would lead to chronic induction of *Tnfaip2/Exoc3*-driven lipid metabolism in tissue-resident stem cells and how this would affect tissue maintenance and cancer initiation.

## Materials and Methods

### Preparation of mouse embryonic fibroblasts

Murine embryonic fibroblasts (MEF) cultures were established from E13.5-day embryos of either wild-type (WT) or *Oct4*-eGFP transgenic C57BL/6J mice as described previously (Takahashi *et al*, 2007). Pregnant female mice, 13.5 days post-plug conformation (E13.5), were sacrificed by cervical dislocation, and *uteri* were removed to isolate the embryos following a quick wash in PBS. The

embryos devoid of placenta, head, visceral tissue, and liver were washed once again in PBS and were minced. The minced embryonic tissues were suspended in 3 ml of TrypLE Select (cat. no. 12563029; Thermo Fisher Scientific) for 15 min at 37°C for further dissociation of tissues. An equivalent amount of mouse embryonic fibroblast (MEF) growth medium, constituted by DMEM containing 15% FBS (Gibco), 1% L-glutamine (Gibco), 1% penicillin/streptomycin (Gibco), and 1% non-essential amino acids (Gibco), was added, and tissues were dissociated by vigorous pipetting. Debris was removed; the dissociated cells were plated on sterile plastic 10-cm tissue culture plates and incubated in a tissue culture incubator at 37°C (Eppendorf, New Brunswick Galaxy 170R) for 24 h. MEFs used for generation of induced Pluripotent Stem cells were not passaged more than three times to avoid replicative senescence.

### Lentiviral RNAi screening and generation of induced pluripotent stem cells (iPSCs)

L. Zender (University Hospital, Tübingen, Germany) provided Cancer 1,000 library. It comprises of a pool of 1,774 sequence-verified miR30-based shRNAs targeting approximately 1,000 mouse orthologs of well characterized as well as putative human cancer-related genes (oncogenes as well as tumor suppressors compiled from microarray expression data and literature mining (Witt *et al*, 2006) along with an *eGFP* reporter was deployed for the RNAi screen. This version of focused RNAi library has been widely used to identify attenuators in different biological contexts (Bric *et al*, 2009; Meacham *et al*, 2009; Wang *et al*, 2012). Briefly, mouse embryonic fibroblasts (MEFs) derived from E13.5 C57BL/6J mice were cotransduced in MEF medium with the shRNA library along with a polycistronic vector overexpressing *Oct4*, *Sox2*, *Klf4*, *c-Myc* (OSKM) carrying mCherry reporter (Sommer *et al*, 2009) in a manner that facilitates single lentiviral integration of both OSKM and the shRNA library with 20 × representation (i.e. 20 infected cells per shRNA and replicate). To this end, the transduction efficiency of both the lentiviral plasmid was kept at 30%. Following 4 days of transduction at 37°C (Eppendorf, New Brunswick Galaxy 170R), cells were resuspended in FACS buffer (2% FBS in HBSS) and filtered through 40-μm cell filters and the double positive (mCherry$^+$/GFP$^+$) cells were FACS-sorted and were allowed to reprogram by plating the cells in the mouse embryonic stem (ES) cell medium (containing 15% KnockOut™ D-MEM (Gibco) supplemented with Knockout Serum Replacement (Gibco), L-glutamine (Gibco), penicillin/streptomycin (Gibco), non-essential amino acids (Gibco), β-mercaptoethanol (Sigma), and 1,000 U/ml leukemia inhibitory factor (Millipore)) for another 14 days at 37°C with 5% CO$_2$ (Thermo Scientific, BBD6220) and the medium was changed on every other day. After 14 days, the emerging iPS cells (SSEA1$^+$/mCherry$^-$) were similarly sorted by staining the cells with SSEA1 antibody (cat. no. 125608; BioLegend) at 1:20 dilution for 30 min. Cells were then washed in PBS twice, pooled, and followed by PCR amplification of the shRNA sequences from the genomic DNA using primers flanking the miR30 cassette (for details see Wang *et al*, 2012) which was then subjected to deep sequencing using Illumina HiSeq2500 platform in 50 cycle/single-end/high-output sequencing mode with extraction of reads in fastq format using bcl2fastq v1.8.4 to follow up the representation of individual shRNAs in these cells compared

to their non-iPS (SSEA1$^-$) counterparts. The data were analyzed as described previously in Wang *et al*, 2012. Cell sorting was performed using FACSAriaIII (Becton Dickinson) with Diva Software (v8.0.1). The experiment was conducted in biological triplicates. For single-hit validation and all other experiments related to the generation of iPSCs, $1 \times 10^5$ MEFs were seeded in 10 cm culture plates and cotransduced in MEF medium with 15 μl of concentrated pHAGE-STEMCCA (comprising of the polycistronic cDNA constructs for *Oct4*, *Sox2*, *Klf4* and *c-Myc* besides mCherry reporter) virus and a vector coding for the shRNAs against candidate genes along with BFP reporter. Four days following transduction, mCherry$^+$/BFP$^+$ cells were FACS-sorted. 50,000 FACS-sorted transduced (mCherry$^+$) cells were reseeded on a well of a 6-well plate and incubated in ES cell medium with or without 5 μM of Etomoxir and replenished every alternative day for another 14 days. iPSC formation was analyzed by AP staining and colony counting or by sorting cells positive for *Oct4*-GFP/SSEA-1 pluripotency marker and negative for mCherry expression using FACSAriaIII (Beckton Dickinson) and analyzed with Diva Software (v8.0.1).

### Mouse RNA purification

Total RNA was extracted from mouse embryonic fibroblasts and embryoid bodies using the Masterpure RNA Purification Kit (Epicentre) as described in Muniz *et al* (2017). The RNA was suspended to a concentration of 100 ng/μl using nuclease-free water (Epicentre). The concentration of the RNA was measured using a spectrophotometer (NanoDrop 2000c; Thermo Fisher Scientific).

### Quantitative RT–PCR (RT–qPCR)

200ng of purified RNA was reverse-transcribed to generate cDNA. The DNA polymerase IQ SYBR Supermix (Bio-Rad) was used according to supplier's information to identify the expression levels of the genes. Real-time PCR was performed with ABI 7500 Real-Time PCR System (7500 Software v2.0.6; Applied Biosystems). Primer sequences are provided in Table EV1. Data were analyzed by relative/comparative quantification method (Pfaffl, 2001). For the calculation of delta-delta Ct-values, delta Ct-values of individual groups were scaled by subtracting the mean delta Ct-value of the respective control group in each of the experiments.

### Production and transduction of lentivirus

Lentiviruses were produced using a 2-plasmid transfection system (psPAX2 and pMD2.G) along with desired plasmid as previously described (Schambach *et al*, 2006). For this Lenti-X, cells (Clontech) were cotransfected with 15 μg of shRNA or polycistronic reprogramming vector, 10 μg of psPAX2 helper plasmid, and 5 μg of pMD2.G at 37°C (Eppendorf, New Brunswick Galaxy 170R). Following 48 h of transfection, viral supernatants were harvested every 12 h on two consecutive days and the lentiviral particles were concentrated by centrifugation at 25,000 rpm for 1.5 h at 4°C and were suspended in PBS. This concentrated lentiviral particle was again filtered using 0.22-μm filter to ensure that it is devoid of any cells. Transduction was carried out in MEF medium supplemented with

8 μg/ml polybrene (Sigma) at 37°C (Eppendorf, New Brunswick Galaxy 170R). After 16 h, the MEF medium was replaced with fresh MEF medium and was kept in an incubator at 37°C (Eppendorf, New Brunswick Galaxy 170R). Four days following transduction, the transduced cells were FACS-sorted on FACSAriaIII (Beckton Dickinson) with Diva Software (v8.0.1).

## Alkaline phosphatase staining

Alkaline phosphatase (AP) staining of iPS colonies was performed with 5-bromo-4-chloro-3-indolyl phosphate (BCIP)/NitroBlue Tetrazolium (NBT) Alkaline Phosphatase (AP) Substrate Kit (cat. no. SK-5400; Vector Laboratories) in accordance to the supplied manual.

## CRISPR/Cas9-mediated generation of Tnfaip2 knockout (Tnfaip2$^{-/-}$) murine ES cell lines

CRISPR-Cas9-mediated *Tnfaip2*$^{-/-}$ mouse ES cells (provided by Dr. Francesco Neri) were generated by the Functional Genomics Core Facility of the FLI. In order to knockout the *Tnfaip2* gene, an oligo-encoding gRNA targeting this gene was inserted into the px458 vector expressing *eGFP* (a kind gift from Dr. Feng Zhang). The Zhang Lab website http://crispr.mit.edu was used to choose the unique gRNA sequence. 4 μg of this plasmid construct harboring the gRNA (sequence information in Table EV1) against *Tnfaip2* was transiently transfected by electroporation in ES cells. 2–3 days later, cells positive for GFP were sorted by FACS using FACSAriaIII (Beckton Dickinson) and analyzed with Diva Software (v8.0.1) in a manner that allows each GFP$^+$ cell to be collected in each well of a 96-well plate. Once the cells grew, the plate was duplicated thereby generating 1 "Master Plate", while the other plate was used for DNA isolation and Sanger sequencing to identify a homozygous knockout clone (sequence information in Fig EV3). The clone with the desired out-of-frame sequence was expanded using the corresponding clone from the "Master Plate".

## Embryoid body formation

Embryoid bodies (EBs) were generated using hanging drop approach and differentiated as described previously (Takahashi & Yamanaka, 2006; Behringer *et al*, 2016). Briefly, an *in vitro* EB differentiation assay through EB formation was conducted using wild-type and *Tnfaip2*$^{-/-}$ ES cells (E-14). $3.75 \times 10^4$ ES cells/ml were resuspended in ES cell medium and rows of 27 μl drops (1,000 cells per drop) on the up-turned inner surface of the lid of the 10cm tissue culture dish were made and incubated at 37°C (Eppendorf, New Brunswick Galaxy 170R) for 3 days to form EBs. The EBs (at day 0) were plated onto 0.1%-gelatin-coated tissue culture dishes to initiate *in vitro* differentiation by incubating for another 3 days in MEF medium at 37°C (Eppendorf, New Brunswick Galaxy 170R). After 3 days (of *in vitro* differentiation), the EBs were harvested.

## Immunostaining of embryoid bodies

Immunofluorescence staining was performed as previously described (Murakami *et al*, 2016). The following primary antibodies

were used: rabbit anti-SOX1 (1:200, Cell Signaling, 4194) and goat anti-T (1:200, R&D Systems, AF2085).

## Sample preparation for Mass Spectrometry (MS)

Embryoid bodies (five biological replicates per condition/time point) were thawed (after storage at −80°C) and were lysed by addition of 300 μl of lysis buffer (fc 4% SDS, 100 mM HEPES, pH 8.5, 50 mM DTT). Samples were then sonicated in a Bioruptor (Diagenode, Belgium; 10 cycles with 1 min on and 30 s off with high intensity @ 20°C) and heated at 95°C for 10 min, before being subjected to another round of sonication in the Bioruptor. The lysates were clarified, and debris were precipitated by centrifugation at 14,000 rpm for 10 min and then incubated with 15 mM of iodoacetamide at room temperature, in the dark for 20 min. After removal of the supernatant, the precipitates were washed twice with 200 μl of a solution of ice cold 80% acetone. The pellets were then allowed to air-dry before being dissolved in digestion buffer at 1 μg/μl (3 M urea in 0.1 M HEPES, pH 8). A 1:100 w/w amount of LysC (Wako sequencing grade) was added to each sample, and then, they were incubated for 4 h at 37°C with shaking (1,000 rpm). The samples were diluted at 1:1 ratio with milliQ water and were incubated with a 1:100 w/w amount of trypsin (Promega) overnight at 37°C, 650 rpm. The digests were then acidified with 10% trifluoroacetic acid and then desalted with Waters Oasis® HLB μElution Plate 30 μm. In this process, the columns were conditioned with $3 \times 100$ μl solvent B (80% acetonitrile; 0.05% formic acid) and equilibrated with $3 \times 100$ μl solvent A (0.05% formic acid). The samples were loaded and washed thrice with 100 μl solvent A. The eluates were dried down and dissolved in 5% acetonitrile with 0.1% formic acid at a concentration of 1 μg/μl. 10 μl were transferred to an MS vial, and 0.25 μl of HRM kit peptides (Biognosys, Zurich, Switzerland) was spiked into each sample prior to analysis by LC-MS/MS.

Peptides were separated using the nanoAcquity UPLC system (Waters) fitted with a trapping (nanoAcquity Symmetry C$_{18}$, 5 μm, 180 μm × 20 mm) and an analytical column (nanoAcquity BEH C$_{18}$, 1.7 μm, 75 μm × 250 mm). The outlet of the analytical column was coupled directly to Orbitrap Fusion Lumos (Thermo Fisher Scientific) using the Proxeon nanospray source. Solvent A was water, 0.1% formic acid, and solvent B was acetonitrile, 0.1% formic acid. 1 μg of samples was loaded with a constant flow of solvent A, at 5 μl/min onto the trapping column. Trapping time was 6 min. Peptides were eluted via the analytical column with a constant flow of 0.3 μl/min. During the elution step, the percentage of solvent B increased in a non-linear fashion from 0% to 40% in 90 min. Total runtime was 115 min, including clean-up and column re-equilibration. The peptides were introduced into the mass spectrometer via a Pico-Tip Emitter 360 μm OD × 20 μm ID; 10 μm tip (New Objective), and a spray voltage of 2.2 kV was applied. The capillary temperature was set at 300°C. The RF ion funnel was set to 30%. Data from pools of each condition/each sample were first acquired in DDA mode to contribute to a sample-specific spectral library. The conditions were as follows: Full scan MS spectra with mass range 350–1,500 *m/z* was acquired in profile mode in the Orbitrap with resolution of 60,000. The filling time was set at maximum of 50 ms with limitation of $2 \times 10^5$ ions. The "Top Speed" method was employed to take the maximum number of precursor ions (with an

intensity threshold of $5 \times 10^4$) from the full scan MS for fragmentation (using HCD collision energy, 30%) and quadrupole isolation (1.4 Da window) and measurement in the Orbitrap (resolution 15,000, fixed first mass 120 *m/z*), with a cycle time of 3 s. The MIPS (monoisotopic precursor selection) peptide algorithm was employed but with relaxed restrictions when too few precursors meeting the criteria were found. The fragmentation was performed after accumulation of $2 \times 10^5$ ions or after filling time of 22 ms for each precursor ion (whichever occurred first). MS/MS data were acquired in centroid mode. Only multiply charged ($2^+$–$7^+$) precursor ions were selected for MS/MS. Dynamic exclusion was employed with maximum retention period of 15 s and relative mass window of 10 ppm. Isotopes were excluded. In order to improve the mass accuracy, internal lock mass correction using a background ion (*m/z* 445.12003) was applied. For data acquisition and processing of the raw data, Xcalibur 4.0 (Thermo Scientific) and Tune version 2.1 were employed. For the data independent acquisition (DIA), the same gradient conditions were applied to the LC as for the DDA and the MS employed was a Thermo Q-Exactive HFX. The RF ion funnel was set to 40%. Full scan MS spectra with mass range 350–1,650 *m/z* was acquired in profile mode in the Orbitrap with resolution of 120,000. The default charge state was set to $3^+$. The filling time was set at maximum of 60 ms with limitation of $3 \times 10^6$ ions. DIA scans were acquired with 34 mass window segments of differing widths across the MS1 mass range. HCD fragmentation (stepped normalized collision energy; 25.5%, 27%, 30%) was applied, and MS/MS spectra were acquired with a resolution of 30,000 with a fixed first mass of 200 *m/z* after accumulation of $3 \times 10^6$ ions or after filling time of 40ms (whichever occurred first). Data were acquired in profile mode.

## MS data analysis

For library creation, the DDA data were searched using MaxQuant (version 1.5.3.28; Martinsried, Germany). The data were searched against a species-specific (*Mus musculus*) UniProt database with a list of common contaminants appended, as well as the HRM peptide sequences. The data were searched with the following modifications: Carbamidomethyl (C) (Fixed) and Oxidation (M)/ Acetyl (Protein N-term; Variable). The mass error tolerance for the full scan MS and MS/MS spectra was set at 20 ppm. A maximum of 1 missed cleavage was allowed. The identifications were filtered to satisfy FDR of 1% on peptide and protein level. A spectral library was created from the MaxQuant output of the DDA runs using Spectronaut (v. 10, Biognosys AG). This library contained 84,835 precursors, corresponding to 5,078 protein groups using Spectronaut protein inference. DIA data were then uploaded and searched against this spectral library in Spectronaut. Relative quantification was performed in the software for each pairwise comparison using the replicates from each condition. The data (candidate table) and data reports were then exported as tables, and further data analysis and visualization were performed with R-studio (version 0.99.902).

## Extraction and analysis of triacylglycerols

Lipids were extracted from ES cells, embryoids and planarians in PBS pH 7.4 by adding methanol, chloroform, and saline (final solvent ratio: 14:34:35:17; Koeberle *et al*, 2010, 2012). The organic phase was evaporated, and the lipid film dissolved and diluted in methanol. 1,2-Dimyristoyl-*sn*-glycero-3-phosphatidylcholine (DMPC) was used as internal standard. To calculate total lipid contents, we normalized lipid signals on the internal standard DMPC and cell numbers (EBs) or protein content (planarians).

Triacylglycerols were separated on an Acquity™ UPLC system (Waters) equipped with a BEH C8 column (1.7 μm, 2.1 × 100 mm, Waters) and detected by a QTRAP 5500 mass spectrometer (Sciex) with an electrospray ionization source as described before (Koeberle *et al*, 2013, 2015). The analysis of triacylglycerols was conducted in the positive ion mode. Acquired mass spectra were processed using Analyst 1.6 (Sciex).

## Electron microscopy

EBs were fixed by covering with Karnovsky fixative (2% paraformaldehyde, 3% glutaraldehyde in 0.1 M Cacodylate buffer, pH 7.3) for 8 h at room temperature. For a better handling during processing, EBs were embedded in 3% Agar. After solidification, small pieces were cut with a razor blade and carried over in a tissue processor (Leica) for all following steps. The blocks were rinsed five times with Cacodylate buffer for 15 min each. Secondary fixation was then carried out with 1% osmium tetroxide + 1% potassium hexacyanidoferrate(II) in 0.1 M Cacodylate buffer at 4°C for 2 h. They were then rinsed twice for 15 min in Cacodylate buffer followed by four times wash in distilled water each for 15 min. Blocks were then dehydrated in a series of acetone with increasing concentrations: 30, 50, 70, 90, 95, and 3 × 100%, each for 30 min. Additional en bloc staining with 1% uranyl acetate in 50% acetone was carried out. Samples were infiltrated by treating them in a mixture of acetone/resin: 3:1, 1:1, 1:3, for 45 min each and treating them thrice with epoxy resin "Epon"(glycid ether 100, SERVA) and 1 × Epon + accelerator (BDMA). Samples were then embedded in flat molds followed by polymerization for 48 h at 60°C. Samples were then trimmed using a Reichert UltraTrim (Leica). Semithin sections of 0.5 μm were stained with Azure staining (Richardson *et al*, 1960) to ensure enough cell material was encountered. Ultrathin sections of 55 nm without post-staining were placed onto copper slot grids coated with a Formvar/Carbon layer. All sections were made with an ultramicrotome "Reichert Ultracut S"(Leica). The images were taken using a transmission electron microscope JEM 1400 (JEOL) with an acceleration voltage of 80 kV and a CCD camera "Orius SC 1000A" (GATAN) analyzed using GATAN MICROSCOPY SUITE software (Version: 2.31.734.0).

## Planarian husbandry

Asexual biotype planarians belonging to the species *Schmidtea mediterranea* were used in this study. They were maintained in MilliQ water containing 1.6 mM NaCl, 1 mM $CaCl_2$, 1 mM $MgSO_4$, 0.1 mM $MgCl_2$, 0.1 mM KCl, and 0.1 g $NaHCO_3$ per liter of water at 19°C with 12-h light and 12-h dark cycles.

## RNAi experiments in planarians

Templates with T7 promoters appended to both strands were generated. Double-stranded RNA (dsRNA) was synthesized by using the *in vitro* transcription MEGAscript RNAi kit (Ambion). dsRNA was

injected ventrally into the planarian following the conventional schedule of three consecutive days of injections followed by another round of injections for three consecutive days twelve days later of the first round of injections as previously described (González-Estévez *et al*, 2012). As a control, dsRNA against *gfp* was used as a sequence not contained in the planarian genome. Planarians of 5 to 5.5 mm length at 7 days of starvation were used for the experiments. Details on the RNAi schedules are in the "Results" section.

## Planarian RNA extraction

Five worms or sorted irradiation sensitive cells (X1) for each condition per biological replicate were homogenized in 500 μl TRIzol reagent (Invitrogen) or TRIzol LS (Invitrogen), respectively, with an autoclaved pestle and left for 5 min at room temperature. Hundred μl of chloroform was added and vortexed for 15 s and left for 15 min at room temperature. It was then centrifuged at 14,000 rpm for 15 min at 4°C. The upper aqueous layer containing the total RNA was collected and resuspended in 250 μl of isopropanol. After incubation at room temperature for 10 min, the RNA-isopropanol mix was centrifuged at 14,000 rpm for 15 min at 4°C. The RNA was then washed in 500 μl of 70% ethanol by vortex for 30 s. Then, it was then centrifuged at 14,000 rpm for 15 min at 4°C. The supernatant was then discarded, and pellet containing the RNA was resuspended in 30 μl of nuclease-free water.

## Immunostaining in planarians

Whole-mount immunohistochemistry was performed as previously described (Cebria & Newmark, 2005). The following primary antibodies were used: anti-VC-1 (1:15,000; kindly provided by Professor K. Watanabe and H. Orii; Sakai *et al*, 2000); 3C11 (or anti-SYNAPSIN; 1:25, Developmental Studies Hybridoma Bank; Cebria, 2008) and Sánchez Alvarado, 2002) and Anti-Histone H3 phosphorylated at serine 10 (1:500; sc-8656-R from Santa Cruz).

## Planarian imaging and quantification

Live planarians were observed with Nikon stereomicroscope (SMZ745T). Each representative image for every condition was captured with Leica microscope camera (MC170 HD). Images for H3S10p immunostaining were taken with a Zeiss AXIO Zoom.V16 (Apotome 2) microscope equipped with Axiocam 506 mono and color and analyzed by AxioVison LE 4.8.2.0 software. Acquisition of images was followed by Apotome stacking. H3S10p quantification was conducted on whole planarians using Object Counter 3D plugin from Fiji. Areas were quantified, and images were processed using Fiji. Immunofluorescence images were acquired using a Zeiss LSM 710 ConfoCor 3.1 (Carl Zeiss) confocal laser scanning microscope. The images were analyzed using Fiji to obtain maximum projections from the stacks and were processed using Fiji.

## Phylogenetic analysis

Protein sequences were aligned with MUSCLE (v3.8.425) with standard settings. Alignment filtering and tree calculation were done using phyML (v3.0, via website http://www.phylogeny.fr/one_task.cgi?task_type=phyml, standard substitution model WAG,

Approximate Likelihood-Ratio Test (aLRT) SH-like). Protein sequences were extracted from the NCBI database with the specified accession numbers. Except the *Schistosoma* sequence which is a concatenation of entries XP_018653839.1 (5' fragment) and XP_018653838.1 (3' fragment). The *Schmidtea mediterranea* sequence was obtained by translation of the specified contig of the PlanMine transcriptome assembly dd_Smed_v6 (http://planmine.mpi-cbg.de/planmine/; Rozanski *et al*, 2019).

## RNA-sequencing (RNA-seq)

RNA integrity and quantification of total RNA was measured using the Agilent Bioanalyzer 2100. Library preparation for planarian (X1 cells) and mouse (EBs) samples were done using Illumina's TruSeq Stranded mRNA (X1 cells) and TruSeq RNA v2 library preparation kit (EBs), respectively, following the manufacturer's description. Quantification and quality check of libraries were done using the Agilent Bioanalyzer 2100 in combination with the DNA 7500 kit. Libraries were sequenced on a HiSeq2500 running in 51 cycle/single-end/high-output mode. The output per sample for X1 and EBs was on average 50mil (X1) and 46mil (EBs) reads, respectively. Sequence information was extracted in fastq format using Illumina's bcl2fastq v1.8.4 (X1) or bcl2FastQ v2.20.0.422 (EBs).

## Transcriptome expression analysis

RNA-seq was conducted on freshly isolated X1 cells (population containing somatic stem cells) from *exoc3(RNAi)*-treated and of *gfp (RNAi)*-treated planarians on day 38 of the injection protocol (Fig 2B, *n*= 3 biological replicates per group). The reads of all six samples were mapped with STAR (version 2.7.2b, parameters: --alignIntronMax 100000 --outSJfilterReads --outSAMmultNmax Unique --outFilterMismatchNoverLmax 0.04; Dobin *et al*, 2013) to the *S. mediterranea* S2F2 genome assembly (dd_Smes_g4) with the repeat filtered version of the smes_v2_SMEG gene annotation, both obtained from PlanMine (Rozanski *et al*, 2019). For each gene, reads that map uniquely to one genomic position were counted with FeatureCounts (version 1.6.5, multi-mapping or multi-overlapping reads were not counted, stranded mode was set to "–s 2"; Liao *et al*, 2014). The table of raw counts per gene per sample was analyzed with R (version 3.6.1) using package DESeq2 (version 1.26.0; Love *et al*, 2014) for differential expression. *exoc3(RNAi)*-treated and *gfp(RNAi)*-treated planarian samples were contrasted, assigning the latter the reference level. For each gene of the comparison, the *P*-value was calculated using the Wald significance test. The resulting *P*-values were adjusted for multiple testing with Benjamini–Hochberg correction. Genes with an adjusted *P*-value < 0.05 are considered differentially expressed (DEGs). The $\log_2$ fold changes (LFCs) were shrunk with lfcShrink(method = "normal") to control for variance of LFC estimates for genes with low read counts.

For the first mouse transcriptome analysis, RNA-seq data of $Tnfaip2^{-/-}$ EBs and WT EBs were analyzed (*n* = 3 biological replicates per group). The reads were quality-trimmed and filtered for low complexity with the tool preprocess from the SGA assembler (version 0.10.13, parameters -q 30 -m 50 –dust; Simpson & Durbin, 2012). The high-quality reads of all six samples were mapped to the mouse reference genome (GRCm38) with the Ensembl gene

annotation (release 90) with TopHat2 (version 2.1.0; Kim *et al*, 2013). For each annotated gene, reads that map uniquely to one genomic position were counted with FeatureCounts (version 1.5.0). The count table was analyzed with DESeq2 (version 1.16.1) in the same way as described in the planarian analysis. *Tnfaip2*$^{-/-}$ samples and WT samples were contrasted, assigning the latter the reference level. The second mouse transcriptome analysis was performed on *Tnfaip2*$^{-/-}$ EBs that have undergone *in vitro* differentiation for 3 days either in the absence or presence of PA/PC (*n* = 3 biological replicates per group). The reads of all six samples were mapped with STAR to the mouse reference genome (GRCm38) with the Ensembl gene annotation (release 99), counted with Feature-Counts, and analyzed with DESeq2 as described in the planarian analysis. *Tnfaip2*$^{-/-}$ samples in presence of PA/PC and *Tnfaip2*$^{-/-}$ samples in absence of PA/PC were contrasted, assigning the latter the reference level.

Gene lists of Gene Ontology (GO) terms were obtained by using AmiGO 2 (data base version 2019-07-02; Carbon *et al*, 2009) with the taxon filter "taxon_subset_closure_label: Mus musculus" and the filter for the respective GO identifier (GO:ID) "regulates_closure:GO:ID", while the following GO:IDs were queried. GO:0007498 (Mesoderm Development, *n* = 118), GO:0007398 (Ectoderm Development, *n* = 17), GO:1902459 (Positive Regulation of Stem Cell Population Maintenance, *n* = 14), and GO:1902455 (Negative Regulation of Stem Cell Population Maintenance, *n* = 7). For GO:1902459, we added *Tfcp2l1*, *Dppa3*, *Fbxo15*, *Zfp42*, and *Klf2* as these ancillary pluripotency factors were missing in the database which are critical to maintain pluripotent state by counteracting pro-differentiation influences (Hackett & Surani, 2014). We also added *Tnfaip2* to GO:1902455 as this study identified *Tnfaip2* as the negative regulator of pluripotency. Significance of non-random overlap between DEGs and genes of a GO term was calculated with the hypergeometric test (R function "phyper") using all measured genes as the background (*n* = 27,364). All heat maps were calculated on size factor-normalized read counts followed by gene-wise normalization (i.e., row *z*-score). Dendrograms at selected heat maps visualize hierarchical clustering of the genes.

### Overlap calculation to planarian somatic stem cell gene sets

To find overlaps between DEGs of this study to published stem cell gene sets, an identifier (ID) mapping was necessary as each gene set is based on different transcriptome assemblies. Solana *et al*, 2012, used the Sm454ESTABI dataset (25,052 sequences, https://doi.org/10.1371/journal.pone.0015617.s004), Zhu used the SmedAsxl_20130210 dataset (81,283 sequences, https://www.ncbi.nlm.nih.gov/Traces/wgs/GCZZ01), Plass used the dd_Smed_v6 dataset (41,745 sequences, http://planmine.mpi-cbg.de/planmine/model/bulkdata/dd_Smed_v6.pcf.contigs.fasta.zip), and Fincher used the dd_Smed_v4 dataset (38,633 sequences, https://idisk.mpi-cbg.de/~brandl/planaria/dd_Smed_v4.nuc.fasta.zip). Each transcriptome was aligned with GMAP (version 2019-09-12, --chimera-margin = 200  --min-trimmed-coverage = 0.9  --min-identity = 0.9; Wu & Watanabe, 2005) to the *S. mediterranea* S2F2 genome assembly (dd_Smes_g4). For each transcriptome, positions where a transcript overlaps with the smes_v2_SMEG gene annotation were identified with bedtools intersect (version v2.28.0, -r -f 0.50 -wa –

wb; Quinlan & Hall, 2010). In the case of dd_Smed_v6, the isoform appendix was removed from the transcript ID.

Five established post-mitotic progeny genes and 32 novel epithelial progenitor markers from Zhu *et al*, 2015; 823 stem cell markers from Solana *et al*, 2012; 199 genes of the cluster "Neoblast1" from Plass *et al*, 2018; 219 genes of the "smedwi-1 high" category (clusters 0, 3, 7 and 8) and 3,974 genes of all clusters from the *smedwi-1*$^{+}$ table from Fincher *et al*, 2018, were extracted to find overlaps with 6,339 DEGs from this analysis. Possible duplicated gene assignments were discarded. Significance of non-random overlap was calculated with the hypergeometric test (R function "phyper") using the number of genes that were actually assigned to the respective transcriptome as the background.

### FACS analysis of planarian X1 population

Planarians were cut into small pieces on ice and dissociated with trypsin as previously described (Hayashi *et al*, 2006). Cells were then filtered and stained with 20 µg/ml Hoechst 33342 (BD Biosciences) and 0.5 µg/ml Calcein-AM (BD Biosciences). Cells were then sorted using BD FACSAria III. The FACS analysis provided the percentage X1, X2, Xins, and other population of live cells. The percentage of X1 population among X1, X2, and Xins was calculated by dividing the percentage of the X1 population with the cumulative percentage of X1, X2, and Xins population and then multiplying by 100.

### Treatment with palmitic acid and palmitoyl-L-carnitine

Palmitic acid and palmitoyl-L-carnitine were used at a concentration that has been reported to induce differentiation of ES cells (Yanes *et al*, 2010). Briefly, ES cells were cultured in MEF medium supplemented with 8 µM each of palmitic acid (cat. no. P5585; Sigma) and palmitoyl-L-carnitine (cat. no. 61251 Sigma) at 37°C (Eppendorf, New Brunswick Galaxy 170R) for 3 days to form EBs. The EBs (at day 0) were plated onto 0.1% gelatin-coated tissue culture dishes to initiate *in vitro* differentiation by incubating for another 3 days in MEF medium supplemented with 8 µM each of palmitic acid and palmitoyl-L-carnitine at 37°C (Eppendorf, New Brunswick Galaxy 170R).

Planarians were injected with freshly prepared BSA-conjugated palmitic acid (PA) and palmitoyl-L-carnitine (PC; 750 µM, five pulse of 32 nl each per worm) 3 h prior to every RNAi treatment.

### Statistical tests

All statistical analyses were performed using GraphPad Prism 8 or R (R Core Team, 2019). (Student) *t*-tests were done as two-sided, unpaired versions unless otherwise defined. Binomial tests were conducted one-sided with the R function binom.test with the alternative hypothesis set to "greater". *P*-values are shown four digits after decimal.

## Data availability

The shRNA sequencing data discussed in this publication have been deposited in NCBI's Sequence Read Archive and are accessible

through SRA accession number PRJNA637081 (https://www.ncbi. nlm.nih.gov/sra/PRJNA637081). The RNA-seq data discussed in this publication have been deposited in NCBI's Gene Expression Omnibus (Edgar *et al*, 2002) and are accessible through GEO Series accession number GSE134302 (https://www.ncbi.nlm.nih.gov/geo/ query/acc.cgi?acc = GSE134302). The mass spectrometry proteomics data are available via ProteomeXchange with identifier PXD014628 (http://proteomecentral.proteomexchange.org/cgi/Ge tDataset?ID = PXD014628). GenBank accession number for *Smed-exoc3* cDNA sequence is MN996227.

**Expanded View** for this article is available online.

## Acknowledgements

This work is funded by the German Research Foundation (DFG) within the CRC-"PolyTarget" (Project B01) and by the European Union (Advance ERC grant to K. L. Rudolph : 323136). We would like to acknowledge the FLI Core Facilities of DNA Sequencing, Flow Cytometry and Imaging. We thank Prof. Lars Zender for the cooperation in providing the lentiviral shRNA library targeting putative cancer-related genes. We also thank Dr. Francesco Neri for providing ES cells. Anti-SYNORF antibody was obtained from the Developmental Studies Hybridoma Bank, University of Iowa. We are grateful to Prof. K. Watanabe and H. Orii for providing the VC1 antibody. We acknowledge the Leibniz Graduate School on Ageing and Age-Related Disease (LGSA).

## Author contributions

SD and KLR conceptualized and designed the study. SD, DAF, MKD, KB, MS, MG, JK, AGo, PR acquired the data. SD, DAF, PK, MKD, KS, JK, AGo, AGr, AK, SP, CGE and KLR analyzed and interpreted the data. SD and KLR wrote the manuscript. All authors read and approved the final manuscript.

## Conflict of interest

The authors declare that they have no conflict of interest.

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
