## [Review Process File · EMBO Reports]

Tnfaip2/Exoc3 driven lipid metabolism is essential for stem cell differentiation & organ homeostasis

Sarmistha Deb, Daniel Felix, Philipp Koch, Maharshi Krishna Deb, Karol Szafranski, Katrin Buder, Mara Sannai, Marco Groth, Joanna Kirkpatrick, Stefan Pietsch, André Gollowitzer, Alexander Groß, Philip Riemenschneider, Andreas Koeberle, Cristina González-Estévez, and Karl-Lenhard Rudolph

DOI: 10.15252/embr.201949328

Corresponding author(s): Karl-Lenhard Rudolph (lenhard.rudolph@leibniz-fli.de)

Review Timeline:

Submission Date:	22nd Sep 19
Editorial Decision:	26th Nov 19
Revision Received:	28th Jun 20
Editorial Decision:	14th Aug 20
Revision Received:	21st Oct 20
Accepted:	29th Oct 20

Editor: Esther Schnapp

Transaction Report:

Dear Lenhard,

Thank you for your patience while your manuscript was peer-reviewed at EMBO reports. I sent your study to 3 referees and so far we have received reports from two of them, which I paste below. Given that both referees are in fair agreement that you should be given a chance to revise the manuscript, I would like to ask you to begin revising your study along the lines suggested by the referees. Please note that this is a preliminary decision made in the interest of time, and that it is subject to change should the third referee offer very strong and convincing reasons for this. As soon as we will receive the final report on your manuscript, we will forward it to you as well.

I think all comments in the reports we have make sense and should be addressed. Please let me know if you disagree and we can discuss this further. Please address all referee concerns in a complete point-by-point response. Acceptance of the manuscript will depend on a positive outcome of a second round of review. It is EMBO reports policy to allow a single round of major revision only and acceptance or rejection of the manuscript will therefore depend on the completeness of your responses included in the next version of the manuscript.

Revised manuscripts should be submitted within three months of a request for revision; they will otherwise be treated as new submissions. Please contact us if a 3-months time frame is not sufficient for the revisions so that we can discuss this further. Given the 6 main figures, I suggest that you layout the manuscript as a full article.

Regarding data quantification, please specify the number "n" for how many independent experiments were performed, the bars and error bars (e.g. SEM, SD) and the test used to calculate p-values in the respective figure legends. This information must be provided in the figure legends. Please also include scale bars in all microscopy images.

IMPORTANT NOTE: we perform an initial quality control of all revised manuscripts before re-review. Your manuscript will FAIL this control and the handling will be DELAYED if the following APPLIES:
1) A data availability section providing access to data deposited in public databases is missing
2) Your manuscript contains statistics and error bars based on n=2 or on technical replicates.
Please use scatter blots in these cases. No error bars can be calculated if n=2.

3) We replaced Supplementary Information with Expanded View (EV) Figures and Tables that are collapsible/expandable online. A maximum of 5 EV Figures can be typeset. EV Figures should be cited as 'Figure EV1, Figure EV2" etc... in the text and their respective legends should be included in the main text after the legends of regular figures.

5) a complete author checklist, which you can download from our author guidelines <<https://www.embopress.org/page/journal/14693178/authorguide>>. Please insert information in the checklist that is also reflected in the manuscript. The completed author checklist will also be part of the RPF.

6) Please note that all corresponding authors are required to supply an ORCID ID for their name upon submission of a revised manuscript (<<https://orcid.org/>>). Please find instructions on how to link your ORCID ID to your account in our manuscript tracking system in our Author guidelines <<https://www.embopress.org/page/journal/14693178/authorguide#authorshipguidelines>>

8) We would also encourage you to include the source data for figure panels that show essential data. Numerical data should be provided as individual .xls or .csv files (including a tab describing the data). For blots or microscopy, uncropped images should be submitted (using a zip archive if multiple images need to be supplied for one panel). Additional information on source data and instruction on how to label the files are available at <<https://www.embopress.org/page/journal/14693178/authorguide#sourcedata>>.

9) Our journal also encourages inclusion of *data citations in the reference list* to directly cite datasets that were re-used and obtained from public databases. Data citations in the article text are distinct from normal bibliographical citations and should directly link to the database records from which the data can be accessed. In the main text, data citations are formatted as follows: "Data ref: Smith et al, 2001" or "Data ref: NCBI Sequence Read Archive PRJNA342805, 2017". In the Reference list, data citations must be labeled with "[DATASET]". A data reference must provide the database name, accession number/identifiers and a resolvable link to the landing page from which the data can be accessed at the end of the reference. Further instructions are available at

<https://www.embopress.org/page/journal/14693178/authorguide#referencesformat>

I look forward to seeing a revised version of your manuscript when it is ready. Please let me know if you have questions or comments regarding the revision.

Kind regards,
Esther

Referee #1:

In this manuscript, Deb and colleagues address the function of the Exocyst component Tnfaip2/Exoc3 in stem cell reprogramming and differentiation. Their findings include that the knock-down of TNFAIP in culture mouse cells increases fibroblast reprogramming efficiency and inhibits ES cell differentiation. These phenotypes correlate with reduced lipid droplet formation and the supplementation of storage lipid precursors rescues differentiation-associated Tnfaip loss of function phenotypes.

In parallel, the authors examine the function of the planarian Exoc3 gene orthologue and report an apparent requirement in stem cell differentiation, which can also be rescued by FA supplementation. On basis of these findings, the authors conclude that they have identified an evolutionarily conserved pathway with important roles in stem cell differentiation and tissue regeneration.

I find the findings of the manuscript genuinely interesting, also for the broad readership of EMBO reports. The probing of deep evolutionary conservation of gene function via functional investigation of the planarian gene homologue adds a further interest dimension to this manuscript. However, some of the key conclusions are insufficiently supported by data. I am not an expert on vertebrate stem cell cultures, hence the predominant focus of my comments on the planarian experiments.

Major points

1) Tnfaip2 gene function in mouse: The broad involvement of the exocyst protein complex in many cellular functions requires additional evidence that the observed phenotypes are indeed all

associated with lipid metabolism. The authors should present RNAseq data to support that the apparent FA-induced rescue of differentiation in Fig 6B-C is reflected in the rescue of the differentiation-associated gene expression profiles in Fig. 4A. In parallel, the effect of the FA supplementation on reprogramming efficiency could be examined.

2) The authors conclude deep evolutionary conservation of Tnfaip/Exoc3 gene function by inferring defects in stem cell differentiation in mouse stem cells and planarians. However, the planarian data in Fig 3 insufficiently supports a differentiation defect in planarians. Criticism: An increase in H3P-positive cells alone could also reflect a mitotic arrest or an optical artefact resulting from the projection of an unchanged or even decreased density of mitotic stem cells onto the much reduced area of the strongly contracted animals (Fig. 3B); The phenotypes shown in Fig. 3E-G are either broadly associated with general defects in the planarian stem cell system (E, F) or not convincing (G- this image does not support a protonephridial defect); the supplemental RNAseq data is limited to a small number of inconclusive and obviously hand-picked genes (see below for detailed points). Therefore, additional data is required to support the conclusion that Exoc3 is required specifically for stem cell differentiation in planarians. Minimally, this requires the demonstration that stem cells accumulate at the expense of differentiating progenitors. Time course experiments would be particularly useful for providing a clearer view of direct versus indirect effects. If the authors cannot provide such data, the text needs to be generalized to state that Smed-exoc3 is required for stem cell driven tissue homeostasis in planarians, which might include effects on differentiation.

3) The authors conclude that Exoc3 in both systems affects stem cell differentiation via the modulation of lipid stores. However, the FA injection rescue (see below for additional criticism) provides the only tentative support for the link between Exoc3 and lipid storage in planarians. The authors therefore need to directly examine the effect of planarian Exoc3(RNAi) on planarian lipid storage. This isn't overly difficult, as protocols for lipid droplet visualization in planarians have been published or the authors could use their own mass spec assay in Fig. 5E to quantify storage lipid levels in control versus Exoc3(RNAi) animals. Such data are strategically important also as additional support for the validity of the rescue experiment.

4) The abstract and text claim the discovery of a new pathway. However, the majority of the data pertains to a single gene (Tnfaip2/Exoc3), the mechanistic connection to lipid droplets remains tentative and as stated in the text, lipid droplets have already been implicated in stem cell differentiation. The "new pathway" terminology is therefore a bit of an overstatement and the respective passages should be rephrased accordingly. Further, the text of the current manuscript is not in a submission-ready state and requires substantial editing (see below).

Minor points

Manuscript:

5) Multiple cut/paste scars persist and some sections are overly convoluted and hard to understand- please polish the text into a submission-ready state and edit for grammar mistakes.

6) The planarian data is poorly integrated, especially in the discussion. Please include some form of comparison between the two experimental systems used in the manuscript, the additional insights that the multi-pronged approach permits and some of the caveats regarding the interpretation of the planarian data (e.g., direct effects in stem cells (as implied by the text) vs. indirect effects of the organism-wide knock-down in other tissues (e.g., in the lipid storing intestine)).

Rescue assay:

7) The complete restoration of progeny cell marker levels after FA injection is remarkable. However, the low number of 3/19 animals that regenerated properly raises concerns in conjunction with the apparently small sample size (5?) for the qPCR assays. If the qPCR samples were to have been biased for "good looking" animals, the apparent rescue might actually be due to RNAi escapers. Please state explicitly how the samples for qPCR were selected and/or add additional experimental data points to alleviate such concern- this point is crucial in conjunction with the direct demonstration of Exoc3 effects on planarian lipid storage (point 3).

RNAseq analysis:

8) S Fig.3b generates the visual impression of a large number of analyzed genes that all follow the same pattern. However, this is NOT the case, as only 5 genes are analyzed. Please filter for isoforms of the same transcript (_1,...2,...3,...4 suffix- see PlanMine user manual) and display only 1 isoform per gene (common practice in the field is the use of the longest isoform).

9) In the case of "H2B" the authors likely analyze two separate genes (different transcript ID; see PlanMine user manual). Please state which of the two gene sequences corresponds to the isoform that was functionally validated in the previous RNAseq isoforms.

10) Please include a range of bona-fide stem cell markers into the analysis/figure, i.e., piwi-1, -2, pcna, bruli...; the more, the better). Rather than hand-picking a few genes, the point that stem cell markers are generally up and differentiation markers are generally down would be most strongly supported by querying one of the published global stem cell/progeny gene sets in this experiment.

Terminology:

11) Please do not use "endo/meso/ectoderm" in reference to planarian stem cell lineages- the implied homology has not been established.

12) "Smed-exoc-3(RNAi) displayed a good knock down efficiency"- please re-phrase with proper scientific terminology.

Referee #3:

In this manuscript, the authors report that lipid metabolism mediated by Tnfaip2/Exoc3 plays the key role for stem cell differentiation and organ maintenance. Knockdown of Tnfaip2 was found to promote the iPSCs generation, however, suppression of Tnfaip2 or its planarian orthologue, Smed-exoc3, in stem cells abrogated differentiation and regeneration. Mechanistically, Tnfaip2 deficiency led to deregulation of TAG synthesis and reduced the expression of Vimentin that is responsible for lipid droplets formation. While the major findings in this study are potentially interesting, the author did not provide sufficient evidence to support the conclusion.

1. shTnfaip2s were found to enhance the somatic reprogramming process, however, the author also observed that Tnfaip2-deficient ESCs exhibited differentiation failure. It is not clear that Tnfaip2 functions to prevent or promote the acquisition and maintenance of pluripotency. The author should examine the expression level of Tnfaip2 change during somatic cell reprogramming and ESCs differentiation.
2. In this study, Tnfaip2 seemed to play the conflicting roles in somatic reprogramming and stem cell differentiation. It would be interesting to investigate whether the iPSCs that generated from MEFs with OSKM and shTnfaip2 induction exhibit differentiation deficiency.
3. As shown in Figure 6, the supplementation of palmitic acid and palmitoyl-L-carnitine rescued the

differentiation defects of *Tnfaip2/exoc3*-deficient pluripotent stem cells in culture and in vivo in planarian. The authors concluded that Vim-dependent lipid droplet (LD) formation mediated by *Tnfaip2/exoc3* was critical for ESC differentiation. However, they did not provide sufficient evidence to support the conclusion. Since Vimentin also is a regulator of EMT process that is well-known to be critical for somatic reprogramming or stem cell differentiation, the author may wish to go further to prove that Vimentin deficiency lead to failure in lipid droplet formation and then ESCs differentiation.

4. The authors claimed *Tnfaip2* regulated the expression of Vimentin and *Cpt1a*. It's better to provide the results of qRT-PCR and western blot besides the proteomics data.

5. Another critical missing part is how *Tnfaip2/exoc3* regulate the protein levels of Vimentin or *Cpt1a*. In Supplementary Figure 6E, the author claimed that there was no additive effect in the enhancement of reprogramming efficiency upon co-depletion of *Tnfaip2* and Vim indicating that *Tnfaip2* and Vim act especially in this regard. It is not surprising that knockdown of Vimentin, the regulator of EMT, significantly increased the somatic reprogramming efficiency. The author may wish to provide more evidence to claim Vimentin is the downstream effector of *Tnfaip2*. It is not clear whether Vimentin could rescue the differentiation defect of ESCs or somatic stem cells with *Tnfaip2/exoc3*.

6. In Figure 3C&D, *Smed-exoc3* knockdown increased the numbers of proliferating stem cells in planarian. The authors explained that *Smed-exoc3* knockdown led to the self-renewal versus differentiation of stem cells. Nevertheless, it is known that the asymmetrical and symmetrical division is essential for stem cell maintenance and tissue regeneration. The authors may wish to address whether stem cells with *Smed-exoc3* knockdown prefer symmetrical division rather than asymmetrical division and whether the lipid drop formation affects the division pattern.

Referee #1:

In this manuscript, Deb and colleagues address the function of the Exocyst component *Tnfaip2/Exoc3* in stem cell reprogramming and differentiation. Their findings include that the knock-down of TNFAIP in culture mouse cells increases fibroblast reprogramming efficiency and inhibits ES cell differentiation. These phenotypes correlate with reduced lipid droplet formation and the supplementation of storage lipid precursors rescues differentiation-associated *Tnfaip* loss of function phenotypes.

In parallel, the authors examine the function of the planarian *Exoc3* gene orthologue and report an apparent requirement in stem cell differentiation, which can also be rescued by FA supplementation. On basis of these findings, the authors conclude that they have identified an evolutionarily conserved pathway with important roles in stem cell differentiation and tissue regeneration.

I find the findings of the manuscript genuinely interesting, also for the broad readership of EMBO reports. The probing of deep evolutionary conservation of gene function via functional investigation of the planarian gene homologue adds a further interest dimension to this manuscript. However, some of the key conclusions are insufficiently supported by data. I am not an expert on vertebrate stem cell cultures, hence the predominant focus of my comments on the planarian experiments.

Major points

1) *Tnfaip2* gene function in mouse: The broad involvement of the exocyst protein complex in many cellular functions requires additional evidence that the observed phenotypes are indeed all associated with lipid metabolism. The authors should present RNAseq data to support that the apparent FA-induced rescue of differentiation in Fig 6B-C is reflected in the rescue of the differentiation-associated gene expression profiles in Fig. 4A. In parallel, the effect of the FA supplementation on reprogramming efficiency could be examined.

Response: We thank the reviewer for his/her positive comments on our manuscript and for the valuable comments, which we all addressed in the revised manuscript and which significantly improved our study.

We followed the reviewer's suggestion and included a new RNA-sequencing analysis on the rescue of differentiation of *Tnfaip2*^{-/-} EBs by fatty acid (FA) treatment. The analysis shows that FA treatment rescues the expression of stemness/differentiation regulating genes (new Figure 3G; new Expanded View 6A and 6B and new Appendix Table 10) that are dysregulated in *Tnfaip2*^{-/-} EBs versus WT EBs (revised Figure 2E- former Figure 4A and revised Expanded View 4A and 4B – former Supplementary Figure 5B and 5C). We highlight the differentially regulated stemness/differentiation genes that were differentially expressed in *Tnfaip2*^{-/-} EBs versus WT EBs but rescued in *Tnfaip2*^{-/-} EBs by FA-treatment (marked with asterisk in new Figure 3G and new Expanded View 6A and 6B (1st paragraph on page 19 of the revise manuscript).

Remark: Gene lists of corresponding Gene Ontology (GO) terms (e.g. “Positive and Negative Regulators of stem cell population maintenance” as well as “Ectodermal and Mesodermal Markers”) were obtained by using AmiGO 2 (data base version 2019-07-02; Carbon et al.,

2009). The corresponding figures show the heatmaps of all differentially expressed genes in the respective experiments: Each of the Figures 2E and 3G shows 19 plus 5 genes, but there are differences in the list of genes based on which genes of the list of indicated AmiGO terms were differentially expressed in the respective experiments. Same explanation holds true for the depiction of genes in Expanded View 4A and 4B and new Expanded View Figure 6A and 6B (1st paragraph on page 19 of the result section).

We also addressed the role of FA in reprogramming. Specifically, we show that supplementation of PA/PC during cellular reprogramming inhibits generation of iPSCs, which is in line with pro-differentiation influence of PA/PC. Similarly, we now demonstrated that treatment with Etomoxir, a CPT1A-specific inhibitor, improves reprogramming efficiency (new Figure 4F and 4G) (2nd paragraph on page 21 of the result section).

2) The authors conclude deep evolutionary conservation of Tnfaip/Exoc3 gene function by inferring defects in stem cell differentiation in mouse stem cells and planarians. However, the planarian data in Fig 3 insufficiently supports a differentiation defect in planarians. Criticism: An increase in H3P-positive cells alone could also reflect a mitotic arrest or an optical artefact resulting from the projection of an unchanged or even decreased density of mitotic stem cells onto the much reduced area of the strongly contracted animals (Fig. 3B); The phenotypes shown in Fig. 3E-G are either broadly associated with general defects in the planarian stem cell system (E, F) or not convincing (G- this image does not support a protonephridial defect); the supplemental RNAseq data is limited to a small number of inconclusive and obviously hand-picked genes (see below for detailed points). Therefore, additional data is required to support the conclusion that Exoc3 is required specifically for stem cell differentiation in planarians. Minimally, this requires the demonstration that stem cells accumulate at the expense of differentiating progenitors. Time course experiments would be particularly useful for providing a clearer view of direct versus indirect effects. If the authors cannot provide such data, the text needs to be generalized to state that *Smed-exoc3* is required for stem cell driven tissue homeostasis in planarians, which might include effects on differentiation.

Response: We thank the reviewer for his/her insightful comment. We followed the suggestion of this reviewer and have included a new time course experiment analyzing the number of stem cells (X1 population) in *Smed-exoc3* depleted planarians versus controls at different time points after starting the iRNA treatment (day-30, -34, and -38). In addition, we also monitored the expression of 5 stem cell marker genes in RNA extracted from total planarians at the same timepoints. The results show an increase in stem cell marker gene expression (new Expanded View 3A) and an increase in the number of X1 cells (new Expanded View 3B and 3C) in planarians that were injected with iRNA against *Smed-exoc3* versus planarians injected with a control iRNA against *gfp*. This increase was gradually developing over time (1st paragraph on page 11 of the result section). In addition, we also conducted new experiments to analyze whether the number of differentiated cells would decrease in *Smed-exoc3*-treated planarians versus control iRNA treated planarians. These experiments revealed a decrease in differentiated cells (new Expanded View 3B and 3D; page 12 of the result section) as well as a decline in the

expression of 2 differentiation markers in response to *Smed-exoc3* depleted planarians (new Appendix Figure 1C; 1st paragraph on page 10 of the result section).

The reviewer is right that a mitotic arrest could also lead to an increase in the X1 population. As X1 cells are mostly in G2/M and the number of X1 increases in *Smed-exoc3* depleted planarian vs. wildtype, it is difficult to employ cell cycle analysis to discriminate G2/M arrest from an increase in X1 neoblast cells. However, an increase G2/M arrested cells usually coincides with an increase in apoptosis. We included an analysis of apoptosis (Annexin V positive cell) in the X1 population, which did not reveal any evidence for an increase in apoptosis (new Appendix Figure 3A) (3rd paragraph on page 13 of the result section). Besides, the expression of the planarian homologue of mammalian *p53/p63* a bona fide mitotic checkpoint gene – was not found to be significantly decreased in our RNA-sequencing analysis of *exoc3(RNAi)*-treated planarians *versus* controls (page 14, first paragraph of the revised manuscript). Together, these data suggested that the increase in X1 population in *exoc3(RNAi)*-treated planarians reflected a differentiation failure rather than a mitotic arrest.

3) The authors conclude that Exoc3 in both systems affects stem cell differentiation via the modulation of lipid stores. However, the FA injection rescue (see below for additional criticism) provides the only tentative support for the link between Exoc3 and lipid storage in planarians. The authors therefore need to directly examine the effect of planarian Exoc3(RNAi) on planarian lipid storage. This isn't overly difficult, as protocols for lipid droplet visualization in planarians have been published or the authors could use their own mass spec assay in Fig. 5E to quantify storage lipid levels in control versus Exoc3(RNAi) animals. Such data are strategically important also as additional support for the validity of the rescue experiment.

Response: We thank the reviewer for this suggestion. We have taken a quantitative approach to measure the lipidomics profile of *exoc3(RNAi)* planarians compared to the control *gfp(RNAi)* counterparts (new Figure 5A). Like *Tnfaip2*^{-/-} EBs, *exoc3(RNAi)* planarians were found to exhibit profound loss of Triacylglycerides (TAG) in comparison to control counterpart (bottom of page 21 of the revise manuscript).

4) The abstract and text claim the discovery of a new pathway. However, the majority of the data pertains to a single gene (*Tnfaip2/Exoc3*), the mechanistic connection to lipid droplets remains tentative and as stated in the text, lipid droplets have already been implicated in stem cell differentiation. The "new pathway" terminology is therefore a bit of an overstatement and the respective passages should be rephrased accordingly. Further, the text of the current manuscript is not in a submission-ready state and requires substantial editing (see below).

Response: We thank the reviewer for this comment. Also, the other reviewers have asked us to delineate the *Exoc3/Tnfaip2 – Vim* pathway more clearly. We have done so and our new experiments indeed confirm that *Tnfaip2 - Vim* act epistatically in regulating lipid metabolism and stem cell maintenance and differentiation (see new Figure 4D-G, Appendix Figure 5A-C, see also responses to 3rd and 4th comments of reviewer 3 below). Based on these new data, we think that it is justified to describe these findings as a new pathway that regulates lipid droplet formation/lipid metabolism and stem cell differentiation (Page 19-21 of the result section).

Minor points

Manuscript:

5) Multiple cut/paste scars persist and some sections are overly convoluted and hard to understand- please polish the text into a submission-ready state and edit for grammar mistakes.

This has been corrected in the revised manuscript.

6) The planarian data is poorly integrated, especially in the discussion. Please include some form of comparison between the two experimental systems used in the manuscript, the additional insights that the multi-pronged approach permits and some of the caveats regarding the interpretation of the planarian data (e.g., direct effects in stem cells (as implied by the text) vs. indirect effects of the organism-wide knock-down in other tissues (e.g., in the lipid storing intestine)).

Response: We followed the reviewer's suggestion and have included following paragraph in the revised discussion: "Unlike in mouse or in other mammals, wherein pluripotent stem cells are restricted to early phases of embryonic development, pluripotent somatic stem cells are present in planarians throughout their life to ensure homeostasis and regeneration of all tissues of the worms. Thus, planarians provide a unique *in vivo* model system to study the function of genes and pathways in regulating pluripotent somatic stem cells *in vivo*. In vertebrates, organ regeneration and homeostasis are driven by organ specific, adult stem cells with restricted differentiation capacity, including multipotent, oligopotent, or unipotent stem cells. The current study shows that it is possible to use planarians to delineate conserved gene functions as well as gene specification that control the function of pluripotent, embryonic stem cells in culture as well as pluripotent, somatic stem cells at the organism level (in planarians). It seems promising to go on to use this approach to identify genetically controlled mechanisms that control the function of restricted, somatic stem cells in more complex organisms such as vertebrates and mammals." (bottom of page 24 of the revised manuscript).

Rescue assay:

7) The complete restoration of progeny cell marker levels after FA injection is remarkable. However, the low number of 3/19 animals that regenerated properly raises concerns in conjunction with the apparently small sample size (5?) for the qPCR assays. If the qPCR samples were to have been biased for "good looking" animals, the apparent rescue might actually be due to RNAi escapers. Please state explicitly how the samples for qPCR were selected and/or add additional experimental data points to alleviate such concern- this point is crucial in conjunction with the direct demonstration of Exoc3 effects on planarian lipid storage (point 3).

Response: We thank the reviewer for this suggestion. Initially the numbers in the figures depicted the number of planarians that retained the defective phenotype (indicating for example that 3/20 *exoc3(RNAi)*-treated planarians showed a defective phenotype instead of labeling it as 17/20 showing restored phenotype). We have corrected this labeling as suggested by the reviewer and also included additional animals for the analysis and hence now 20 out of 23

(20/23) *exoc3(RNAi)*-treated planarians exhibited restoration of differentiation potential. In addition, we have involved a second independent experimentalist to evaluate the number of animals showing the depicted phenotype. The corrected figures are included in the revised manuscript.

RNAseq analysis:

8) S Fig.3b generates the visual impression of a large number of analyzed genes that all follow the same pattern. However, this is NOT the case, as only 5 genes are analyzed. Please filter for isoforms of the same transcript (_1,...2,...3,...4 suffix- see PlanMine user manual) and display only 1 isoform per gene (common practice in the field is the use of the longest isoform).

9) In the case of "H2B" the authors likely analyze two separate genes (different transcript ID; see PlanMine usermanual). Please state which of the two gene sequences corresponds to the isoform that was functionally validated in the previous RNAseq isoforms.

10) Please include a range of bona-fide stem cell markers into the analysis/figure, i.,e., *piwi-1*, -2, *pcna*, *bruli*...; the more, the better). Rather than hand-picking a few genes, the point that stem cell markers are generally up and differentiation markers are generally down would be most strongly supported by querying one of the published global stem cell/progeny gene sets in this experiment.

Response: We thank the reviewer for the advice on improving the depiction of our RNA-sequencing analysis of X1 cells from *Smed-exoc3* depleted planarians and controls (*gfp-RNAi* treated).

We now provide a completely unbiased analysis on the enrichment of stem cell and progenitor cell related genes in our data set. Specifically, we compared DEGs of *Smed-exoc3* depleted X1 versus control treated (*gfp-RNAi*) planarians from our RNA-sequencing experiment with published data sets of stem cells from 3 different papers: Solana et al., 2012; Plass et al., 2018, Fincher et al., 2018 (new Appendix Figure 1D and 1E and Appendix Table 4 and 6; page 11 of the revised manuscript). The analysis shows a significant overlap of DEGs in X1 cells of *Smed-exoc3* depleted planarians vs. controls with the published neoblast gene expression signatures indicating that depletion *Smed-exoc3* alters the neoblast transcriptome signature in planarians.

In addition, there was a significant overlap of our DEGs in X1 cells of *Smed-exoc3* depleted planarians vs. controls with published gene expression signatures of distinct sub-clusters of *smedwi-1+* cells (Fincher et al.) and in fully differentiated Xins cells (Zhu et al. 2015) (see Appendix Figure 1D and E and page 12 of the revised manuscript). These data support the interpretation that *exoc3(RNAi)* disturbs the differentiation of neoblast cells.

Terminology:

11) Please do not use "endo/meso/ectoderm" in reference to planarian stem cell lineages- the implied homology has not been established.

Review: As per the suggestion, we have removed these terms throughout the text and from the relevant figures (revised Figure 2A and 2B – former Figure 3F and 3G; revised Appendix Figure 2D and 2E – former Supplementary Figure 4D and 4E).

12) "*Smed-exoc-3(RNAi)* displayed a good knock down efficiency"- please re-phrase with proper scientific terminology.

Response: We changed the text as follows “RT-qPCR analysis of total (whole body derived) RNA from *exoc3(RNAi)*-treated planarians compared to controls revealed a good knockdown of *Smed-exoc3*” (1st paragraph on page 9 of the result section).

Referee #2:

The Manuscript by Deb et al entitled "Tnfaip2/Exoc3 driven lipid metabolism is essential for stem cell differentiation & organ homeostasis" involves an RNAi screen that aims at identifying genes that impair the generation of mouse induced pluripotent stem (iPS) cells, revealing Tnfaip2 to enhance formation of iPS cells. Tnfaip2 is a target of inflammatory TNF α and NF κ B pathways. To study the function of the gene in a pluripotency context, a knockdown of the planarian Tnfaip2 orthologue *exoc3* has been performed. Downregulation of *exoc3* yielded an increased planarian neoblast stem cell pool whereas differentiation is blocked. Moreover, by CRISPR/Cas genome engineering a murine Tnfaip2 knockout ES cell line was generated. The study reports that these KO cells do show an aberrant differentiation phenotype as judged by embryoid body (EB) formation assay. Furthermore, the study aims at deciphering the putative mechanism of Tnfaip2 loss-of-function and claims that vimentin expression is controlled by Tnfaip2. This is stated to result in differentiation defects by impairing lipid metabolism. Supplementation of free fatty acids to the cells appears to rescue the differentiation defects of Tnfaip2 KO cells.

The study is clearly written and very well organized. The materials and methods section is sound and seems to be sufficiently comprehensive to allow reproduction of the results. Also the results are clearly structured and well presented. The study would have benefited from more comprehensive studies of mutants at cellular level. E.g. the Tnfaip2 KO is poorly described and no data on functional validation given. The differentiation analyses is based on EB formation and follows a very short protocol of 6 days only. In this respect Fig. 4A might be misleading. Global RNA-seq data is presented that is appropriately being analyzed. However, this has not been carried out in the rescue experiment (Fig. 6), which is solely based on EB formation and a quite rough quantification of morphological characteristics (rosettes of surrounding EBs), representing a rather preliminary state of these findings. A more comprehensive staining against germ layer markers enabling insight into particular differentiation defects would be more informative.

Response: We thank the reviewer for his/her positive comments on our manuscript and for the valuable comments, which we all addressed in the revised manuscript and which significantly improved our study.

We followed the reviewer's suggestion and included a new RNA-sequencing analysis on the rescue of differentiation of *Tnfaip2*^{-/-} EBs by fatty acid (FA) treatment (new Figure 3G; new Expanded View 6A and 6B and new Appendix Table 10; 1st paragraph on page 19 of the result section). The analysis shows that FA treatment rescues the expression of stemness/differentiation regulating genes that are dysregulated in *Tnfaip2*^{-/-} EBs versus WT EBs (revised Figure 2E and revised Expanded View 4A and 4B). We highlight the differentially regulated stemness- and differentiation-associated genes that were differentially expressed in *Tnfaip2*^{-/-} EBs versus WT EBs but rescued in *Tnfaip2*^{-/-} EBs by FA-treatment (marked with asterisk in new Figure 3G and new Expanded View 6A and 6B).

Remark: Gene lists of corresponding Gene Ontology (GO) terms (e.g. “Positive and Negative Regulators of stem cell population maintenance” as well as “Ectodermal and Mesodermal Markers”) were obtained by using AmiGO 2 (data base version 2019-07-02; Carbon et al., 2009). The corresponding figures show the heatmaps of all differentially expressed genes in the respective experiments: Each of the Figures 2E and 3G shows 19 plus 5 genes, but there are differences in the list of genes based on which genes of the list of indicated AmiGO terms were differentially expressed in the respective experiments. Same explanation holds true for the depiction of genes in Expanded View 4A and 4B and Expanded View 6A and 6B.

We have also included immunostaining images with differentiation markers of ectoderm and mesoderm lineages like Brachyury (T) and SOX1 respectively to demonstrate repression of these markers at protein level in cells in *Tnfaip2*^{-/-} EBs as compared to WT EBs (new Expanded View 4C and page 15 of the revised manuscript).

We conducted similar immunofluorescence analysis to demonstrate activation of the T and SOX1 upon supplementation of PA/PC to *Tnfaip2*^{-/-} EBs undergoing *in vitro* differentiation (new Expanded View 6C; 1st paragraph on page 19 of the revised manuscript).

Furthermore, in response to reviewer-1, we also analyzed the role of FA in reprogramming. Specifically, we now show that in line with pro-differentiation influence of PA/PC, treatment with Etomoxir, a CPT1A-specific inhibitor, improves reprogramming efficiency (new Figure 4F and 4G) (2nd paragraph on page 21 of the revised manuscript).

Altogether, we think that our revised data more clearly describe the phenotype of the mutant ES cells versus WT and provide further support for the role of *Tnfaip2* and lipid metabolism in differentiation of ES cells as well as in reprogramming of MEFs into iPS cells.

The discussion is clearly written and based on the experimental observations. The interpretation of experimental results against the published background is mostly sound and conclusive. The findings by Zhang et al. reporting that iPS-derived mesenchymal stem cells show mitochondrial defects after *Tnfaip2* inhibition should have been discussed (Stem Cell Reports, 2016 doi: 10.1016/j.stemcr.2016.08.009).

Answer: The paper refers to the role of *Tnfaip2* in the formation of tunneling nanotubes (TNTs), which can mediate the transfer from mitochondria from one cell to another. It is a very interesting aspect of *Tnfaip2* gene function. Although, not being directly related to our work, we thank the reviewer for pointing this out and we have now incorporated the exciting finding of Zhang et al., 2016 (1st paragraph on page 26 in the discussion section of the revised manuscript).

Minor

points:

- The third sentence in the abstract: "Tnfaip2 knockout, embryonic stem cells (ESCs) exhibit ..." is odd and needs rewording.
- again in the abstract: write reduced expression OF Vimentin (Vim)

Response: We have accordingly edited the abstract (3rd and 4th sentence).

Referee #3:

In this manuscript, the authors report that lipid metabolism mediated by *Tnfaip2*/Exoc3 plays the key role for stem cell differentiation and organ maintenance. Knockdown of *Tnfaip2* was found to promote the iPSCs generation, however, suppression of *Tbfaip2* or its planarian orthologue, *Smed-exoc3*, in stem cells abrogated differentiation and regeneration. Mechanistically, *Tnfaip2* deficiency led to deregulation of TAG synthesis and reduced the expression of Vimentin that is responsible for lipid droplets formation. While the major findings in this study are potentially interesting, the author did not provide sufficient evidence to support the conclusion.

1. sh*Tnfaip2*s were found to enhance the somatic reprogramming process, however, the author also observed that *Tnfaip2*-deficient ESCs exhibited differentiation failure. It is not clear that *Tnfaip2* functions to prevent or promote the acquirement and maintenance of pluripotency. The author should examine the expression level of *Tnfaip2* change during somatic cell reprogramming and ESCs differentiation.

Response: We thank the reviewer for his/her positive comments on our manuscript and for the valuable comments, which we all addressed in the revised manuscript and which significantly improved our study.

Following the insightful comment of the reviewer, we have analyzed the expression of *Tnfaip2* during somatic cell reprogramming and ESC. The experiments show that the expression of *Tnfaip2* increases at later time points during cellular reprogramming, while in iPSCs we detected a profound repression of *Tnfaip2* (new Figure 1E; 2nd paragraph on page 7 of the result section). On the other hand, *Tnfaip2* was found to be activated upon formation of EBs and during the first days of *in vitro* differentiation (new Figure 2F; 1st paragraph on page 15 of the result section).

2. In this study, *Tnfaip2* seemed to play the conflicting roles in somatic reprogramming and stem cell differentiation. It would be interesting to investigate whether the iPSCs that generated from MEFs with OSKM and sh*Tnfaip2* induction exhibit differentiation deficiency.

Response: The reviewer is right that it is possible that the knockdown of *Tnfaip2* could also compromise the differentiation capacity of the iPSCs. However, the knockdown efficiency of *Tnfaip2* was only 70-80% in our studies (revised Expanded View 1B – former Supplementary Figure 1C). The abrogation of differentiation potential by *Tnfaip2* knockdown may in fact be incomplete in these shRNA studies. We think that the experiments on *Tnfaip2* knockout ESCs versus WT controls provide a cleaner experimental system to address this question. We have thus extended the molecular analysis of the *Tnfaip2* KO ESCs vs WT ESCs to gain further insights in the functional role of *Tnfaip2* in differentiation (see new Figure 3G; new Expanded View 6A and 6B and new Appendix Table 10 and page 19 of the revised manuscript).

3. As shown in Figure 6, the supplementation of palmitic acid and palmitoyl-L-carnitine rescued the differentiation defects of *Tnfaip2*/exoc3-deficient pluripotent stem cells in culture and *in vivo* in planarian. The authors concluded that Vim-dependent lipid droplet (LD) formation mediated by *Tnfaip2*/exoc3 was critical for ESC differentiation. However, they did not provide sufficient evidence to support the conclusion. Since Vimentin also is a regulator of EMT process that is well-known to be critical for somatic reprogramming or stem cell differentiation, the author may

wish to go further to prove that Vimentin deficiency lead to failure in lipid droplet formation and then ESCs differentiation.

Response: We thank the reviewer for his/her insightful comment. To further substantiate the conclusion the *Tnfaip2/Exoc3* act through *Vim*-mediated lipid metabolism contributes to the differentiation defect, we have now included a series of new experiments into the revised manuscript:

- (i) We analyzed triacyl glycerol (TAG) in *Smed-exoc3*-depleted planarians compared to controls. The experiment shows a profound reduction in TAGs as in *Tnfaip2* deficient ESC compared to WT controls (new Figure 5A) (3rd paragraph on page 21 of the result section),
- (ii) We conducted RNA-sequencing analysis on the rescue of differentiation of *Tnfaip2*^{-/-} EBs by fatty acid (FA) treatment (new Figure 3G; new Expanded View 6A and 6B and new Appendix Table 10). The analysis shows that FA treatment rescues the expression of stemness/differentiation regulating genes that are dysregulated in *Tnfaip2*^{-/-} EBs versus WT EBs (revised Figure 2E- former Fig. 4A and revised Expanded View 4A and 4B – former Supplementary Figure 5B and 5C). We highlight the differentially regulated stemness/differentiation genes that were differentially expressed in *Tnfaip2*^{-/-} EBs versus WT EBs but rescued in *Tnfaip2*^{-/-} EBs by FA-treatment (marked with asterisk in new Figure 3G and new Expanded View 6A and 6B) (1st paragraph on page 19 of the result section).
- (iii) We combined knockdown of *Tnfaip2* with knockdown of *Vim* to analyze whether the knockdown of both genes would have additive effects in enhancing reprogramming of MEFs into iPSCs (revised Figure 4C and Expanded View 6F – Former Supplementary Figure 6E and 6F). This is not the case supporting the conclusion that *Tnfaip2* dependent effect on *Vim* act in the same pathway to increase reprogramming of MEFs into iPSCs (2nd paragraph on page 19 of the result section).
- (iv) We combined knockdown and overexpression of *Vim* and *Tnfaip2* to analyze whether the two genes act epistatically in influencing reprogramming of MEFs into iPSCs. The experiment shows that, the knockdown of *Tnfaip2* fails to enhance reprogramming when combined with overexpression of *Vim*. In contrast, the knockdown of *Vim* enhances reprogramming in the presence of *Tnfaip2* overexpression (new Figure 4D and 4E). These results support the conclusion that *Vim* acts downstream of *Tnfaip2* in inhibiting reprogramming (page 20 of the result section).

Together, our new data support the conclusion the *Tnfaip2/Vim* mediated lipid metabolism impairs reprogramming of MEFs into iPSCS and is required for differentiation of ESCs in EB cultures. We agree with the reviewer that *Vim* mediated EMT may also be involved in the observed phenotypes. It could even be possible *Tnfaip2/Vim* mediated lipid metabolism is involved in the regulation of EMT by *Vim*. We included this thought in the revised discussion (3rd paragraph on page 23 of the discussion section).

4. The authors claimed *Tnfaip2* regulated the expression of Vimentin and *Cpt1a*. It's better to provide the results of qRT-PCR and western blot besides the proteomics data.

Response: We followed the reviewer's suggestion and analyzed *Vim* expression in *Tnfaip2*^{-/-} EBs during EB differentiation by RT-qPCR (new Appendix Figure 4B; 2nd paragraph on page 16 of the result section). Moreover, we included the expression profile of *Vim* and *Tnfaip2* upon depletion of *Tnfaip2* and *Vim* respectively (new Appendix Figure 5A and 5B). The experiments show that knockdown of *Tnfaip2* led to suppression of *Vim* (new Appendix Figure 5A), whereas the knockdown of *Vim* didn't have a significant effect on *Tnfaip2* expression (new Appendix Figure 5B; page 20 of the result section).

We have also included RT-qPCR based analysis of *Cpt1a* expression in *Tnfaip2*^{-/-} EBs during EB differentiation (new Expanded View 5D). The analysis shows a reduction of *Cpt1a* expression (new Expanded View 5D, 3rd paragraph on page 18 of the result section). Overall these data support the conclusion that *Tnfaip2* regulates *Vim* and *Cpt1a* expression.

In addition, the revised manuscript also provides functional prove for *Cpt1a* dependent effect of *Tnfaip2* on enhancing the reprogramming efficiency of MEFs into iPSCs. A new experiment was conducted to treat shScr infected MEFs and shTnfaip2 infected MEFs that were also infected with a reprogramming vector with Etomoxir, a specific *Cpt1a* inhibitor. While treatment of shScr MEFs with Etomoxir enhanced reprogramming efficiency to a similar extent as the shTnfaip2-infection of MEFs, treatment with Etomoxir had no effect on the reprogramming of shTnfaip2-infected MEFs (new Figure 4F and 4G). Together, these data support the conclusion that the induction of *Cpt1a* contributes to *Tnfaip2*-mediated suppression of reprogramming (2nd paragraph on page 21 of the result section).

5. Another critical missing part is how Tnfaip2/exoc3 regulate the protein levels of Vimentin or Cpt1a. In Supplementary Figure 6E, the author claimed that there was **no additive effect** in the enhancement of reprogramming efficiency upon co-depletion of Tnfaip2 and Vim indicating that Tnfaip2 and Vim act especially in this regard. It is not surprising that knockdown of Vimentin, the regulator of EMT, significantly increased the somatic reprogramming efficiency. The author may wish to provide more evidence to claim Vimentin is the downstream effector of Tnfaip2. It is not clear whether Vimentin could rescue the differentiation defect of ESCs or somatic stem cells with Tnfaip2/exoc3.

Response: We thank the reviewer for his/her insightful comment. We included a new experiment on reprogramming to functionally address this question. We combined the knockdown of *Vim* or *Tnfaip2* with the overexpression of *Tnfaip2* or *Vim* to analyze whether the two genes act epistatically and which one is upstream in influencing reprogramming of MEFs into iPSCs. The experiment shows that *Vim* acts downstream of *Tnfaip2* in inhibiting reprogramming (new Figure 4D and 4E). Specifically, the knockdown of *Tnfaip2* fails to enhance reprogramming when combined with overexpression of *Vim*. In contrast, the knockdown of *Vim* enhances reprogramming in the presence of *Tnfaip2* overexpression (new Figure 4D and 4E). These results support the conclusion that *Vim* acts downstream of *Tnfaip2* in inhibiting reprogramming (page 20 of the result section).

6. In Figure 3C&D, Smed-exoc3 knockdown increased the numbers of proliferating stem cells in planarian. The authors explained that Smed-exoc3 knockdown led to the self-renewal versus differentiation of stem cells. Nevertheless, it is known that the asymmetrical and symmetrical division is essential for stem cell maintenance and tissue regeneration. The authors may wish to address whether stem cells with Smed-exoc3 knockdown prefer symmetrical division rather than asymmetrical division and whether the lipid drop formation affects the division pattern.

Response: This is an excellent point and a very interesting question. However, the current technology in the field of planarian research does not allow us to address this question. The reviewer has sparked our interest in this question, and we are currently planning to move into genetic mouse studies to address this question. In mice, it is possible to get definitive answers on asymmetric vs. symmetric stem cell division by using transplantation assays, such as the paired-daughter assay making use of single cell transplants of hematopoietic stem cell that have undergone one cell division in culture. We are interested in doing this but we cannot conduct this under the time frame of the current study and hope for the understanding of the reviewer.

Dear Lenhard,

Thank you for the submission of your revised manuscript. We have now received the comments from both referees and I am happy to tell you that both are overall happy with the revisions of your study.

However, referee 1 still raises several important points that will all need to be addressed for publication of your manuscript here. I would thus like to invite you to address these remaining concerns and send us a newly revised manuscript as soon as possible.

A few other changes will also be required:

Given that important data seem to be present in the supplementary information, I suggest that you add more main figures and change the format of your manuscript to a full-length Article with separate results and discussion sections.

We do agree with referee 1 that the possibilities for adding extra/supplementary information to our manuscripts might be confusing, and we are currently working on this issue. I suggest that your excel files with new data in it should be called Dataset EV1, etc. You can have as many EV Datasets, tables and movies as required, but the number of EV figures is restricted to a maximum of 5. Also, you can move all other supplementary data to the Appendix pdf file, but they need to fit into a pdf file, and no large excel tables can be included. The Appendix also needs a table of content with page numbers and the figures are called Appendix Figure S1, etc. Please correct all figures names and cite all items in the main manuscript file. You can also find all info about our file types in our guide to authors online.

Please reduce the list of keywords to 5.

In the authors contributions there are 2 AGs, which should be differentiated by using the 2nd letter of the surname.

Our reference format has changed to Harvard style, please correct.

I attach to this email a related manuscript file with comments from our data editors. Please address all comments in the final manuscript.

All publicly deposited data - such as the shRNA sequences data - must be accessible before the online publication of your study.

Please avoid overstatements in the abstract.

EMBO press papers are accompanied online by A) a short (1-2 sentences) summary of the findings and their significance, B) 2-3 bullet points highlighting key results and C) a synopsis image that is 550x200-600 pixels large (the height is variable). You can either show a model or key data in the synopsis image. Please note that text needs to be readable at the final size. Please send us this information along with the revised manuscript.

I look forward to seeing a final form of your manuscript when it is ready.

Referee #1:

The revised version of the manuscript includes substantial amounts of new data and text revisions. Important improvements include the additional cell culture RNAseq data, the quantification of triglycerides in *exoc3*(RNAi) planarians and text changes that better integrate the two model systems. However, the manuscript is still not publication-ready. Remaining issues include:

1) The split of the supplemental material into "Expanded View" and "Appendix Figures" is entirely unserviceable to both reviewers and potential readers- irrespective of whether this originates with the journal or the authors, the final version of this manuscript needs to have a single, consolidated list of supplemental figures.

2) I remain skeptical that the claim of "new pathway discovery" is justifiable on basis of the data. To me at least, a "pathway" is a chain of multiple cause-consequence interactions that collectively explain a cellular phenotype. To claim a pathway on basis of two proteins seems a bit of a stretch, especially if their mechanistic interplay is entirely obscure and quite possibly indirect, as in the case of *vim* and *tnfaip2*.

The conceptual lynchpin of the planarian section, namely the demonstration of a stem cell differentiation failure in *exoc3*(RNAi) planarians, still remains rather circumstantial. The added analyses are lacking important controls or are poorly presented and the bits of relevant data are buried deeply within the supplement. Specific problems include:

3) The FACS quantification that the authors added to corroborate the accumulation of stem cells at the expense of stem cell progeny/differentiated cells is lacking essential controls. Without an irradiation control to verify the positioning of the X2/Xins gates and the low number of replicates relative to the inter-replicate variation, apparent population shifts are just as likely to reflect the shift in the X1 population or technical noise, rather than a depletion of "differentiated" cells. The apparently 7.3% Xins cells in *exoc3*(RNAi) animals in the day 34 FACS plot shown are a case in point, as an animal comprised of ~93% stem cells and early progeny would hardly be able to maintain the degree of anatomical organization evident in Fig. 1G or 2A-C. Bottom line: As is, the data insufficiently supports the claimed decrease in "differentiated" cells. Instead, the authors might want to consider integrating some of the FACS data into figure 1 as additional evidence for the relative accumulation of stem cells.

4) Although the qPCR analysis of progeny markers NB.21.11e (now generally referred to as *prog-1* by the community- the authors might want to cite the relevant study) indeed documents an initial decrease of *prog-1* expression, expression recovers at later time points in spite of the continued increase in stem cell marker genes - this is inconsistent with the assumed depletion of differentiating cells at the expense of stem cells and this caveat needs to be acknowledged.

5) The new experiments meant to support that the increased number of H3P-positive cells reflects an increase in the numbers of normally cycling stem cells rather than abnormal cycling of an unchanged number of stem cells, are inconclusive at best. The level of apoptosis within the mitotic cell population can hardly provide strong evidence for one or the other and without a positive control, it remains doubtful whether the author's FACS assay would have sufficient sensitivity for picking up a small and transient fraction of apoptotic cells amongst the sorted X1 cells. Similarly, the lack of upregulation of the "bona-fide mitotic checkpoint gene p53/p63" simply cannot be cited as evidence without a prior demonstration that planarian p53/p63 is indeed upregulated in mitosis. In short: These data weaken the manuscript, rather than strengthening it.

6) Once again, please drop the claim of reduced protonephridial density on basis of the images in Fig. 2B and appendix Fig. 2- protonephridia remain present and the difference in protonephridial density in the shown images are within the variation range that can be expected on basis of technical noise/inter-animal variation or that can be attributed to the evident head regression. Plus, protonephridial defects tend to manifest in edema, which is not evident in Fig. 1G. Bottom line: Either remove this data or present quantitative evidence in the sense of numbers of protonephridial units/projected area.

7) Possibly the strongest evidence for a stem cell differentiation defect in exoc3(RNAi) is buried within the expanded RNAseq analyses in the appendix figure and the appendix tables. I write possibly, because as is, the data is imply shard to fathom. Specifically:

- Please specify whether the exoc3(RNAi) X1 fractions were compared to X1 fractions of GFP(RNAi) controls or to whole GFP(RNAi) animals- this remains unclear and evidently important for the interpretation of the data.
- Please label the bar graphs with the specific gene category, rather than the study authors (the refs should go in the figure legend). Moreover, the text remains murky regarding the original definition of the "differentiation gene sets"- this is essential information for judging the relevance of the analysis and the authors cannot expect readers to scrutinize the original publications for this information.
- Provided that the analyzed gene sets are indeed enriched for bona-fide differentiation associated genes, the authors need to present the data in a way that directly visualizes the down- or upregulation of specific genes (rather than merely the fraction of differentially expressed genes within the set, as in the present figures). The heat maps in the stem cell sections of the manuscript provide one example of how to accomplish this.
- I remain puzzled as to why the authors restricted their analysis to FACS sorted neoblasts, rather than simply comparing gene expression differences between whole exoc3(RNAi)/GFP(RNAi) animals. This would have afforded a direct opportunity to quantify global up/down regulation of stem cell genes and/or progeny markers, while the present X1 analysis can only detect relative shifts within the population. e.g., differences in cell cycle progression...

Though the link between exoc3 and planarian stem cell differentiation remains weak, the quantitative depletion of triglycerides and the phenotypic rescue by FA supplementation nicely complements the vertebrate stem cell data. I can therefore cautiously recommend publication of a substantially revised manuscript, provided that the authors i) re-work their added planarian exoc3(RNAi) RNAseq analysis to incorporate the above concerns; ii) ensure that the salient bits of data are in the main figures and not in supplement; iii) omit the inconclusive support pertaining to a lack of cell cycle effects (p53/p63; annexin FACS plots); iv) undertake appropriate text revisions to ensure congruency between data shown and claims made (e.g., already in the abstract: "...Smed-exoc3 abrogates in vivo differentiation of somatic stem cells...").

Minor point:

8) Once again, my request for "Scientific terminology" refers to QUANTITATIVE statements- please replace "good knock-down" by something along the lines of "... reduced mRNA levels by > 70 % (reference to figure) in comparison to untreated controls".

9) Please omit the statement that you carried out a "lipidomic" analysis- the targeted quantification of triglycerides is not a lipidome analysis.

Referee #3:

The authors have addressed all my concerns. The reviewer has no more question.

Referee #1:

The revised version of the manuscript includes substantial amounts of new data and text revisions. Important improvements include the additional cell culture RNAseq data, the quantification of triglycerides in exoc3(RNAi) planarians and text changes that better integrate the two model systems. However, the manuscript is still not publication-ready. Remaining issues include:

1) The split of the supplemental material into "Expanded View" and "Appendix Figures" is entirely unserviceable to both reviewers and potential readers- irrespective of whether this originates with the journal or the authors, the final version of this manuscript needs to have a single, consolidated list of supplemental figures.

Response: We followed the reviewer and have included all the "Appendix Figures" into main or EV Figures. The revised manuscript now includes only Main Figures (12) and Expanded View Figures (3). In addition, the manuscript includes 10 datasets showing Excel tables underlying the data analysis.

2) I remain skeptical that the claim of "new pathway discovery" is justifiable on basis of the data. To me at least, a "pathway" is a chain of multiple cause-consequence interactions that collectively explain a cellular phenotype. To claim a pathway on basis of two proteins seems a bit of a stretch, especially if their mechanistic interplay is entirely obscure and quite possibly indirect, as in the case of vim and tnfaip2.

Response: We have addressed this point throughout the manuscript by:

(i) referring to a "new mechanism" instead of a "new pathway" that controls ES cell differentiation and organ maintenance in planarians (see yellow highlights in the abstract and 2nd paragraph and throughout the manuscript).

and

(ii) avoiding statements on stem cell differentiation in planarians by rather referring to defects in organ maintenance (see yellow highlights in the abstract and 2nd paragraph and throughout the manuscript). The reviewer is right that cell cycle arrest could also affect the production of differentiated cells. As these two processes are interconnected (proliferation and differentiation) and since we cannot exclude the possibility of cell cycle being involved in the observed phenotypes, we have reworded the manuscript throughout.

The conceptual lynchpin of the planarian section, namely the demonstration of a stem cell differentiation failure in exoc3(RNAi) planarians, still remains rather circumstantial. The added analyses are lacking important controls or are poorly presented and the bits of relevant data are buried deeply within the supplement. Specific problems include:

3) The FACS quantification that the authors added to corroborate the accumulation of stem cells at the expense of stem cell progeny/differentiated cells is lacking essential controls. Without an irradiation control to verify the positioning of the X2/Xins gates and the low number of replicates relative to the inter-replicate variation, apparent population shifts are just as likely to reflect the shift in the X1 population or technical noise, rather than a depletion of

"differentiated" cells. The apparently 7.3% Xins cells in *exoc3*(RNAi) animals in the day 34 FACS plot shown are a case in point, as an animal comprised of ~93% stem cells and early progeny would hardly be able to maintain the degree of anatomical organization evident in Fig. 1G or 2A-C. Bottom line: As is, the data insufficiently supports the claimed decrease in "differentiated" cells. Instead, the authors might want to consider integrating some of the FACS data into figure 1 as additional evidence for the relative accumulation of stem cells.

Response: We followed the reviewer's suggestion and integrated the FACS data on the relative number of neoblast stem cell increases in *Smed-exoc3*-depleted planarians into the main figure (Figure 3C). Other FACS data were removed.

4) Although the qPCR analysis of progeny markers NB.21.11e (now generally referred to as *prog-1* by the community- the authors might want to cite the relevant study) indeed documents an initial decrease of *prog-1* expression, expression recovers at later time points in spite of the continued increase in stem cell marker genes - this is inconsistent with the assumed depletion of differentiating cells at the expense of stem cells and this caveat needs to be acknowledged.

Response: We followed the reviewer's suggestion: In the revised manuscript *NB.21.11e* and *NB.32.1g* are referred to as *prog-1* and *prog-2*, respectively. Regarding the statement on the expression level of *prog-1* (*NB.21.11e*) at later time points (Appendix Figure 1C – now Figure 3F), we respectfully disagree with the reviewer indicating that this result would be inconsistent with our interpretation of the data. We agree that there is an increase in the expression of *prog-1* at later time point after RNAi (day 38). However, the level of *prog-1* expression in *Smed-exoc3*-depleted worms at this timepoint still remains significantly lower than the level in control worms ($p=0.0096$, new Figure 3F). Thus, the result on the impaired expression of this marker gene of differentiation stands very much in line with reduced generation of differentiated cells in *Smed-exoc3*-depleted worms versus controls.

5) The new experiments meant to support that the increased number of H3P-positive cells reflects an increase in the numbers of normally cycling stem cells rather than abnormal cycling of an unchanged number of stem cells, are inconclusive at best. The level of apoptosis within the mitotic cell population can hardly provide strong evidence for one or the other and without a positive control, it remains doubtful whether the author's FACS assay would have sufficient sensitivity for picking up a small and transient fraction of apoptotic cells amongst the sorted X1 cells. Similarly, the lack of upregulation of the "bona-fide mitotic checkpoint gene p53/p63" simply cannot be cited as evidence without a prior demonstration that planarian p53/p63 is indeed upregulated in mitosis. In short: These data weaken the manuscript, rather than strengthening it.

Response: We followed the reviewer's suggestion and removed this part of the FACS analysis as well as the p53 data from the revised manuscript.

6) Once again, please drop the claim of reduced protonephridial density on basis of the images in Fig. 2B and appendix Fig. 2- protonephridia remain present and the difference in protonephridial density in the shown images are within the variation range that can be expected on basis of technical noise/inter-animal variation or that can be attributed to the evident head regression. Plus, protonephridial defects tend to manifest in edema, which is not evident in Fig. 1G. Bottom line: Either remove this data or present quantitative evidence in the sense of numbers of protonephridial units/projected area.

Response: We followed the advice of the reviewer to remove this part of the analysis of impaired organ homeostasis in *Smed-exoc3*-depleted planarians, since the remaining data on the depletion of neuronal cells (new Figure 4A) and photosensitive cells (new Figure 4B) are fully sufficient to make this point.

7) Possibly the strongest evidence for a stem cell differentiation defect in *exoc3*(RNAi) is buried within the expanded RNAseq analyses in the appendix figure and the appendix tables. I write possibly, because as is, the data is imply shard to fathom. Specifically:
a. Please specify whether the *exoc3*(RNAi) X1 fractions were compared to X1 fractions of GFP(RNAi) controls or to whole GFP(RNAi) animals- this remains unclear and evidently important for the interpretation of the data.

Response: We clarified the description that we indeed compared *exoc3*(RNAi) X1 fractions to *gfp*(RNAi) X1 fractions in these experiments. We have now also included this information to the figure legends of the new Figure 5 and new EV2.

b. Please label the bar graphs with the specific gene category, rather than the study authors (the refs should go in the figure legend). Moreover, the text remains murky regarding the original definition of the "differentiation gene sets"- this is essential information for judging the relevance of the analysis and the authors cannot expect readers to scrutinize the original publications for this information.

Response: We have followed this suggestion from the reviewer. The information was given in the result text but we have now included the original names of the gene sets along with the publication in the revised Figure labeling (see new Figure 5A). We agree that this is much clearer.

c. Provided that the analyzed gene sets are indeed enriched for bona-fide differentiation associated genes, the authors need to present the data in a way that directly visualizes the down- or upregulation of specific genes (rather than merely the fraction of differentially expressed genes within the set, as in the present figures). The heat maps in the stem cell sections of the manuscript provide one example of how to accomplish this.

Response: This was a very good suggestion and we have followed the reviewer's advice. In the revised manuscript we have included heatmaps depicting the overlap of genes that (i) are regulated in neoblast stem cells of *Smed-exoc3*-depleted worms versus controls and (ii) are expressed in neoblast stem cells (Plass et al., 2018; Fincher et al., 2018; Solana et al., 2012) or in differentiated cells (Zhu et al., 2015). These heatmaps depict whether the genes in the overlap are up or downregulated in response to *Smed-exoc3*-depletion. For readability we have focused the heatmap depiction on those genes that show the strongest difference in expression in response to *Smed-exoc3*-depletion (\log_2 fold-change) > 0.5 or < -0.5 (new Figure 5B-E; last paragraph on page 10 to 1st paragraph on page 12 of the result section). In addition, the full gene list of the overlaps is provided in the Dataset EV 4 and 6. In these datasets, we have included an analysis of the literature highlighting the function of the *Smed-exoc3*-regulated genes in stem cells and differentiation (Column R of the dataset). This literature analysis was focused on those genes showing the strongest difference in expression in response to *Smed-exoc3*-depletion (\log_2 fold-change) > 0.5 or < -0.5 as depicted in Dataset EV 4 and 6. We think that this new depiction is much better and very interesting for the readers.

While all of the differentiation marker genes (from the Zhu et al.) were downregulated in *Exoc3*-depleted planarians vs controls (Figure 5E), the neoblast related genes were both up-

and downregulated in *Exoc3*-depleted planarians. We think that this finding stands in line with the possibility (suggested by this reviewer) that the neoblast cells are possibly arrested at certain transitional states between stemness and differentiation thus leading to an up- or downregulation of the respective genes that characterize these transition stages. We included this thought in the result description (highlighted on page 11 of the revised manuscript).

d. I remain puzzled as to why the authors restricted their analysis to FACS sorted neoblasts, rather than simply comparing gene expression differences between whole *exoc3*(RNAi)/GFP(RNAi) animals. This would have afforded a direct opportunity to quantify global up/down regulation of stem cell genes and/or progeny markers, while the present X1 analysis can only detect relative shifts within the population. e.g., differences in cell cycle progression...

Response: We agree with the reviewer that RNA-seq on whole animals would have helped to confirm the accumulation of stem cells in *Smed-exoc3*-depleted planarians. However, this kind of an analysis was already included (new Figure 2C and 3E,F).

In our analysis on transcriptome changes, we actually intended to analyze gene expression in neoblast stem cells to see whether defects in the generation of differentiated cells were associated with changes in gene expression in neoblast cells from *Smed-exoc3*-depleted vs. wildtype worms, which turned out to be the case. We think that the new depiction (see above response) makes this analysis even more interesting as it shows that *Smed-exoc3*-depletion leads to a deregulation of genes that have a role in stem cell function and differentiation. We agree with the reviewer that this deregulation could have multiple reasons including effects of *Exoc3*-deletion on cell cycle progression of neoblast stem cells.

Though the link between *exoc3* and planarian stem cell differentiation remains weak, the quantitative depletion of triglycerides and the phenotypic rescue by FA supplementation nicely complements the vertebrate stem cell data. I can therefore cautiously recommend publication of a substantially revised manuscript, provided that the authors i) re-work their added planarian *exoc3*(RNAi) RNAseq analysis to incorporate the above concerns; ii) ensure that the salient bits of data are in the main figures and not in supplement; iii) omit the inconclusive support pertaining to a lack of cell cycle effects (p53/p63; annexin FACS plots); iv) undertake appropriate text revisions to ensure congruency between data shown and claims made (e.g., already in the abstract: "...*Smed-exoc3* abrogates in vivo differentiation of somatic stem cells...").

Response: We thank the reviewer for being supportive for publication and we cautiously addressed all his/her comments as outlined in the above responses and as highlighted throughout the revised manuscript.

Minor point:

8) Once again, my request for "Scientific terminology" refers to QUANTITATIVE statements- please replace "good knock-down" by something along the lines of "... reduced mRNA levels by > 70 % (reference to figure) in comparison to untreated controls".

Response: We have specified this statement as suggested (1st paragraph on page 9).

9) Please omit the statement that you carried out a "lipidomic" analysis- the targeted quantification of triglycerides is not a lipidome analysis.

Response: We have edited the sentence accordingly (2nd paragraph on page 16).

Prof. Karl-Lenhard Rudolph
Leibniz Institute on Aging - Fritz Lipmann Institute (FLI)
Stem Cell Aging
Beutenbergstraße 11
Jena 07745
Germany

Dear Prof. Rudolph,

I am very pleased to accept your manuscript for publication in the next available issue of EMBO reports. Thank you for your contribution to our journal.

At the end of this email I include important information about how to proceed. Please ensure that you take the time to read the information and complete and return the necessary forms to allow us to publish your manuscript as quickly as possible.

Please note that under the DEAL agreement of German scientific institutions with our publisher Wiley, you could be eligible for publication of your article in the open access format in a way that is free of charge for the authors. Please contact either the administration at your institution or our publishers at Wiley (emboreports@wiley.com) for further questions.

As part of the EMBO publication's Transparent Editorial Process, EMBO reports publishes online a Review Process File to accompany accepted manuscripts. As you are aware, this File will be published in conjunction with your paper and will include the referee reports, your point-by-point response and all pertinent correspondence relating to the manuscript.

If you do NOT want this File to be published, please inform the editorial office within 2 days, if you have not done so already, otherwise the File will be published by default [contact: emboreports@embo.org]. If you do opt out, the Review Process File link will point to the following statement: "No Review Process File is available with this article, as the authors have chosen not to make the review process public in this case."

Should you be planning a Press Release on your article, please get in contact with emboreports@wiley.com as early as possible, in order to coordinate publication and release dates.

Thank you again for your contribution to EMBO reports and congratulations on a successful publication. Please consider us again in the future for your most exciting work.

THINGS TO DO NOW:

You will receive proofs by e-mail approximately 2-3 weeks after all relevant files have been sent to our Production Office; you should return your corrections within 2 days of receiving the proofs.

Please inform us if there is likely to be any difficulty in reaching you at the above address at that time. Failure to meet our deadlines may result in a delay of publication, or publication without your corrections.

All further communications concerning your paper should quote reference number EMBOR-2019-49328V3 and be addressed to emboreports@wiley.com.

Should you be planning a Press Release on your article, please get in contact with emboreports@wiley.com as early as possible, in order to coordinate publication and release dates.

Corresponding Author Name: Karl Lenhard Rudolph

Manuscript Number: EMBOR-2019-49328V1